# CausalProfiler: Generating Synthetic Benchmarks for Rigorous and Transparent Evaluation of Causal Machine Learning

**Panayiotis Panayiotou** [* 1]   **Audrey Poinsot** [* 2 3]   **Alessandro Leite** [4]
**Nicolas Chesneau** [2]   **Marc Schoenauer** [3]   **Özgür Şimşek** [1]

## Abstract

Causal machine learning aims to answer "what if" questions using machine learning algorithms, making it a promising tool for high-stakes decision-making. Yet, empirical evaluation practices remain limited. Existing benchmarks often rely on a handful of hand-crafted or semi-synthetic datasets, leading to brittle, non-generalizable conclusions. To bridge this gap, we introduce CausalProfiler, a synthetic benchmark generator for causal machine learning methods. Based on a set of explicit design choices about the class of causal models, queries, and data considered, CausalProfiler randomly samples causal models, data, queries, and ground truths constituting the synthetic causal benchmarks. In this way, causal machine learning can be rigorously and transparently evaluated under a variety of conditions. This work offers the first random generator of synthetic causal benchmarks with coverage guarantees and transparent assumptions operating on the three levels of causal reasoning: observation, intervention, and counterfactual. We demonstrate its utility by evaluating several state-of-the-art methods under diverse conditions and assumptions, both in and out of the identification regime, illustrating the types of analyses and insights CausalProfiler enables.

## 1. Introduction

Causal machine learning (Causal ML) seeks to estimate the effects of interventions and counterfactuals using machine learning techniques (Kaddour et al., 2022), enabling principled decision making in many fields, for example, in medicine and public policy. Despite the theoretical maturity and growing relevance of Causal ML, current research practices lack rigorous evaluations of how proposed methods would perform under realistic and diverse conditions, limiting their practical utility (Curth et al., 2024; Feuerriegel et al., 2024; Poinsot et al., 2025; Berrevoets et al., 2024).

Real-world evaluation of Causal ML is particularly challenging due to the unobservability of counterfactual outcomes (Holland, 1986). Researchers can rely only on limited real-world data sources, such as randomized controlled trials, which are expensive, ethically constrained, and often yield little data (Greenland & Brumback, 2002; Tennant et al., 2021). As a result, existing benchmarks often rely on a small number of semi-synthetic datasets, such as Syntren (den Bulcke et al., 2006) and ACIC2016 (Dorie et al., 2019), or on model-driven synthetic datasets generated from fitted causal mechanisms (Neal et al., 2020; Parikh et al., 2022; Athey et al., 2024; de Vassimon Manela et al., 2024). These datasets inevitably encode modeling assumptions that are rarely made explicit and whose validity is difficult to assess beyond the original study context (Poinsot et al., 2025). In parallel, many works rely on handcrafted synthetic datasets, useful for supporting theoretical arguments but fragile for empirical evaluation: a few manually selected models can overstate performance by aligning with method-specific assumptions (Gentzel et al., 2019; Gamella et al., 2025; Reisach et al., 2021; Herdeanu et al., 2026). More broadly, lessons from predictive machine learning demonstrate that narrow, static benchmarks can give a false sense of reliability (Geirhos et al., 2020; Herrmann et al., 2024; Freiesleben & Grote, 2023; Longjohn et al., 2024). Together, these observations underscore the need for structured diversity: systematic variation of tasks under explicit, controllable assumptions.

Here, we take a concrete step toward addressing these fundamental concerns about evaluation in the field. Specifically, we introduce a synthetic benchmark generator, called *CausalProfiler,* that enables robust empirical evaluations grounded in transparently-defined, synthetic causal datasets.

---

[1]Department of Computer Science, University of Bath, UK [2]Ekimetrics, Paris, France [3]TAU, LISN, Inria Saclay, France [4]INSA Rouen Normandie, Normandie University, LITIS, Rouen, France. Correspondence to: Panayiotis Panayiotou <pp2024@bath.ac.uk>, Audrey Poinsot <audrey.poinsot@ekimetrics.com>.

*Proceedings of the $43^{rd}$ International Conference on Machine Learning*, Seoul, South Korea. PMLR 306, 2026. Copyright 2026 by the author(s).

Central to our approach is the notion of a *Space of Interest (SoI)* (Definition 5.1), defining the domain from which causal datasets are sampled. Given an *SoI*, CausalProfiler samples Structural Causal Models (SCMs), data, and queries, and estimates the ground truth value of the queries to enable the evaluation of Causal ML methods. The assumptions are explicit, and dataset characteristics can be systematically varied through the *SoI*. Hence, CausalProfiler enables transparent, controlled, repeatable, and diverse sampling of synthetic causal datasets.

CausalProfiler shifts the focus of empirical evaluation from performance on individual datasets to trends and patterns across a *SoI*, reframing the evaluation question from "what dataset to use" to specifying a *SoI* that defines the scope of evaluation. This enables researchers to evaluate performance across a well-defined set of conditions—on graph density, or causal mechanisms complexity, for instance—to understand how accurately a method estimates causal targets when its assumptions for identifcation hold, and how it behaves under controlled violations of those assumptions. Compared to conventional evaluations in the current literature, using CausalProfiler yields more robust and reliable performance estimates by enabling the systematic exploration of failure modes, generalization limits, and assumption sensitivities that would otherwise remain hidden.

Although synthetic evaluation cannot replace real data, it offers the only reliable access to ground-truth causal queries, since counterfactuals are unobservable and many assumptions are unfalsifiable (Holland, 1986). CausalProfiler brings a much-needed complement to real-world studies by enabling transparent, diverse, and controlled synthetic experiments to support method development. Rather than serving as a benchmark for direct real-world deployment, CausalProfiler is intended as an evaluation framework for detailed and assumption-aware analysis of causal methods under controlled conditions.

We make two primary contributions. First, we present CausalProfiler[1] , the first open-source benchmark generator that enables principled sampling of synthetic causal datasets with coverage guarantees, supporting transparency and reproducibility in Causal ML evaluation across the three levels of causal reasoning: observation, intervention, and counterfactual. Secondly, we demonstrate through experiments that evaluation with CausalProfiler yields richer and more robust insights than the current standard practice.

## 2. Related Work

Despite methodological advances, Causal ML lacks a rigorous and systematic paradigm for empirical evaluation.

[1]The code is available at https://github.com/panispani/causal-profiler

Existing benchmarks largely fall into two categories. Semi-synthetic datasets, such as synthetic outcome datasets (Dorie et al., 2019; Shimoni et al., 2018; Hill, 2011) and model-based semi-synthetic datasets (Neal et al., 2020; Parikh et al., 2022; Athey et al., 2024; de Vassimon Manela et al., 2024), combine real covariates and simulated outcomes under assumed structural models. These approaches aim to preserve realistic covariate distributions while enabling partial access to ground truth. Fully synthetic datasets, in contrast, are generated entirely from researcher-defined SCMs, allowing for greater control over data-generating mechanisms and access to ground truth. While these approaches differ in their trade-offs, both have important limitations.

Synthetic evaluations often lack realism, relying on overly simplistic mechanisms such as additive noise or linear functions, and frequently omitting robustness analyses (Gentzel et al., 2019; Curth et al., 2024; Poinsot et al., 2024; 2025). Such evaluations rarely reflect the complexity of real-world causal processes and are insufficient to test the limits of causal inference methods.

Both synthetic and semi-synthetic datasets are shaped by researcher-defined design choices, such as the causal graph structure, the form of the outcome function, and the noise distribution. These decisions, often made implicitly, can introduce hidden biases that favor certain methods (Curth et al., 2021; Cheng et al., 2022; Feuerriegel et al., 2024). Such assumptions are rarely documented or systematically varied, hindering reproducibility and fair method comparison (Poinsot et al., 2024; 2025).

In addition, existing benchmarks are typically small in scale and narrow in scope, often covering only a limited range of causal settings, raising concerns on overfitting and generalization (Gentzel et al., 2019; Berrevoets et al., 2024). For instance, it has been shown that even minor changes to the data-generating process can lead to dramatic shifts in performance rankings (Curth et al., 2021).

Finally, methods are often evaluated only under conditions that guarantee identifiability, offering little insight into robustness under assumption violations, as is common in real-world settings (Petersen, 2024; Hutchinson et al., 2022).

In short, without broader and more transparent evaluation across diverse causal settings, the field risks drawing conclusions that do not generalize. For Causal ML to have a wide practical impact, there is a need to move beyond fixed benchmarks toward evaluation frameworks that enable transparent, controlled, and diverse experimentation across well-defined spaces of causal assumptions.

Recent works have sought to address some of these problems, introducing tools to generate synthetic SCMs for causal discovery (Kalainathan et al., 2020; Gupta et al., 2023; Rudolph et al., 2023) and support query estimation

from hand-specified models (Sharma & Kiciman, 2020; Textor et al., 2017; Abril-Pla et al., 2023). However, none of these approaches support all components required for robust evaluation of Causal ML methods. First, the causal discovery benchmarks do not provide ground truth for interventional or counterfactual queries. Further, query estimation frameworks typically require manual SCM specification and do not support random sampling, diversity control, or analysis of the resulting task distribution. In contrast, CausalProfiler integrates SCM sampling, query ground-truth computation, and coverage guarantees into a unified framework. To the best of our knowledge, this is the first benchmark generator that enables systematic exploration of how Causal ML methods behave across spaces of SCMs and queries defined by user-specified constraints.

## 3. Background & Notation

We use capital letters (e.g., $X$) for random variables, lowercase (e.g., $x$) for realizations, boldface (e.g., $\mathbf{x}, \mathbf{X}$) for vectors and sets, and double-stroke capital letters (e.g., $\mathbb{M}$) for classes.

The Pearl Causal Hierarchy (PCH) (Pearl & Mackenzie, 2018) classifies causal reasoning into three levels: associational ($\mathcal{L}_1$), interventional ($\mathcal{L}_2$), and counterfactual ($\mathcal{L}_3$). Associative questions use only observed data, whereas interventional and counterfactual questions require assumptions about the data-generating process. Importantly, lower levels are insufficient to answer higher-level questions in almost all causal models (Bareinboim et al., 2022).

*Structural Causal Models* (SCMs) (Pearl, 2009) provide a representation that allows reasoning on the three levels of the PCH. An SCM is a tuple $\mathcal{M} = (\mathbf{V}, \mathbf{U}, \boldsymbol{\mathcal{F}}, P(\mathbf{U}))$, where $\mathbf{V}$ is a set of endogenous variables, $\mathbf{U}$ is a set of exogenous variables, $\boldsymbol{\mathcal{F}}$ is a set of structural equations, $V_i = f_i(\boldsymbol{PA}(V_i), \mathbf{U}_{V_i})$, called *causal mechanisms*, and $P(\mathbf{U})$ defines a distribution over the exogenous variables $\mathbf{U}$. SCMs induce a distribution $P_{\mathcal{M}}(\mathbf{V})$ over the endogenous variables $\mathbf{V}$, called the *entailed distribution*. We consider two types of endogenous variables: the observed variables, denoted $\mathbf{V}_O$, and the unobserved variables, denoted $\mathbf{V}_H$, where $\mathbf{V} = \mathbf{V}_O \cup \mathbf{V}_H$ and $\mathbf{V}_O \cap \mathbf{V}_H = \emptyset$.

The *causal graph* is an acyclic directed mixed graph over the endogenous variables. Directed edges $X \rightarrow Y$ encode causal dependencies via causal mechanisms, where $X \in \boldsymbol{PA}(Y)$ is called a parent of $Y$, and bidirected edges $X \leftrightarrow Y$ indicate latent confounding due to shared exogenous causes.

An *intervention* replaces one or more structural equations to model external manipulations. A common example is a *hard intervention*, $\boldsymbol{do}(T = t)$, which fixes a variable's value, disconnecting it from its causes. This defines a new SCM and alters the induced distribution. *Counterfactual*

questions build on interventions: given an observed realization, called the *factual* realization, they ask what would have happened under an intervention different from the one actually taken. They are evaluated by conditioning on observed variables (abduction), modifying the SCM with the intervention (action), and predicting outcomes under the new distribution (prediction)—a process known as the *three-step procedure* (Pearl, 2009).

A *causal query* refers to a probabilistic statement about the effect of hypothetical manipulations of the data-generating process. This includes intervention queries, such as the Average Treatment Effect (ATE), and counterfactual queries, such as the Counterfactual Total Effect (Ctf-TE) (Plečko & Bareinboim, 2024). A query is *identifiable* if its value can be uniquely determined from data, given a set of assumptions (e.g., a causal sufficiency) (Pearl, 2009). *Identifiability* refers to whether causal queries can be empirically estimated, and under what assumptions.

For additional background, please refer to Appendix A and Pearl (2009).

## 4. Problem Formulation

Causal inference aims to answer causal queries using data drawn from an unknown SCM. Let $\mathcal{M}^\star = (\mathbf{V}, \mathbf{U}, \boldsymbol{\mathcal{F}}, P(\mathbf{U}))$ denote the unknown ground truth SCM. A causal query $Q$, such as an average treatment effect, defined over $\mathcal{M}^\star$ has ground truth value $Q^\star = Q(\mathcal{M}^\star)$. As $\mathcal{M}^\star$ is unknown, causal estimators rely on a set of causal assumptions $\mathbf{H}$ (for instance, causal sufficiency) and available data $D$ drawn from $\mathcal{M}^\star$ to produce an estimate $\hat{Q}$ of the target quantity $Q^\star$. Definition 4.1 below formally describes the elements of a causal dataset.

> **Definition 4.1** (Causal dataset). A *causal dataset* is a tuple $\mathcal{D} = (Q, Q^\star, D, \mathcal{G}^\star, \mathbf{H}^\star)$ constructed from a known SCM $\mathcal{M}^\star = (\mathbf{V}, \mathbf{U}, \boldsymbol{\mathcal{F}}, P(\mathbf{U}))$ where:
>
> $Q$ is a causal query defined over endogenous variables $\mathbf{V}$,
>
> $Q^\star = Q(\mathcal{M}^\star)$ is the value of the query $Q$,
>
> $D = \{D_k \sim P_{\mathcal{M}^\star}(\mathbf{V} \mid \boldsymbol{do}(\mathbf{V}_k) = \mathbf{v}_k)\}_{k=1}^I$ is a collection of samples under $I$ interventional settings[1],
>
> $\mathcal{G}^\star$ is the causal graph associated with $\mathcal{M}^\star$, and
>
> $\mathbf{H}^\star$ is the set of assumptions satisfied by $\mathcal{M}^\star$.
>
> ---
> [1] The observational setting corresponds to samples drawn from $P_{\mathcal{M}^\star}(\mathbf{V})$.

In this work, we develop a generator of causal datasets

following Definition 4.1 such that, given an error metric $E(\hat{Q}, Q^\star)$, Causal ML methods can be evaluated both in the identification-consistent regime—where the assumed causal graph and assumptions used by the estimator, denoted $(\mathcal{G}, \mathbf{H})$, match the ground truth $(\mathcal{G}^\star, \mathbf{H}^\star)$—and under controlled misspecification. Here, $\mathcal{G}$ represents the graph provided to a method (e.g., a partial or misspecified graph for robustness testing), and $\mathbf{H}$ represents the assumptions that the method relies on during estimation. This setup enables systematic comparison of Causal ML methods both under ideal conditions, where identification holds, and under deviations from the ground truth that test robustness.

The interpretation of the error $E(\hat{Q}, Q^\star)$ depends on whether the assumptions required to identify $Q$ are satisfied. When they hold, the error measures finite-sample estimation accuracy for an identifiable causal target. When they fail, the same metric instead reflects robustness under assumption violation, including extrapolation and inductive bias. CausalProfiler supports both settings: users can enforce identification assumptions or deliberately violate them to study failure modes.

**Remark on causal discovery.** Causal datasets, as defined above, can also be used for evaluating causal discovery algorithms. Each dataset already includes the ground-truth causal graph $\mathcal{G}^\star$, allowing direct assessment of discovery methods. Thus, the query $Q$ can be left empty.

# 5. Sampling Causal Datasets with CausalProfiler

To generate causal datasets, CausalProfiler relies on a parametric specification of the sampling domain, called the *Space of Interest (SoI)*. Given an *SoI*, CausalProfiler samples an SCM (Section 5.2) and generates a corresponding causal dataset (Section 5.3). Appendices B to E contain the pseudocode for the sampling algorithms.

## 5.1. Defining a Space of Interest

The central abstraction of the proposed framework is the *Space of Interest (SoI)* (Definition 5.1), which provides a standardized way to specify synthetic causal datasets (Definition 4.1).

> **Definition 5.1** (Space of Interest). A *Space of Interest (SoI)* is a tuple $\mathcal{S} = (\mathbb{M}, \mathbb{Q}, \mathbb{D})$, where $\mathbb{M}$ is a class of SCMs, $\mathbb{Q}$ a class of causal queries, and $\mathbb{D}$ a class of data.

The mathematical definition of an *SoI* is intentionally open-ended: it specifies the classes of SCMs, queries, and data abstractly, without constraining how these components are parameterized. The concrete parameters exposed in the

current CausalProfiler implementation are one instantiation of this definition[2]. As the framework evolves, the parameter list will expand. To help readers understand the current implementation, we list the main parameter groups below; full descriptions, default values and examples can be found in Appendix B.

The parameters defining the class of SCMs $\mathbb{M}$ are those specifying (1) the *causal graph:* number of variables, expected edge density, proportion of hidden variables, optional predefined graph, (2) the *causal mechanisms:* mechanism family (linear, neural, tabular), discrete cardinalities, custom mechanism arguments, noise mode (e.g., additive), and (3) the *noises (exogenous variables):* noise distribution, distribution arguments (e.g., mean), and number of noise regions for discrete variables. The parameters defining the class of queries $\mathbb{Q}$ are query type (e.g., ATE), number of queries per SCM, specific queries, NaN-handling options, and kernel parameters for approximating conditioning in continuous SCMs (kernel type, bandwidth, custom kernels). The parameter defining the data class $\mathbb{D}$ is the number of samples generated.

## 5.2. Sampling Structural Causal Models

CausalProfiler first samples a directed acyclic graph over a set of endogenous variables, defining the SCM's causal structure. If specified in the *SoI*, CausalProfiler samples a subset of endogenous variables, $\mathbf{V}_H$, to be treated as unobserved and excluded from the observed dataset. To expose only the visible causal structure to the user, we apply Verma's latent projection algorithm (Verma, 1993) to the full causal graph, which produces an acyclic directed mixed graph.

Given the causal graph, CausalProfiler then assigns each endogenous variable a *mechanism* based on its parents and an exogenous noise distribution set by the *SoI*. It supports two types of mechanisms. *Discrete mechanisms*, also called regional discrete mechanisms (see Appendix D.1 for a formal definition), which support binary and categorical treatments, are defined tabularly by associating each element of a partition of the exogenous noise with distinct parents-to-child mappings. This enables controllable stochasticity and complexity, including highly non-linear and non-invertible behavior. The *SoI* also specifies how such mechanisms are sampled (e.g., with rejection-based sampling, see Appendix D.2). *Continuous mechanisms* are defined using parametric function families—such as neural networks or linear functions—with randomly initialized parameters.

---

[2]The current implementation of CausalProfiler supports only $\mathcal{L}_1$ data and ATE, CATE, and Ctf-TE queries.

## 5.3. Sampling Causal Datasets

**Data $D$.** Given an SCM $\mathcal{M}^\star$ sampled from the *SoI*, we generate an observational dataset $D$ by sampling i.i.d. data points from the entailed distribution of $\mathcal{M}^\star$ over observed variables. This involves forward-sampling from the structural equations in topological order, using the noise distributions specified for each variable and marginalizing out any latent variables.

**Query $Q$.** We first sample endogenous observable variables to serve as treatment variables, outcome variables, covariates, and factuals, depending on the query class of the *SoI*. By default, realizations are drawn from a large, separately sampled observational dataset, rather than from the theoretical variable domains. This ensures that queries are well-defined and correspond to realizable variable configurations under the SCM. To support different research goals, *SoIs* can be configured to relax this behavior to stress-test robustness. For causal discovery, query sampling can be disabled to generate datasets more efficiently given that they already include their ground-truth graph $\mathcal{G}^\star$.

**Query ground truth $Q^\star$.** Each query is estimated by drawing samples from the (manipulated) ground truth SCM: interventional queries via do-operations (action and prediction), and counterfactual queries via the three-step procedure (Pearl, 2009).

**Ground truth causal graph $\mathcal{G}^\star$.** As discussed in Section 5.2, $\mathcal{G}^\star$ is built as the latent projection of the ground-truth SCM's causal graph over the observed variables.

**Ground truth causal assumptions $\mathbf{H}^\star$.** Some assumptions are guaranteed directly by the *SoI* specification (e.g., variable types, cardinalities, presence of hidden variables). To characterize additional assumptions not fixed by the *SoI*, we provide an analysis module that can help quantify them (e.g., linearity via Pearson correlation or monotonicity). Appendix F provides the list of supported properties for analysis.

**Coverage guarantee.** Proposition 5.1 (proof provided in Appendix I) establishes that, with sufficiently expressive discrete mechanisms, CausalProfiler can assign strictly positive probability to any causal dataset within the specified *SoI*, guaranteeing $\mathcal{L}_3$-expressivity. Importantly, this proposition is not a statement about uniformity or absence of sampling bias: some SCMs may still have much lower probability of being sampled than others. Instead, the result guarantees that the sampling procedure is expressive enough to represent the full target space induced by the *SoI*. We include an analysis of the induced distribution of the sampled datasets in Appendix G.

---

> **Proposition 5.1** (Coverage)**.** For a Space of Interest $\mathcal{S} = (\mathbb{M}, \mathbb{Q}, \mathbb{D})$, whose class of structural causal models is a class of regional discrete SCMs[1] with the maximum number of noise regions, denoted $\mathbb{M}_{\text{RD-SCM}, r=R_{\max}}$, any causal dataset $\mathcal{D} = (Q, Q^\star, D, \mathcal{G}^\star, \mathbf{H}^\star)$ has a strictly positive probability to be generated.
>
> $$\forall \mathcal{S} = (\mathbb{M}, \mathbb{Q}, \mathbb{D}) \ s.t. \ \mathbb{M} \subseteq \mathbb{M}_{\text{RD-SCM}, r=R_{\max}},$$
>
> $$P(\mathcal{D}|\mathcal{S}) > 0$$
>
> ---
> [1]A formal definition can be found in Appendix D.1.

Taken together, these design choices reflect four properties that are essential for rigorous synthetic evaluation in Causal ML (Poinsot et al., 2025): Taken together, these design choices reflect four properties that are essential for rigorous synthetic evaluation in Causal ML (Poinsot et al., 2025): **transparency**, by making all assumptions explicit through the *SoI* parametrization, which serves as a declarative specification of the evaluation domain; **repeatability**, through randomized but seed-controlled sampling procedures that allow SCMs and queries to be reproduced across runs; **bias awareness**, supported by the coverage guarantee and empirical distribution analysis module; and **experimental control**, by exposing a wide range of configurable *SoI* parameters that enable causal datasets to be generated under specific assumptions and research objectives.

### 5.4. Summary of the entire sampling procedure

To sample a causal dataset $\mathcal{D} = (Q, Q^\star, D, \mathcal{G}^\star, \mathbf{H}^\star)$ as defined in Definition 4.1, CausalProfiler takes as input a *SoI* specifying distributions over SCMs, queries, and datasets (Section 5.1). CausalProfiler first samples a causal graph $\mathcal{G}^\star$ and causal mechanisms to build a SCM (Section 5.2). Then, from the SCM, CausalProfiler samples an observational dataset $D$, a query $Q$ and computes its ground-truth value $Q^\star$ (Section 5.3). The assumptions $\mathbf{H}^\star$ of the built SCM arise both from the user-specified *SoI* parameters (e.g., variable types or mechanism classes) and from the sampling procedure which randomly generates properties that can be identified through the assumption analysis module (Appendix F). Appendix H presents a visual overview of the sampling strategy.

## 6. Experiments

### 6.1. Verification of Benchmark Correctness

To validate the soundness of our benchmark generator, we perform consistency checks across the three levels of the PCH. Using the SCM sampler and query estimator of CausalProfiler, we test whether sampled SCMs satisfy the

Markov condition, do-calculus rules, and the structural counterfactual axioms (Pearl, 2009). We use discrete SCMs to allow exhaustive enumeration of conditioning sets for statistical tests. To ensure robustness, we iterate over a *SoI* parameter grid spanning the number of variables, edge density, cardinalities, and noise regions. See Appendix J for full details and results.

$\mathcal{L}_1$: **Markov property verification.** We test whether d-separations in the causal graph imply conditional independencies in the entailed observational distribution of the sampled SCMs. For each SCM, we enumerate d-separated triplets $(A, B, C)$ and test $A \perp B \mid C$ with Pearson's $\chi^2$ test (Pearson, 1900), filtering low-sample strata (Koehler & Larntz, 1980) and correcting for multiple tests (Benjamini & Hochberg, 1995). About 5% of tests fail, mostly due to finite-sample variability.

$\mathcal{L}_2$: **Do-calculus verification.** We test whether the three rules of do-calculus hold empirically. For each rule, we identify variable tuples satisfying its graphical preconditions. We then use the query estimator to generate two interventional datasets corresponding to the rule's left- and right-hand sides. We compare the resulting distributions with Pearson's $\chi^2$ test, filtering low-sample strata (Koehler & Larntz, 1980) and correcting for multiple tests (Benjamini & Hochberg, 1995). About 5.5% of tests fail, mostly due to finite-sample variability.

$\mathcal{L}_3$: **Structural counterfactual axiom verification.** We test whether the axioms of *composition*, *effectiveness*, and *reversibility* (Definition J.2 in Appendix J) hold for sampled SCMs. Because the axioms involve deterministic functional relationships, we count only exact matches of the query estimator. All axioms hold exactly across our samples, confirming the estimator's consistency with structural counterfactual semantics.

## 6.2. Comparison to existing benchmarks

We illustrate CausalProfiler's contribution to SCM diversity for evaluating Causal ML methods by comparing its SCMs (sampled over a *SoI* grid spanning number of variables, edge density, cardinalities, noise regions, and dataset size) with two existing benchmarks: the synthetic SCMs from the Causal Normalizing Flows (CausalNF) by Javaloy et al. (2023) and the CANCER and EARTHQUAKE models from R package bnlearn (Scutari, 2019). For interpretable visualization, we apply two-dimensional t-SNE (Maaten & Hinton, 2008) to the computable metrics of the analysis module (Appendix F), with a perplexity set to 30, as shown in Figure 1.

Figures 1a and 1b serve complementary purposes. Figure 1a shows that the 11 CausalNF SCMs occupy a very narrow region of the metric space, whereas sampling across an *SoI*

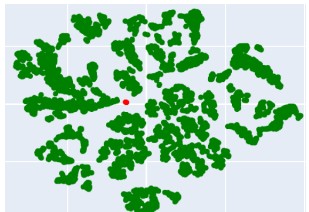 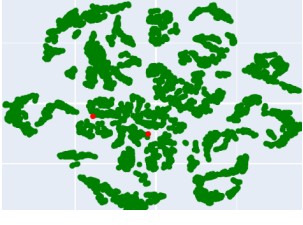

*(a)* CausalProfiler vs. CausalNF    *(b)* CausalProfiler vs. CANCER and EARTHQUAKE

*Figure 1.* Two-dimensional t-SNE plots of SCMs by CausalProfiler (shown in green) and by established benchmarks (shown in red), characterized by metrics from the analysis module.

with CausalProfiler yields much broader diversity. This illustrates the motivation for using a configurable generator rather than relying on a hand-crafted synthetic dataset. Figure 1b illustrates two properties of CausalProfiler: (i) its ability to reproduce datasets whose characteristics resemble well-known datasets (overlap in the embedding), and (ii) the additional diversity that emerges when sampling broadly across an SoI relative to a small set of fixed models. The two bnlearn networks (out of 24 available) were selected because their characteristics (e.g., number of nodes) match the SoI used in this visualization. Further details and results are in Appendices G.3 and G.4.

## 6.3. Method Evaluation using CausalProfiler

We next evaluate several recent causal inference methods across diverse *SoIs*. Our goal is not to exhaustively benchmark each method but to showcase the types of structured empirical investigations that CausalProfiler enables, especially on exploring robustness and violations of causal assumptions. Experiments that do not enforce the assumptions required for identification should be interpreted as robustness tests. In such settings, error may reflect extrapolation and estimator inductive biases in addition to finite-sample estimation error.

For each *SoI*, we evaluate every method using five random seeds, sampling 100 SCMs per seed. Each SCM yields one training set and five queries with ground-truth values, and results are aggregated across SCMs and seeds (fully described in Algorithm 12 in Appendix K). Experiments were run on a single Intel Core i9-14900K machine (24 cores, 32 threads, 96GB RAM), fully parallelized on CPU.

Performance is assessed by mean squared error between predicted and true query values. For each method and *SoI*, we report mean error, standard deviation, runtime, and failure rate (due to numerical issues or exceptions). In addition, box plots visualize the full error distribution through the median, interquartile range (IQR), and whiskers at $1.5 \times$ IQR.

We compare CausalNF (Javaloy et al., 2023), Neural Causal

*Table 1.* Performance summary of CausalNF, DCM, NCM, and VACA on the ATE experiments.

| Space | Method | Mean Error | Std Error | Max Error | Runtime (s) | Fail Rate (%) |
|---|---|---|---|---|---|---|
| Linear-Medium | DCM | **0.1530** | 1.5289 | 33.9766 | 16541.2 | 0.00 |
| | NCM | 0.4618 | 0.9001 | 9.6134 | 7384.7 | 0.00 |
| | VACA | 0.4209 | 0.6195 | 2.3807 | 2734.5 | 53.40 |
| | CausalNF | 0.4625 | **0.8985** | 9.6079 | 13790.4 | 0.00 |
| NN-Medium | DCM | 0.0276 | 0.0114 | 0.0746 | 15894.4 | 0.00 |
| | NCM | 0.0111 | 0.0121 | 0.1484 | 7322.8 | 0.00 |
| | VACA | **0.0090** | **0.0077** | 0.0479 | 5759.6 | 5.00 |
| | CausalNF | 0.0160 | 0.0107 | 0.1209 | 10732.7 | 0.00 |
| NN-Large | DCM | 0.0267 | 0.0100 | 0.0739 | 19166.2 | 0.00 |
| | NCM | 0.0101 | 0.0103 | 0.1161 | 9450.6 | 0.00 |
| | VACA | **0.0090** | **0.0094** | 0.0535 | 5690.8 | 11.60 |
| | CausalNF | 0.0159 | 0.0105 | 0.1535 | 15114.8 | 0.00 |
| NN-Large-LowData | DCM | 0.0777 | 0.0445 | 0.3701 | 2412.1 | 0.00 |
| | NCM | **0.0097** | **0.0107** | 0.1263 | 404.7 | 0.00 |
| | VACA | 0.0103 | 0.0134 | 0.1043 | 5217.4 | 0.00 |
| | CausalNF | 0.0359 | 0.0146 | 0.1712 | 22138.2 | 0.00 |

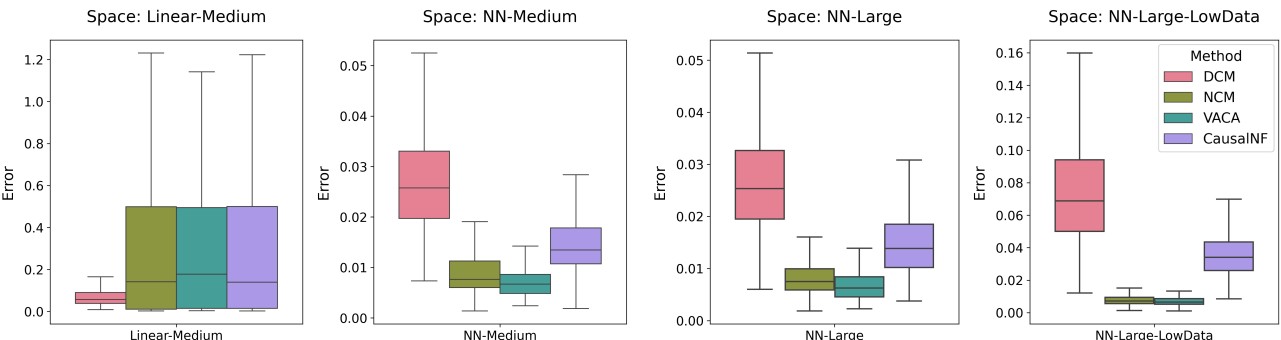

*Figure 2.* Box plots showing ATE estimation errors across different *SoIs*.

Models (NCM) (Xia et al., 2023), Variational Causal Graph Autoencoder (VACA) (Sánchez-Martin et al., 2022), and Diffusion-based Causal Models (DCM) (Chao et al., 2024).

Additional experiments (including causal discovery), extended results, and *SoI* configurations are provided in Appendix K.

### 6.3.1. EXPERIMENT 1: GENERAL EVALUATION ACROSS DIVERSE SCMS

To showcase CausalProfiler's flexibility, we evaluate Average Treatment Effect (ATE) estimates of VACA, CausalNF, DCM, and NCM on continuous-variable SCMs across four *SoIs*: (1) Linear-Medium, using linear SCMs (15-20 nodes, 1000 samples), (2) NN-Medium, using neural SCMs with a 2-layer ReLU network (8 hidden units, 15-20 nodes, 1000 samples), (3) NN-Large, using larger neural SCMs (20-25 nodes, 1000 samples), and (4) NN-Large-LowData, identical to NN-Large but with 50 samples. Results are presented

in Table 1 and Figure 2.

We first examine the results in the Linear-Medium setting in comparison to NN-Medium. In the Linear-Medium setting, DCM achieves the lowest average error (0.1530), indicating strong overall performance. However, a few occasional large failures (maximum error was 33.98) make its standard deviation notably high (1.5289) despite the narrow interquartile range visible in Figure 2. These results suggest that DCM is effective for most queries but can produce large errors in rare cases—potentially problematic in safety-critical applications matching this *SoI*. VACA performs competitively with lower maximum error and faster runtime, but suffers a high failure rate (53.4%) due to occasional numerical instabilities (invalid floating-point values). In the NN-Medium setting, where the causal mechanisms are small neural networks, DCM's advantage disappears, and VACA emerges as the best performer, with the lowest error mean (0.0090) and standard deviation (0.0077), with failure rate reduced to 5%. We observe that DCM becomes the weakest per-

former in this setting, revealing that method rankings are highly sensitive to the underlying functional form of the mechanisms. This underscores the need for practitioners to evaluate methods within the *SoI* most relevant to their application. Lastly, NN SCMs yield lower errors than linear ones. A plausible explanation is an inductive-bias match with the evaluated neural methods and the tendency of small randomly initialized NNs to produce relatively smooth, low-frequency functions that are easier to estimate from finite data (Rahaman et al., 2019).

We next investigate the effect of reducing data availability by comparing NN-Large (1000 samples) to NN-Large-LowData (50 samples). DCM is strongly affected: its error nearly triples (from 0.0267 to 0.0777) and its IQR expands noticeably. CausalNF also shows sensitivity to low-data regimes. In contrast, both VACA and NCM maintain stable performance, with nearly unchanged mean and standard deviation. Notably, VACA achieves a 0% failure rate and shows unexpectedly strong robustness under limited data.

While not intended as a comprehensive benchmark, these comparisons illustrate the types of insights enabled by our framework. Across the selected *SoIs*, DCM performs well on average but can produce large outlier errors or become less stable in low-data settings. Conversely, VACA shows promising generalization even with limited data, though it occasionally fails on certain SCMs. These findings are specific to the explored *SoIs* and should not be taken as general conclusions. Rather, they show how our framework enables structured, *SoI*-specific evaluations, helping practitioners assess which methods may be more suitable for their own modeling context.

### 6.3.2. EXPERIMENT 2: COUNTERFACTUAL ESTIMATION ON DISCRETE SCMS

This experiment evaluates robustness in counterfactual estimation on discrete-variable SCMs. We test CausalNF and DCM, which were originally designed for continuous settings, motivated by prior work showing that CausalNF can approximate discrete distributions (Javaloy et al., 2023; de Vassimon Manela et al., 2024).

We consider three discrete *SoIs*: (1) D2-Reject, with 10-15 nodes, binary variables, and rejection-based mechanism sampling; (2) D4-Unbias, with the same graph size but 4-category variables and unbiased random mechanism sampling; and (3) D2L-Unbias, with larger graphs (20-30 nodes), binary variables, and unbiased random mechanism sampling. The results are shown in Table 2.

DCM has lower mean error in all three discrete settings. The difference is substantial in the two binary-variable spaces: on D2-REJECT, DCM achieves a mean error of 0.1974 compared to 0.5233 for CausalNF, and on D2L-UNBIAS,

0.1895 compared to 0.4340. In contrast, the results on D4-UNBIAS do not indicate a clear winner: DCM obtains a moderately lower mean error (0.4087 versus 0.4783), while CausalNF has a lower standard deviation (0.3149 versus 0.3418) and a lower maximum error (1.0000 versus 2.0000).

DCM's lower mean error comes with a substantial runtime cost. CausalNF is approximately $16\times$, $15\times$, and $21\times$ faster than DCM on D2-REJECT, D4-UNBIAS, and D2L-UNBIAS, respectively. Moreover, in the two unbiased-sampling regimes, CausalNF avoids the largest errors observed for DCM: its maximum error is 1.0000 in both settings, compared with 2.0000 and 1.5455 for DCM.

Overall, the experiments show a tradeoff in the considered settings between either lower average error or faster runtime and less severe worst-case errors. These results demonstrate the utility of our framework in systematically evaluating methods. This evaluation is not meant as a definitive comparison, only as a demonstration of how method tradeoffs can be explored in a structured way using CausalProfiler.

## 7. Limitations and Future Work

Open-source research frameworks such as CausalProfiler are never fully finished. They evolve continuously through community contributions as the field advances.

**Diversify Spaces of Interest.** Several directions remain open for extending the supported *SoIs* in CausalProfiler, such as scaled and mixed-variable SCMs, sampling interventional training data, more realistic data-generating scenarios (e.g., selection bias or measurement noise), and extensions beyond tabular data to time-series or text. Another important direction is to automate the exploration of *SoIs*, for example, searching for assumption regimes that reveal a method's failure modes, to reduce reliance on manual specification.

**Causal Datasets Distribution.** While the coverage proposition (Proposition 5.1) guarantees that any causal dataset has a positive probability of being sampled within a given *SoI* with sufficiently expressive discrete mechanisms, it does not characterize the distribution of generated datasets. As discussed in Appendix G, certain classes of SCMs remain unlikely to be sampled unless explicitly specified in the *SoI* (e.g., linear SCMs). Hence, aggregated results should be interpreted with the understanding that sampled causal datasets are not uniformly distributed. We recommend using the analysis module (Appendix F) to identify underrepresented attributes, as they vary between *SoIs*.

Reducing distributional bias is an important future research direction. Achieving a perfectly balanced distribution over all metrics is inherently impossible. For instance, uniform sampling over discrete mechanism functions biases toward non-bijective ones, since bijections are not dense in

*Table 2.* Performance summary of CausalNF and DCM on the discrete experiments. Bolded are numbers of interest.

| Space | Method | Mean Error | Std. Error | Max Error | Runtime (s) | Fail Rate |
|---|---|---|---|---|---|---|
| D2-Reject | DCM | **0.1974** | **0.2450** | 1.0000 | 6.2790 | 0% |
| | CausalNF | 0.5233 | 0.3286 | 1.0000 | **0.3923** | 0% |
| D4-Unbias | DCM | **0.4087** | 0.3418 | 2.0000 | 6.4504 | 0% |
| | CausalNF | 0.4783 | **0.3149** | 1.0000 | **0.4401** | 0% |
| D2L-Unbias | DCM | **0.1895** | **0.2481** | 1.5455 | 11.0358 | 0% |
| | CausalNF | 0.4340 | 0.3277 | 1.0000 | **0.5351** | 0% |

the function space. Future work may enable finer control over dataset distributions and underrepresented attributes, depending on the guarantees one wishes to enforce. A promising avenue is stratified sampling, which would provide weighted coverage of selected attributes. Currently, controllable *SoI* parameters (e.g., number of nodes) are sampled uniformly, but emergent attributes follow skewed distributions induced by generation. For controllable *SoI* parameters, stratification could be achieved via weighted sampling over groups of *SoIs*. For emergent properties, approximate stratification may require rejection sampling or new sampling algorithms that enforce global constraints during generation. Such improvements could provide finer control over assumption distributions, enabling evaluations that better target the assumptions and regimes relevant to specific real-world domains.

**Bridging the simulation-to-real gap.** While synthetic evaluation is indispensable (Poinsot et al., 2025), it is insufficient to fully assess method capabilities, as results may not transfer to real-world settings. In CausalProfiler, alignment with real domains currently relies on manually specified *SoIs*, guided by domain expertise or empirical features. A key direction for future work is to develop methods that automatically map real data to sets of *SoIs*, enabling principled semi-synthetic evaluation pipelines where *SoIs* are shaped by empirical evidence rather than fixed assumptions. However, mapping from observational data to *SoIs* is a fundamentally underconstrained problem, and any such inference must be handled with care, given the challenges around identifiability and inductive bias.

## 8. Conclusion

This work introduces CausalProfiler, a synthetic causal dataset generator for evaluating Causal ML methods across the three levels of the Pearl Causal Hierarchy. At its core is the notion of a *Space of Interest*, which replaces the ad hoc choice of fixed evaluation datasets with a principled specification of the entire evaluation scope, i.e., classes of causal models, queries and data. This shift enables transparent, repeatable, and assumption-aware assessments under diverse causal conditions. After demonstrating that the causal datasets generated by CausalProfiler are causally consistent

and can be similar to existing benchmarks while also being more diverse, we show that the performance of state-of-the-art Causal ML methods varies substantially across different *Spaces of Interest*, underscoring the importance of rigorous, distribution-level evaluation. CausalProfiler is not intended to replace real-data studies or targeted evaluations, but to complement them. By enabling systematic exploration, it helps uncover failure modes, expose robustness to violated assumptions, and highlight unexpected strengths that may motivate new research directions. In this way, CausalProfiler marks a first step toward a more complete evaluation ecosystem for Causal ML.

## Impact Statement

This paper presents work whose goal is to advance the field of Machine Learning. There are many potential societal consequences of our work, none which we feel must be specifically highlighted here.

## Acknowledgements

This research was supported by the UKRI Centre for Doctoral Training in Accountable, Responsible and Transparent AI (ART-AI) [EP/S023437/1], by the Engineering and Physical Sciences Research Council (EPSRC) [EP/X025470/1], the French National Research Agency (ANR) under the France 2030 program, under the reference 23-PEIA-004, and by the Artificial Intelligence for Safe and Smart Mobility Chair (Grant No. ANR-23-CPJ1-0099-01). We thank Adrián Javaloy for helpful feedback.

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

# A. Additional Definitions & Notation

**Definition A.1** (Semi-Markovian and Markovian SCMs)**.** An SCM is said to be *semi-Markovian* (Pearl, 2009) if its set of structural equations is acyclic, meaning there exists an ordering of the equations such that for any two functions $f_i, f_j \in \mathcal{F}$, if $f_i < f_j$, then $V_j \notin \mathbf{PA}(V_i)$. This condition ensures that the causal dependencies among endogenous variables form a directed acyclic graph. An SCM is *Markovian* (Pearl, 2009) if the exogenous variables influencing different endogenous variables are mutually independent. Formally, for all distinct $V_i, V_j \in \mathbf{V}$, we have $\mathbf{U}_{V_i} \perp\!\!\!\perp \mathbf{U}_{V_j}$. This implies the absence of latent confounding, allowing the model to be fully described by a DAG with independent noise terms.

**Definition A.2** (Causal Graph of a Semi-Markovian SCM)**.** The causal graph of a Semi-Markovian (Bareinboim et al., 2022) SCM is an acyclic directed mixed graph with:

- Directed edge $V_i \to V_j$ if $V_i \in \mathbf{PA}(V_j)$

- Bi-directed edge $V_i \leftrightarrow V_j$ if $\mathbf{U}_{V_i} \not\perp\!\!\!\perp \mathbf{U}_{V_j}$

## A.1. Interventional Quantities ($\mathcal{L}_2$)

**Average Treatment Effect (ATE):**

$$\text{ATE}_{T \to Y} = \mathbb{E}[Y|\boldsymbol{do}(T=1)] - \mathbb{E}[Y|\boldsymbol{do}(T=0)]$$

**Conditional Average Treatment Effect (CATE):**

$$\text{CATE}_{T \to Y}(\mathbf{x}) = \mathbb{E}[Y|\boldsymbol{do}(T=1), \mathbf{X} = \mathbf{x}] - \mathbb{E}[Y|\boldsymbol{do}(T=0), \mathbf{X} = \mathbf{x}]$$

## A.2. Counterfactual Quantities ($\mathcal{L}_3$)

A counterfactual query such as $P(Y_{\boldsymbol{do}(T=t)}|\mathbf{V}_F = \mathbf{v}_F)$ is computed by abduction (conditioning on factual data), action (intervening), and prediction (computing the outcome) (Pearl, 2009).

**Counterfactual Total Effect (Ctf-TE):**

$$\text{Ctf-TE}_{T \to Y}(y, t, c, \mathbf{v}_F) = P(y_{\boldsymbol{do}(T=t)}|\mathbf{V}_F = \mathbf{v}_F) - P(y_{\boldsymbol{do}(T=c)}|\mathbf{V}_F = \mathbf{v}_F)$$

Originally, (Zhang & Bareinboim, 2018) defined counterfactual direct, indirect, and spurious effects by conditioning on the factual realization of one variable. Later, (Plečko & Bareinboim, 2024) generalized this to allow the factual evidence to be any subset $\mathbf{V}_F$ of endogenous variables, enabling more granular and flexible counterfactual analyses.

# B. Space of Interest

## B.1. Configurable Parameters of a Space of Interest

Each Space of Interest is defined by a set of parameters that control the *SCM space*, the causal queries of interest (*Query space*), and the dataset used for estimation (*Data space*). Table 3 provides an overview of all configurable parameters in a Space of Interest instance, along with their default values. Some parameters are only relevant under specific conditions—for instance, kernel parameters are used only with continuous variables (e.g., when evaluating conditional expectations), function sampling strategies apply exclusively to discrete mechanisms, noise regions apply only for discrete SCMs, and noise mode is ignored for tabular mechanisms (noise is already embedded in the table). Note that one can use symbolic expressions involving N (the number of nodes) and V (the cardinality of a variable) to define parameters that depend on sampled values. For example, the expected number of edges can be set as 0.5 * N, or the number of noise regions in a discrete SCM can be set to V.

*Table 3.* Parameters defining a Space of Interest instance and their default values. The double lines in the table conceptually separate the SCM space, Query space, and Data space.

| Category | Parameter | Default Value |
|---|---|---|
| SCM structure | Number of endogenous variables | [5, 15] |
| | Variable dimensionality | [1, 1] |
| | Expected number of edges (required) | — |
| | Proportion of hidden variables | 0.0 |
| | Predefined causal graph | — |
| Mechanisms | Mechanism family (e.g., Linear, NN, Tabular) | Linear |
| | Mechanism arguments (used to define custom NN/tabular mechanisms) | — |
| | Endogenous variable cardinality (for discrete variables only) | 2 |
| | Variable type | Continuous |
| | Discrete function sampling (for discrete variables only) | Sample Rejection |
| | Noise mode | Additive |
| Noise | Noise distribution | Uniform |
| | Noise distribution arguments | [-1, 1] |
| | Number of noise regions (for discrete variables only) | N |
| Query | Number of queries per sample | 1 |
| | Query type | ATE |
| | Specific query (overrides random query sampling) | — |
| | Whether to allow queries that evaluate to NaN | False |
| | Whether to disable query sampling (e.g., for causal discovery) | False |
| Kernel | Kernel type | Gaussian |
| | Kernel bandwidth | 0.1 |
| | Custom kernel function | — |
| Data | Number of samples in the set of observed data | 1000 |

### PARAMETER DESCRIPTIONS AND TYPICAL VALUES

We briefly summarize the role of each parameter and the typical values it can take. Unless otherwise stated, scalar parameters may be given as fixed values, ranges (e.g., (a, b)), or simple expressions in N (number of nodes) and V (variable cardinality).

**SCM structure.**

- **Number of endogenous variables.** Range for the number of nodes in each sampled graph (e.g., [5, 15]). A value is drawn from this range for each SCM.

- **Variable dimensionality.** Range for the dimensionality of each variable (typically [1, 1] in our experiments, but higher-dimensional variables are supported).

- **Expected number of edges (required).** Controls graph density via the expected total number of edges. Can be a fixed integer, a range, or an expression such as 0.5 * N or log(N).

- **Proportion of hidden variables.** Fraction of endogenous variables that are hidden in the returned graph, data, and queries (a float in $[0, 1]$); $0.0$ means no hidden variables.

- **Predefined causal graph.** Fixed graph to be used for all SCMs. If unset, graphs are sampled according to the structural parameters above.

**Mechanisms.**

- **Mechanism family.** Choice of functional form for the structural mechanisms (e.g., linear, neural network, tabular), given by an enum.

- **Mechanism arguments.** Optional hyperparameters passed to the chosen mechanism family (e.g., hidden-layer sizes for neural networks, or explicit tables for tabular mechanisms).

- **Endogenous variable cardinality.** Cardinality (or range of cardinalities) for discrete variables (e.g., 2 or `(2, 4)`). Ignored when `variable type` is continuous.

- **Variable type.** Whether variables are continuous or discrete. This determines which mechanism and noise options are applicable.

- **Discrete function sampling.** Strategy for sampling discrete mechanisms (e.g., sample-rejection, enumeration, or random sampling). More information about these strategies in Appendix D.2.

- **Noise mode.** How noise enters the structural equations (e.g., additive or multiplicative). This is ignored for tabular mechanisms, where stochasticity is already encoded in the table.

**Noise.**

- **Noise distribution.** Distribution from which exogenous noise variables are drawn (e.g., uniform).

- **Noise distribution arguments.** Parameters of the noise distribution (e.g., `[-1, 1]` for a uniform distribution on $[-1, 1]$).

- **Number of noise regions.** Used to specify the number of noise regions in mechanisms. The more the number of noise regions, the more random / stochastic the mechanism is. Setting to 1 yields deterministic mechanisms.

**Query space.**

- **Number of queries per sample.** Number of causal queries generated for each SCM.

- **Query type.** Type of causal query to sample (e.g., ATE, CATE, or Ctf-TE), specified via an enum.

- **Specific query.** Optional string specifying a fixed query to evaluate. If provided, this overrides random query sampling.

- **Allow NaN queries.** Whether to include queries whose numerical estimates evaluate to NaN (e.g., due to lack of support). By default, such queries are excluded.

- **Disable query sampling.** If set to `True`, no queries are sampled or evaluated (useful for causal discovery tasks where only data and graphs are needed).

**Kernel weighting (continuous conditioning only).**

- **Kernel type.** Choice of kernel used to approximate conditioning for continuous variables (e.g., Gaussian, epsilon).

- **Kernel bandwidth.** Bandwidth parameter controlling the smoothness of the kernel weighting (and acting as an epsilon threshold when using an epsilon kernel).

- **Custom kernel function.** Optional user-specified (in Python) kernel function.

**Data space.**

- **Number of samples in the set of observed data.** Size of the dataset generated for each SCM.

## B.2. Guidelines for defining a Space of Interest

This section presents general guidelines on how researchers and practitioners could define the Spaces of interest depending on the analysis they want to carry out.

**Testing a new method without a predefined application.** Begin by evaluating the method in settings where its assumptions hold. If an assumption can be enforced directly through SoI parameters (e.g., no hidden variables, linear mechanisms), fix those parameters accordingly. Otherwise, sample from a broader SoI and use the assumption-analysis module to retain only SCMs satisfying the assumption.

Next, assess robustness by gradually introducing assumption violations. Assumptions that can be varied explicitly (e.g., increasing the proportion of hidden variables) should be adjusted directly through the SoI. For assumptions that cannot be controlled parametrically, sample broadly and filter using the assumption-analysis module. The module can also quantify the *degree* of violation, enabling sensitivity analyses.

**Comparing multiple methods without a predefined application.** Follow the same two-stage structure. First evaluate all methods in SoIs where their assumptions are jointly satisfied (verification). Then introduce controlled assumption violations to study comparative robustness. This yields a principled, assumption-aware comparison rather than a collection of isolated tests.[3]

**Evaluating methods for a specific application or use case.** Fix all SoI parameters that are known from domain expertise (e.g., variable types, expected graph sparsity, presence of latent confounding). Then vary the remaining uncertain parameters to span the plausible causal conditions for the application. This produces a well-defined set of SCMs consistent with the use case, enabling structured, domain-grounded evaluation. We now illustrate this with a concrete example.

### B.2.1. EXAMPLE 1: PRICE ELASTICITY

A company's analytical marketing team wishes to estimate the price elasticity of one of its products. The team has access to three years of sales data and price history, as well as competitors' prices and inflation trends. The team knows that calculating price elasticity involves determining the ATE of price on sales for various price values. In addition, the team has also constructed a causal graph corresponding to the decision-making process used to set the product price and its effect on sales, as shown in Figure 3. In fact, the price is set based on competitors' prices, inflation (because production costs are highly correlated with it), and a set of other factors for which they do not have historical data to include in the modeling. These factors are also assumed to be used by competitors.

The team wants to calculate discrete elasticity using non-parametric DoubleML methods (Chernozhukov et al., 2018). However, they do not know which method is more suitable for their setting. Hence, the team decides to use CausalProfiler to perform their own comparison and define the following set of SoI parameters:

- **SCM structure parameters**: The team decides to use the option of using a predefined causal graph corresponding to the one in Figure 3.

- **Mechanisms parameters**:
    - Variable type: Continuous, as all the variables in this use case are continuous.
    - Mechanism family: Neural Networks, as no assumption is made about the functional form of the causal mechanisms.
    - Given the previously made choices, the other parameters have no influence on the generation.

- **Noise parameters**:
    - Noise distribution: all the available noise distributions are considered, as no assumption is taken about the form of the distribution.

---

[3]Our experiments in Sections 6.3.1 and 6.3.2 illustrate the types of analyses enabled by CausalProfiler but are not intended as full comparative evaluations.

- Noise distribution arguments are the default ones, as neither the mean nor the scale of the noise should drastically affect the generation, as we are using randomly initialized Neural Networks as causal mechanisms.

- **Query parameters**:

  - The team decides to define a set of specific queries rather than randomly sampling them, as they are interested in a single pair of treatment and outcome variables.

- **Kernel parameters**: Default Kernel parameters.

- **Data parameter**: The number of samples is varied between the number of observations they have for the past year and for the three past years. Indeed, the team would ideally measure the price elasticity over the past year to have the most recent measurement, but is also ready to include older data (maximum three years old) to provide more observations to the model if it drastically changes its accuracy.

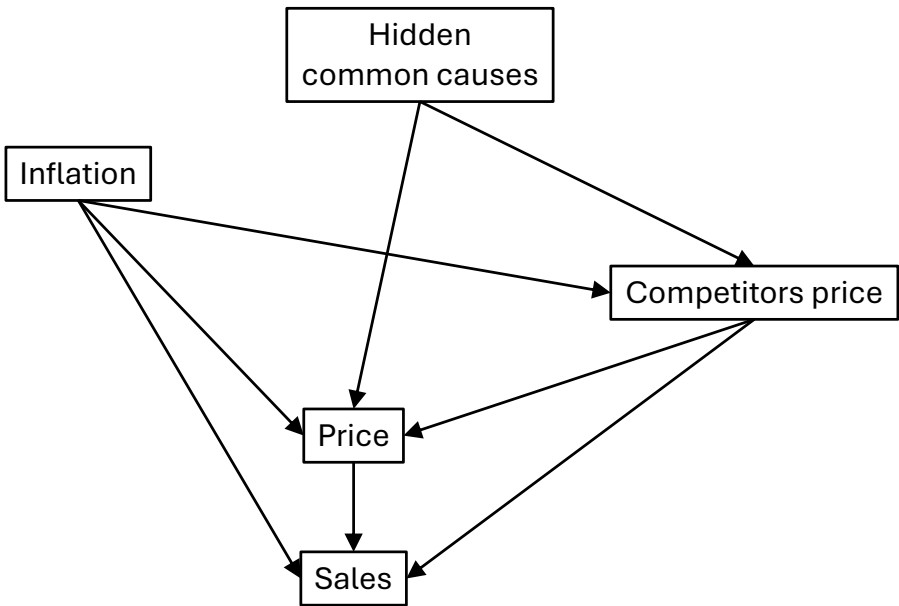

*Figure 3.* Causal graph of the price elasticity example.

### B.2.2. EXAMPLE 2: CLIENT RETENTION

A luxury company's analytical marketing team wants to launch a client retention analysis program to identify which actions and client characteristics have the greatest potential to reduce churn. In particular, they aim to conduct this analysis across their various products, including bags, jewelry, fragrance, and other items.

The team knows that churn analysis is fundamentally a counterfactual question. Hence, they are interested in comparing some Causal ML methods able to estimate counterfactual effects. However, instead of directly using methods to generate semi-synthetic data to perform model selection for each churn analysis per product, they first prefer to perform a more general synthetic analysis for the following two reasons:

- First, some clients' data is subject to the GDPR regulation, requiring the data to be anonymized before being analyzed and used to train any machine learning model. However, it is not yet clear to the team how the data must be processed to be GDPR compliant. Hence, to avoid wasting time, they decide to begin method comparison using synthetic data while figuring out how the data can be anonymized for semi-synthetic data generation.

- Second, the company would like to reduce the risk associated with dependencies on external libraries. Indeed, giving the team the opportunity to choose whatever method they want for each churn analysis may imply a large number of different methods, resulting in an increasing amount of work for monitoring updates of external libraries, debugging,

and verifying their compliance for release to production after the proof-of-concept phase. Hence, the team is interested in finding a set of up to 5 methods that would enable them to achieve good accuracy across all the churn analyses they want to perform. Identifying such a set would enable the team to reduce the number of methods to study to perform the actual model selection with semi-synthetic data.

Hence, the team decides to use CausalProfiler to perform the above-presented comparison and define the following set of SoI parameters:

- **SCM structure parameters**: As the purchase behaviors over the products might be quite different (e.g., frequency of purchase of makeup is much higher than bags), not the same variables are measured for all the product retention studies. Hence, the parameters are varied over the different settings of the products' retention datasets.

  - Number of endogenous variables: between 10 and 20
  - Variable dimensionality: 1
  - Proportion of hidden variables: between 10 and 30 percent, because the team knows there are some phenomena they are not able to measure.
  - Expected number of edges: $0.7 \frac{(1+p)N[(1+p)N-1]}{2}$ with $N$ being the number of edges and $p$ the proportion of hidden variables. Such an expression means that the generated DAG contains 70% of the edges of the corresponding fully connected DAG. Indeed, the generated graphs must be dense, as all the variables in the retention datasets are highly correlated.

- **Mechanisms parameters**:

  - Variable type: Discrete. Indeed, all the variables that the team gathered are discrete, such as the indicator whether the client is a churner or not, the number of products bought in the last three months, the number of emails the client received in the last three months, the age group of the client, etc
  - Mechanism family: Tabular (corresponding to regional discrete mechanisms)
  - Endogenous variable cardinality: between 3 and 8, because even if the team has detailed information with a higher cardinality of variables, they are willing to keep the analysis understandable by a human. Hence, they estimate that a discrete variable with more than eight possible values becomes too complex for strategic marketing optimization. In addition, the team is open to getting less granular results (by grouping some values in a single category, eg, age groups of 20-30 and 30-40 grouped into 20-40) if the accuracy of the methods drastically increases with a lower cardinality of variables.
  - Discrete function sampling: Sampling rejection to keep sampling quite fast wile still keeping control over the level of stochasticity of the causal mechanisms through the number of noise regions parameters
  - Noise mode: It has no impact on the discrete data generation.

- **Noise parameters**: Except for the number of noise regions, the noise parameters have no influence on the generation, as we are using discrete mechanisms, so the default noise parameters are kept.

  - Number of noise regions: between 5 and 50, as some causal mechanisms (representing purchase behavior of some products) might be more deterministic than for some other products. For instance, products with a low purchase frequency are hence much more impacted by many events that might occur between two purchases.

- **Query parameters**: As the team is interested in the counterfactual effect of all the variables they measure on retention, they decide to use the random query sampling option.

  - Number of queries per sample: 100 to be able to perform some accuracy stability analyses within the same SCM
  - Query type: Ctf-TE
  - Whether to allow queries that evaluate to NaN: True, because the team wants to see if the methods are able to flag whenever a query is impossible to answer instead of trying to return a result anyway
  - Whether to disable query sampling: False

- **Kernel parameters**: Not needed as the data are discrete

- **Data parameter**: The number of samples is varied between the minimum and maximum number of client retention data they have over the different products on which they want to perform the retention analyses.

## C. Causal Graph Sampling

We first generate a random Directed Acyclic Graph (DAG) that specifies causal relations between variables. This structure is then extended by designating a subset of variables as hidden/unobserved, enabling the creation of both Markovian and semi-Markovian SCMs depending on the *SoI* spec. We separate these two steps in separate algorithms for clarity. First, Algorithm 1 samples a DAG over a unique type of variables, not yet distinguishing between observable and unobservable variables. To do so, the list of nodes is defined as a list of integers imposed to be the topological order of the DAG (line 1). Then, for each node (line 4), its number of parents is sampled from a Binomial law of parameters $i - 1$ and $p_{edge}$ with $i$ the rank of the node in the topological order (line 5). The actual parents are sampled from the set of nodes having a smaller topological rank (line 6) which guarantees that the generated graph is a DAG. Second, from the generated DAG, Algorithm 2 simply creates the two sets of observables and unobservable variables by sampling $p_h.|\mathbf{V}|$ unobservable node among the total set of nodes (line 3).

---

**Algorithm 1** Generate a Random DAG with Expected Degree

**Inputs:** number of nodes $N$, expected degree $d$

1: $V \leftarrow \{1, \dots, N\}$
2: $E \leftarrow \{\}$
3: $p_{edge} \leftarrow \frac{2d}{N-1}$
4: **for** $i \in [1, N]$ **do**
5:     $N_{\mathbf{PA}(i)} \sim B(i - 1, p_{edge})$
6:     $\mathbf{PA}(i) \leftarrow N_{\mathbf{PA}(i)}$ nodes sampled without replacement from $V$
7:     $E \leftarrow E \cup \{j \rightarrow i \mid j \in \mathbf{PA}(i)\}$
8: **end for**

**Output:** $\mathcal{G} = \{V, E\}$

---

**Algorithm 2** Generate a DAG with Observed and Hidden Variables

**Inputs:** number of nodes $N$, expected degree $d$, proportion of hidden variables $p_h$

1: $\mathcal{G} = (V, E) \leftarrow DAG\_sampling(N, d)$ *(see Algorithm 1)*
2: $N_h \sim B(N, p_h)$
3: $V_h \leftarrow N_h$ nodes sampled without replacement from $V$
4: $V_o \leftarrow V \backslash V_h$

**Output:** $\mathcal{G} = \{V = V_o V_h, E\}$

---

As some variables in the DAG are unobserved, we expose only the observed structure to the user in the form of an acyclic directed mixed graph. To obtain this, we apply Verma's latent projection algorithm to the causal graph of each sampled regional discrete SCM (see Algorithm 3). If a method requires the true SCM, including the hidden confounders, that can be accessed as well.

---

**Algorithm 3** Projection Algorithm (Verma, 1993)

**Input:** an acyclic directed mixed graph $\mathcal{G} = \{\mathbf{V_O}, \mathbf{V_H}, \mathbf{E}\}$, with $\mathbf{V_O}$ the set of observed variables, $\mathbf{V_H}$ the set of hidden variables and $\mathbf{E}$ the mixed edges

1: $\mathbf{E}' \leftarrow \{\}$
2: **for** $A, B \in \mathbf{V_O}$ **do**
3:     **if** there is a directed path $A \rightarrow \dots \rightarrow B$ in $\mathcal{G}$ with all intermediate nodes belonging to $\mathbf{V_H}$ **then**
4:         $\mathbf{E}' \leftarrow \mathbf{E}' \cup \{A \rightarrow B\}$
5:     **end if**
6:     **if** there is a collider-free path $A \leftarrow \dots \rightarrow B$ in $\mathcal{G}$ with all intermediate nodes belonging to $\mathbf{V_H}$ **then**
7:         $\mathbf{E}' \leftarrow \mathbf{E}' \cup \{A \leftrightarrow B\}$
8:     **end if**
9: **end for**
10: $\mathbf{G}' \leftarrow \{\mathbf{V_O}, \mathbf{E}'\}$

**Output:** $\mathbf{G}'$ the latent projection of $\mathbf{G}$ over $\mathbf{V_O}$

---

# D. Sampling Discrete SCMs

## D.1. Regional Discrete SCMs

Regarding discrete SCMs, we sample discrete Markovian SCMs which we refer to as **Regional discrete SCMs** as presented in Definition D.1 below.

---

**Definition D.1. Regional discrete SCM**

A **regional discrete SCM** is a markovian SCM $\mathcal{M} \coloneqq (\mathbf{V}, \mathbf{U}, \mathcal{F}, P(\mathbf{U}))$ where:

- $\mathbf{V} = \{V_1, ..., V_d\}$ the set of finite discrete endogenous variables is divided into two sets $\mathbf{V}_o$ and $\mathbf{V}_h$ respectively representing the set of observed and hidden variables such that $\mathbf{V} = \mathbf{V}_o \cup \mathbf{V}_h$ and $\mathbf{V}_o \cap \mathbf{V}_h = \emptyset$

- $\mathbf{U} = \{U_1, ..., U_d\}$ the set of mutually independent continuous exogenous variables is such that $\forall i \in [1, d]$, $U_{V_i} = U_i$

- $\mathcal{F}$ the structural equations are regional discrete mechanisms as defined in Definition D.2

The class of regional discrete SCMs is denoted $\mathbb{M}_{\mathrm{RD-SCM}}$.

---

**Definition D.2. Regional discrete mechanism**

Given $\mathbf{I}_V = \{I_V^r\}_{r \in [1,R]}$ a partition of $R$ parts of $\Omega_{U_V}$ and $m_V = \{m_V^r : \Omega_{\boldsymbol{PA}(V)} \mapsto \Omega_V\}_{r \in [1,R]}$ a set of $R$ distinct mappings from $\Omega_{\boldsymbol{PA}(V)}$ to $\Omega_V$, the **regional discrete mechanism** of an endogenous variables $V$ is a function $f_V : \Omega_{\boldsymbol{PA}(V)}, \Omega_{U_V} \mapsto \Omega_V$ such that:

$$f_V(\boldsymbol{PA}(V),\ u_V) = m_r(\boldsymbol{PA}(V) \mapsto V) \text{ when } u_V \in I_V^r$$

$I_V^r$ and $m_r$ are called the $r^{th}$ noise region and mapping of the regional discrete mechanism $f_V$.

---

**Remark on $\Omega_{U_V}$ and $R$:**   In the definition of a regional discrete mechanism (Definition D.2), no constraints are imposed on $\Omega_{U_V}$. However, if $\Omega_{U_V}$ is discrete, then $|\Omega_{U_V}| \geq R$ is required to form a partition of $R$ elements of $\Omega_{U_V}$. Consequently, in order to be able to constitute such a partition for any finite $R$, we decided to consider continuous exogenous variables in the definition of a regional discrete SCM (Definition D.1). In addition, since the $m_V^r$ mappings are considered distinct and there are exactly $|\Omega_V|^{|\Omega_{\boldsymbol{PA}(V)}|}$ different mappings from $V$ to $\boldsymbol{PA}(V)$, $R \leq |\Omega_V|^{|\Omega_{\boldsymbol{PA}(V)}|}$ is required.

Even if regional discrete SCMs are Markovian, the fact that they contains two types of endogenous variables (i.e., observed and unobserved by the user) enables the representation of complex situations where not all variables are observable. This induces the presence of potential hidden confounders from the user's perspective. As a result, the causal sufficiency assumption is no longer always respected. In our parametric definition of a *SoI*, this phenomenon is controlled by the parameter specifying the proportion of unobserved variables among the endogenous variables. Thus, if this parameter is set to 0, the *SoI*'s class of SCMs is included in the class of causally sufficient discrete SCMs.

The complexity of discrete mechanisms can be controlled by the number of noise regions $R$. Indeed, as the number of noise regions increases, so does the complexity of the causal mechanism, in the sense that it becomes a mixture of a larger number of mappings. The distribution of a variable given its parents is, hence, more stochastic. As a result, the user-defined class of regional discrete SCMs can be very broad. This provides an additional degree of complexity to make our synthetic causal datasets less trivial.

The class of regional discrete SCMs got inspired by the class of Regional Canonical Models by (Xia et al., 2023) and the class of canonical SCMs by (Zhang et al., 2022). We decided to define our own class rather than using one of these two classes for two reasons. First, canonical SCMs are very expensive to sample particularly because of the presence of confounded components. Second, even if Regional Canonical Models are designed to be less expensive because their

expressivity can be regulated via the number of noise regions to consider, they lose some interesting properties such as the non overlapping of the noise regions which is crucial to favor a strong dependence between the user choice of the number of noise regions and the complexity of the generated mechanisms. Moreover, Regional Canonical Models still rely on confounded components which is the major source of complexity at the sampling stage. Hence, we defined the class of Regional discrete SCMs to not have to deal with confounded components at the sampling stage (instead we rely on a projection algorithm after sampling, see Appendix C) and to regulate mechanisms expressivity through the use of non-overlapping noise regions.

### D.2. Discrete Mechanism Sampling strategies

We use *regional discrete mechanisms* (Definition D.2), which define tabular mappings from parent variables to a target variable, conditioned on regions of the exogenous noise space. By default, each region induces a distinct mapping, enabling both stochasticity and high functional expressivity.

To generate these mechanisms, we support three sampling strategies described below. All methods define a partition of the exogenous noise domain $\Omega_U$ into $R$ regions, and assign a parent-to-child mapping to each region. Let $C$ be the cardinality of the variables, and $\Omega_{\text{Pa}(V)}$ the space of parent configurations for variable $V$.

**Controlling complexity.**  The number of possible mappings from parent configurations to output values grows as $|\Omega_V|^{|\Omega_{\text{Pa}(V)}|}$. To keep simulations tractable, users can control the number of noise regions $R$. When $R$ is small, sampling provides diverse but lightweight mechanisms. When $R$ approaches the total number of mappings, full enumeration becomes feasible but computationally expensive.

We now describe the three supported sampling strategies.

EXHAUSTIVE PARTITION

This strategy enumerates all possible mappings from parent configurations to output values and assigns each one to a distinct noise region ($R = |\Omega_V|^{|\Omega_{\text{Pa}(V)}|}$), ensuring complete coverage of the function space. This method guarantees maximal functional diversity across regions and can serve as a stress test for generalization under highly non-linear mechanisms. This is the only strategy where the number of noise regions is not decided by the user but rather set to the maximum. The exhaustive partition sampling strategy is the one to use if one wants the coverage guarantee (Proposition 5.1) to apply.

SAMPLE REJECTION

This strategy samples parent-to-output mappings uniformly at random, rejecting duplicates to ensure that each region corresponds to a distinct function. As mappings are sampled with replacement, rejection may require several attempts when $R$ approaches the number of possible mappings.

We provide below, in Algorithm 4, a pseudocode version of this strategy. The algorithm proceeds as follows. For each endogenous variable $V$ (line 2) a regional discrete mechanism is created. To do so, the domain of $V$ is first initialized with a list of integers corresponding of the cardinality specified in the *SoI* (line 3). Then, if the number of noise regions $R$ specified in the *SoI* is larger than the maximum number of noise regions, the maximum number of noise regions is used to generate the regional discrete mechanism (lines 4-5). The partition of the noise regions is built as consecutive intervals of random size resulting from the ordering of $R - 1$ sampled realizations of the uniform exogenous distribution (lines 6 to 9 and 13). Finally, for each noise region $r$ (line 12), mappings $m_V^r$ are sampled till one mapping not already used for other noise regions is sampled (lines 15 to 18). This is why this algorithm is denoted as the "sample rejection" approach. One can note that there are two sources of randomness in this algorithm: the size of the noise regions and the sampled mappings whenever the number of noise regions is not maximal.

UNBIASED RANDOM ASSIGNMENT

In this strategy, each noise region is assigned a mapping sampled independently and without enforcing uniqueness. As a result, multiple regions may correspond to the same function from parent configurations to outputs.

For example, suppose a variable has one binary parent taking values in $\{0, 1\}$, and the output variable takes values in $\{0, 1, 2\}$. One randomly sampled mapping might assign output 0 to parent value 0, and output 2 to parent value 1. Since mappings are sampled independently for each region, this same function ($0 \to 0, 1 \to 2$) may appear in multiple regions by

chance.

This approach reflects scenarios where mechanisms are drawn independently from a distribution over functions, without enforcing any requirements on uniqueness or coverage. As a result, the effective variability in the entire system may be lower compared to other strategies, but the sampling is a lot more computationally efficient.

---

**Algorithm 4** Generating regional discrete mechanisms with sample rejection

---

**Inputs:** set of endogenous variables $\mathbf{V}$ of cardinality $C$, causal graph $\mathcal{G}$, $\Omega_U$ domain of exogenous variables, number of noise regions $R$

1: $\mathcal{F} \leftarrow \{\}$
2: **for** $V \in \mathbf{V}$ **do**
3: $\quad \Omega_V \leftarrow \{1, \dots, C\}$
4: $\quad \Omega_{PA_{\mathcal{G}}(V)} \leftarrow \{1, \dots, C\}^{|PA_{\mathcal{G}}(V)|}$
5: $\quad R \leftarrow \min(R, |\Omega_V|^{|\Omega_{PA(V)}|})$
6: $\quad l_{\min} \leftarrow \inf(\Omega_U)$
7: $\quad l_{\max} \leftarrow \sup(\Omega_U)$
8: $\quad \mathbf{L} = \{l_i \sim \mathcal{U}[l_{\min}, l_{\max}] \mid i \in [1, R-1]\} \cup \{l_{\min}, l_{\max}\}$
9: $\quad$ Sort $\mathbf{L}$ in ascending order
10: $\quad f_V \leftarrow \{\}$
11: $\quad m_V \leftarrow \{\}$
12: $\quad$ **for** $r \in [1, R]$ **do**
13: $\quad\quad I_V^r \leftarrow [\mathbf{L}_r, \mathbf{L}_{r+1}[$ with $\mathbf{L}_r$ the $r^{th}$ element of $\mathbf{L}$
14: $\quad\quad m_V^r \leftarrow \{\}$
15: $\quad\quad$ **while** $m_V^r = \{\}$ or $m_V^r \in m_V$ **do**
16: $\quad\quad\quad m_V^r \leftarrow |\Omega_{PA(V)}|$ elements sampled with replacement from $\Omega_V$
17: $\quad\quad$ **end while**
18: $\quad\quad m_V \leftarrow m_V \cup m_V^r$
19: $\quad\quad f_V \leftarrow f_V \cup \{m_V^r; I_V^r\}$
20: $\quad$ **end for**
21: $\quad \mathcal{F} \leftarrow \mathcal{F} \cup f_V$
22: **end for**

**Output:** $\mathcal{F}$

---

# E. Query Sampling and Estimation

In this work, we consider the following types of queries: Average Treatment Effect (ATE), Conditional Average Treatment Effect (CATE) and Counterfactual Total Effect (Ctf-TE). Their definitions can be found in Appendix A. All the queries can be defined for sets of covariates and factuals belonging to the set of endogenous variables. In other words, we do not implement multi-interventions, but we consider conditioning and observing factuals on several variables. Finally, the values taken by these variables (e.g., treatment and control values for ATE) must belong to their definition domain. The only parameter that controls the queries class is the type of queries chosen by the user (i.e., ATE, CATE and Ctf-TE). Thus, the class of considered queries can be defined as follows:

$$\mathbb{Q}_{\text{ATE}} = \{\text{ATE}_{T \to Y}(t, c) \mid T, Y \subseteq \mathbf{V} \text{ and } t, c \in \Omega_T\}$$

$$\mathbb{Q}_{\text{CATE}} = \{\text{CATE}_{T \to Y|\mathbf{X}}(t, c, \mathbf{x}) \mid T, Y \subseteq \mathbf{V}, \ \mathbf{X} \subseteq \mathbf{V} \backslash \{T, Y\} \text{ and } t, c \in \Omega_T, \ \mathbf{x} \in \Omega_{\mathbf{X}}\}$$

$$\mathbb{Q}_{\text{Ctf-TE}} = \{\text{Ctf-TE}_{T \to Y}(y, t, c, \boldsymbol{v}_F) \mid T, Y, \boldsymbol{V}_F \subseteq \mathbf{V} \text{ and } t, c \in \Omega_T, \ y \in \Omega_Y, \ \boldsymbol{v}_F \in \Omega_{\boldsymbol{V}_F}\}$$

Formally speaking, we have not integrated the causal graph as a causal query but rather as a hypothesis or prior knowledge. Indeed, except for causal discovery tasks, the causal graph is most often assumed to be known (or at least some information derived from the graph, such as the constitution of a valid adjustment set, or a valid causal ordering). Nevertheless, one can use our random causal dataset generator to evaluate causal discovery or causal representation learning methods. To do so, one just needs to retrieve the causal graph from the causal dataset directly instead of using a query.

Finally, a user can also implement a specific query and use it to generate synthetic causal datasets. To do this, the user has to use the Query class in our code base.

## E.1. Query Sampling

As the values taken by variables in the queries have to belong to their definition domain, we draw realizations from a large, separately sampled observational dataset. Indeed, given the randomness of the causal mechanisms, we cannot know in advance the domain over which the SCMs are defined. Even when variable cardinalities are fixed, the sampled mechanisms may be non-surjective, making certain values impossible to observe. For this reason, we approximate the domain of definition through data sampling, ensuring that queries are computed only for realizable variable configurations. Moreover, since the dataset given to the user is smaller to the one we use for query sampling and estimation, it is possible that queries use values outside of the observational dataset or that they are non-identifiable. Explicitly enabling queries to be outside the observed dataset can be useful for studying generalization—especially in settings where the support is known, such as linear SCMs. However, we let for future work the devlopement of a user-configurable option in *SoIs*, for instance, allowing users to define a custom domain for the query variables.

The following algorithms detail the procedures for sampling ATE, CATE, and Ctf-TE queries. In these algorithms, given a dataset $D$, a variable $X$ and a realization $x$ of $X$, we use the notation $D_{|X}$ (resp. $D_{|X=x}$) to represent the dataset $D$ restricted to the variable $X$ (resp. restricted to the samples whose $X$ realization equals $x$). In addition, $B(n, p)$ denotes the Binomial law of parameters $n$ and $p$.

---

**Algorithm 5** Generating sets of observed data

---

**Inputs:** causal graph $\mathcal{G}$, causal mechanisms $\mathcal{F}$, distribution of the exogenous variables $P(\mathbf{U})$, dataset size $N$

1: $D \leftarrow \{\}$
2: $D_o \leftarrow \{\}$
3: $\{\mathbf{u}_1, \ldots, \mathbf{u}_N\} \sim P(\mathbf{U})$
4: **for** $V \in \mathbf{V}$ following a causal order given by $\mathcal{G}$ **do**
5: $\quad \{\mathbf{pa}(V)_1, \ldots, \mathbf{pa}(V)_N\} \leftarrow D_{|\mathbf{PA}(V)}$
6: $\quad \{u_{V_1}, \ldots, u_{V_N}\} \leftarrow D_{|\mathbf{U}_V}$
7: $\quad \{v_1, \ldots, v_N\} \leftarrow f_V(\{\mathbf{pa}(V)_1, \ldots, \mathbf{pa}(V)_N\}, \{u_{V_1}, \ldots, u_{V_N}\})$
8: $\quad D \leftarrow D \cup \{v_1, \ldots, v_N\}$
9: $\quad$ **if** $V \in \mathbf{V}_o$ **then**
10: $\quad\quad D_o \leftarrow D_o \cup \{v_1, \ldots, v_N\}$
11: $\quad$ **end if**
12: **end for**

**Output:** $D_o$

---

**Algorithm 6** Generating ATE queries

---

**Inputs:** set of observable endogenous variables $\mathbf{V}_o$, training set $D$

1: $T \leftarrow$ one variable randomly sampled from $\mathbf{V}_o$
2: $Y \leftarrow$ one variable randomly sampled from $\mathbf{V}_o$
3: $t \leftarrow$ one realization of $T$ randomly sampled from $D_{|T}$
4: $c \leftarrow$ one realization of $T$ randomly sampled from $D_{|T}$

**Output:** $Q_{ATE} = \{T, Y, t, c\}$

---

**Algorithm 7** Generating CATE queries

---

**Inputs:** set of observable endogenous variables $\mathbf{V}_o$, training set $D$

1: $T \leftarrow$ one variable randomly sampled from $\mathbf{V}_o$
2: $Y \leftarrow$ one variable randomly sampled from $\mathbf{V}_o$
3: $d_{\mathbf{X}} \leftarrow$ an integer randomly sampled from $[1, \ldots, |\mathbf{V}_o| - 2]$
4: $\mathbf{X} \leftarrow d_{\mathbf{X}}$ variables randomly sampled from $\mathbf{V}_o \backslash \{T, Y\}$
5: $t \leftarrow$ one realization of $T$ randomly sampled from $D_{|T}$
6: $c \leftarrow$ one realization of $T$ randomly sampled from $D_{|T}$
7: $\mathbf{x} \leftarrow$ one realization of $\mathbf{X}$ randomly sampled from $D_{|\mathbf{X}}$

**Output:** $Q_{CATE} = \{T, Y, \mathbf{X}, t, c, \mathbf{x}\}$

---

**Algorithm 8** Generating Ctf-TE queries

---

**Inputs:** set of observable endogenous variables $\mathbf{V}_o$, training set $D$

1: $T \leftarrow$ one variable randomly sampled from $\mathbf{V}_o$
2: $Y \leftarrow$ one variable randomly sampled from $\mathbf{V}_o$
3: $d_{\mathbf{V}_F} \leftarrow$ an integer randomly samples from $[1, \ldots, |\mathbf{V}_o|]$
4: $\mathbf{V}_F \leftarrow d_{\mathbf{V}_F}$ variables randomly sampled from $\mathbf{V}_o$
5: $t \leftarrow$ one realization of $T$ randomly sampled from $D_{|T}$
6: $c \leftarrow$ one realization of $T$ randomly sampled from $D_{|T}$
7: $\mathbf{v}_F \leftarrow$ one realization of $\mathbf{V}_F$ randomly sampled from $D_{|\mathbf{V}_F}$

**Output:** $Q_{CTF-TE} = \{T, Y, \mathbf{V}_F, t, c, \mathbf{v}_F\}$

---

### E.2. SCM-Based Query Estimation

Each query is evaluated by modifying the SCM, sampling the exogenous variables, and computing expectations over the outcomes. In practice, we simulate interventions and counterfactuals by directly manipulating structural equations and

conditioning on sampled variables. Our implementation supports efficient batch estimation using the same random seeds for reproducibility.

Queries that yield `NaN` estimates can optionally be rejected and resampled, depending on the *SoI* settings. `NaN` estimates appear if the corresponding sampled query is undefined (e.g., conditioning on a zero-probability event). However, to evaluate the ability of some models to identify if the query is undefined instead of trying to answer it, `NaN` estimates can be interesting to keep. This is why we decided to let users choose this option through a parameter of the *SoI*.

The following algorithms detail the procedures for estimating ATE, CATE, and Ctf-TE queries.

---

**Algorithm 9** Estimating ATE queries

---

**Inputs:** ATE query to estimate $Q = \{T, Y, t, c\}$, causal graph $\mathcal{G}$, causal mechanisms $\mathcal{F}$, distribution of the exogenous variables $P(\mathbf{U})$, number of samples to draw for estimation $N$

1: $\{\mathbf{u}_1, \ldots, \mathbf{u}_N\} \sim P(\mathbf{U})$
2: $D_t \leftarrow \{\mathbf{u}_1, \ldots, \mathbf{u}_N\}$
3: **for** $V \in \mathbf{V}$ following a causal order given by $\mathcal{G}$ **do**
4:     **if** $V = T$ **then**
5:         $\{v_1, \ldots, v_N\} \leftarrow \{t, \ldots, t\}$
6:     **else**
7:         $\{\mathbf{pa}(V)_1, \ldots, \mathbf{pa}(V)_N\} \leftarrow D_{t|\boldsymbol{PA}(V)}$
8:         $\{u_{V_1}, \ldots, u_{V_N}\} \leftarrow D_{t|\mathbf{U}_V}$
9:         $\{v_1, \ldots, v_N\} \leftarrow f_V(\{\mathbf{pa}(V)_1, \ldots, \mathbf{pa}(V)_N\}, \{u_{V_1}, \ldots, u_{V_N}\})$
10:    **end if**
11:    $D_t \leftarrow D_t \cup \{v_1, \ldots, v_N\}$
12: **end for**
13: $D_c \leftarrow \{\mathbf{u}_1, \ldots, \mathbf{u}_N\}$
14: **for** $V \in \mathbf{V}$ following a causal order given by $\mathcal{G}$ **do**
15:    **if** $V = T$ **then**
16:       $\{v_1, \ldots, v_N\} \leftarrow \{c, \ldots, c\}$
17:    **else**
18:       $\{\mathbf{pa}(V)_1, \ldots, \mathbf{pa}(V)_N\} \leftarrow D_{c|\boldsymbol{PA}(V)}$
19:       $\{u_{V_1}, \ldots, u_{V_N}\} \leftarrow D_{c|\mathbf{U}_V}$
20:       $\{v_1, \ldots, v_N\} \leftarrow f_V(\{\mathbf{pa}(V)_1, \ldots, \mathbf{pa}(V)_N\}, \{u_{V_1}, \ldots, u_{V_N}\})$
21:    **end if**
22:    $D_c \leftarrow D_c \cup \{v_1, \ldots, v_N\}$
23: **end for**
24: $Q^\star \leftarrow \text{avg}(D_{t|Y}) - \text{avg}(D_{c|Y})$
**Output:** $Q^\star$

---

**Algorithm 10** Estimating CATE queries

**Inputs:** CATE query to estimate $Q = \{T, Y, \mathbf{X}, t, c, \mathbf{x}\}$, causal graph $\mathcal{G}$, causal mechanisms $\mathcal{F}$, distribution of the exogenous variables $P(\mathbf{U})$, number of samples to draw for estimation $N$

1: $\{\mathbf{u}_1, \ldots, \mathbf{u}_N\} \sim P(\mathbf{U})$
2: $D_t \leftarrow \{\mathbf{u}_1, \ldots, \mathbf{u}_N\}$
3: **for** $V \in \mathbf{V}$ following a causal order given by $\mathcal{G}$ **do**
4:     **if** $V = T$ **then**
5:         $\{v_1, \ldots, v_N\} \leftarrow \{t, \ldots, t\}$
6:     **else**
7:         $\{\mathbf{pa}(V)_1, \ldots, \mathbf{pa}(V)_N\} \leftarrow D_{t|\boldsymbol{PA}(V)}$
8:         $\{u_{V_1}, \ldots, u_{V_N}\} \leftarrow D_{t|\mathbf{U}_V}$
9:         $\{v_1, \ldots, v_N\} \leftarrow f_V(\{\mathbf{pa}(V)_1, \ldots, \mathbf{pa}(V)_N\}, \{u_{V_1}, \ldots, u_{V_N}\})$
10:    **end if**
11:    $D_t \leftarrow D_t \cup \{v_1, \ldots, v_N\}$
12: **end for**
13: $D_c \leftarrow \{\mathbf{u}_1, \ldots, \mathbf{u}_N\}$
14: **for** $V \in \mathbf{V}$ following a causal order given by $\mathcal{G}$ **do**
15:    **if** $V = T$ **then**
16:       $\{v_1, \ldots, v_N\} \leftarrow \{c, \ldots, c\}$
17:    **else**
18:       $\{\mathbf{pa}(V)_1, \ldots, \mathbf{pa}(V)_N\} \leftarrow D_{c|\boldsymbol{PA}(V)}$
19:       $\{u_{V_1}, \ldots, u_{V_N}\} \leftarrow D_{c|\mathbf{U}_V}$
20:       $\{v_1, \ldots, v_N\} \leftarrow f_V(\{\mathbf{pa}(V)_1, \ldots, \mathbf{pa}(V)_N\}, \{u_{V_1}, \ldots, u_{V_N}\})$
21:    **end if**
22:    $D_c \leftarrow D_c \cup \{v_1, \ldots, v_N\}$
23: **end for**
24: $D_t \leftarrow D_{t|\mathbf{X}=\mathbf{x}}$
25: $D_c \leftarrow D_{c|\mathbf{X}=\mathbf{x}}$
26: $Q^\star \leftarrow \mathrm{avg}(D_{t|Y}) - \mathrm{avg}(D_{c|Y})$

**Output:** $Q^\star$

---

**Algorithm 11** Estimating Ctf-TE queries

---

**Inputs:** Ctf-TE query to estimate $Q = \{T, Y, \mathbf{V}_F, t, c, \mathbf{v}_F\}$, causal graph $\mathcal{G}$, causal mechanisms $\mathcal{F}$, distribution of the exogenous variables $P(\mathbf{U})$, number of samples to draw for estimation $N$

1: $\{\mathbf{u}_1, \ldots, \mathbf{u}_N\} \sim P(\mathbf{U})$
2: $D_{\mathbf{U}_{\mathbf{v}_F}} \leftarrow \{\mathbf{u}_1, \ldots, \mathbf{u}_N\}$
3: **for** $V \in \mathbf{V}$ following a causal order given by $\mathcal{G}$ **do**
4:      $\{\mathbf{pa}(V)_1, \ldots, \mathbf{pa}(V)_N\} \leftarrow D_{\mathbf{U}_{\mathbf{v}_F} | \mathbf{PA}(V)}$
5:      $\{u_{V_1}, \ldots, u_{V_N}\} \leftarrow D_{\mathbf{U}_{\mathbf{v}_F} | \mathbf{U}_V}$
6:      $\{v_1, \ldots, v_N\} \leftarrow f_V(\{\mathbf{pa}(V)_1, \ldots, \mathbf{pa}(V)_N\}, \{u_{V_1}, \ldots, u_{V_N}\})$
7:      $D_{\mathbf{U}_{\mathbf{v}_F}} \leftarrow D_{\mathbf{U}_{\mathbf{v}_F}} \cup \{v_1, \ldots, v_N\}$
8: **end for**
9: $D_{\mathbf{U}_{\mathbf{v}_F}} \leftarrow D_{\mathbf{U}_{\mathbf{v}_F} | \mathbf{V}_F = \mathbf{v}_F}$
10: $M \leftarrow |D_{\mathbf{U}_{\mathbf{v}_F}}|$
11: $\{\mathbf{u}_1, \ldots, \mathbf{u}_M\} \leftarrow D_{\mathbf{U}_{\mathbf{v}_F} | \mathbf{U}}$
12: $D_t \leftarrow \{\mathbf{u}_1, \ldots, \mathbf{u}_M\}$
13: **for** $V \in \mathbf{V}$ following a causal order given by $\mathcal{G}$ **do**
14:      **if** $V = T$ **then**
15:          $\{v_1, \ldots, v_N\} \leftarrow \{t, \ldots, t\}$
16:      **else**
17:          $\{\mathbf{pa}(V)_1, \ldots, \mathbf{pa}(V)_N\} \leftarrow D_{t | \mathbf{PA}(V)}$
18:          $\{u_{V_1}, \ldots, u_{V_N}\} \leftarrow D_{t | \mathbf{U}_V}$
19:          $\{v_1, \ldots, v_N\} \leftarrow f_V(\{\mathbf{pa}(V)_1, \ldots, \mathbf{pa}(V)_N\}, \{u_{V_1}, \ldots, u_{V_N}\})$
20:      **end if**
21:      $D_t \leftarrow D_t \cup \{v_1, \ldots, v_N\}$
22: **end for**
23: $D_c \leftarrow \{\mathbf{u}_1, \ldots, \mathbf{u}_M\}$
24: **for** $V \in \mathbf{V}$ following a causal order given by $\mathcal{G}$ **do**
25:      **if** $V = T$ **then**
26:          $\{v_1, \ldots, v_N\} \leftarrow \{c, \ldots, c\}$
27:      **else**
28:          $\{\mathbf{pa}(V)_1, \ldots, \mathbf{pa}(V)_N\} \leftarrow D_{c | \mathbf{PA}(V)}$
29:          $\{u_{V_1}, \ldots, u_{V_N}\} \leftarrow D_{c | \mathbf{U}_V}$
30:          $\{v_1, \ldots, v_N\} \leftarrow f_V(\{\mathbf{pa}(V)_1, \ldots, \mathbf{pa}(V)_N\}, \{u_{V_1}, \ldots, u_{V_N}\})$
31:      **end if**
32:      $D_c \leftarrow D_c \cup \{v_1, \ldots, v_N\}$
33: **end for**
34: $Q^\star \leftarrow \text{avg}(D_{t|Y}) - \text{avg}(D_{c|Y})$

**Output:** $Q^\star$

---

## F. Analysis Module Metrics

In order to analyze the characteristics of the sampled SCMs we implemented the following metrics. Let us imagine we sampled an SCM $\mathcal{M} := (\mathbf{V}, \mathbf{U}, \mathcal{F}, P(\mathbf{U}))$ with $\mathbf{V} = (\mathbf{V}_o, \mathbf{V}_h)$ and whose causal graph is denoted $\mathcal{G}$. The projection of $\mathcal{G}$ over the observable variables $\mathbf{V}_o$ is denoted $\mathcal{G}_{\mathbf{V}_o}$.

**Analysis of the causal graph $\mathcal{G}$:**

- Average in-degree: $\bar{d}_{in} = \frac{1}{|\mathbf{V}|} \sum_{V \in \mathbf{V}} |\boldsymbol{PA}(V)|$

- Variance of in-degree: $\text{var}(d_{in}) = \frac{1}{|\mathbf{V}|} \sum_{V \in \mathbf{V}} (|\boldsymbol{PA}(V)| - \bar{d}_{in})^2$

- Average number of ancestors: $\overline{|An(V)|} = \frac{1}{|\mathbf{V}|} \sum_{V \in \mathbf{V}} |An(V)|$ where $An(V)$ denotes the set of ancestors of $V$

- Variance of number of ancestors: $\text{var}(|An(V)|) = \frac{1}{|\mathbf{V}|} \sum_{V \in \mathbf{V}} (|An(V)| - \overline{|An(V)|})^2$

- Average number of descendants: $\overline{|De(V)|} = \frac{1}{|\mathbf{V}|} \sum_{V \in \mathbf{V}} |De(V)|$ where $De(V)$ denotes the set of descendants of $V$

- Variance of number of descendants: $\text{var}(|De(V)|) = \frac{1}{|\mathbf{V}|} \sum_{V \in \mathbf{V}} (|De(V)| - \overline{|De(V)|})^2$

- Average length of causal paths: $\overline{L} = \frac{1}{|\mathbf{P}_\mathcal{G}|} \sum_{p \in \mathbf{P}_\mathcal{G}} |p|$ where $\mathbf{p}_\mathcal{G}$ denotes the set of directed paths in $\mathcal{G}$

- Variance length of causal paths: $\text{var}(L) = \frac{1}{|\mathbf{P}_\mathcal{G}|} \sum_{p \in \mathbf{P}_\mathcal{G}} (|p| - \overline{L})^2$

- Maximum length of causal paths: $L_{\max} = \max_{p \in \mathbf{P}_\mathcal{G}} |p|$

**Analysis of the projected causal graph $\mathcal{G}_{\mathbf{V}_o}$:**

- Average number of siblings[4]: $\overline{|Si(V)|} = \frac{1}{|\mathbf{V}_o|} \sum_{V \in \mathbf{V}_o} |Si(V)|$ where $Si(V)$ denotes the set of siblings of $V$

- Variance of number of siblings: $\text{var}(|Si(V)|) = \frac{1}{|\mathbf{V}_o|} \sum_{V \in \mathbf{V}_o} (|Si(V)| - \overline{|Si(V)|})^2$

- Number of maximal confounded components (c-comps)[5]: $|\mathbf{C}|$ where $\mathbf{C}$ denotes the set of maximal c-comps in $\mathcal{G}_{\mathbf{V}_o}$

- Average size of maximal c-comps: $\overline{|\mathbf{C}|} = \frac{1}{|\mathbf{C}|} \sum_{C \in \mathbf{C}} |C|$

- Variance of the size of maximal c-comps: $\text{var}(|\mathbf{C}|) = \frac{1}{|\mathbf{C}|} \sum_{C \in \mathbf{C}} (|C| - \overline{|\mathbf{C}|})^2$

**Analysis of the observational distribution $P_\mathcal{M}(\mathbf{V_o})$:**

- Minimum probability of the joint distribution: $p_{\mathbf{V}_o,\min} = \min_{\mathbf{v}_o \in \Omega_{\mathbf{V}_o}} P_\mathcal{M}(\mathbf{V}_o = \mathbf{v}_o)$

- Proportion of events with a null probability: $p_0 = \frac{1}{|\Omega_{\mathbf{V}_o}|} \sum_{\mathbf{v}_o \in \Omega_{\mathbf{V}_o}} \mathbf{1}_{P_\mathcal{M}(\mathbf{V_o}=\mathbf{v}_o)=0}$ where $\mathbf{1}_-$ denotes the indicator function

- Minimum probability of the marginal distributions:

$$p_{\min} = \min_{V \in \mathbf{V}_o} \min_{v \in \Omega_V} P_\mathcal{M}(V = v)$$

---

[4]Two variables are considered siblings if they are linked by a bi-directed edge.
[5]We use (Tian & Pearl, 2002) definition of (maximal) confounded components.

- Average minimum probability of the marginal distributions:

$$\bar{p}_{\min} = \frac{1}{|\mathbf{V}_o|} \sum_{V \in \mathbf{V}_o} \frac{1}{|\Omega_V|} \min_{v \in \Omega_V} P_{\mathcal{M}}(V = v)$$

- Variance of the minimum probability of the marginal distributions:

$$\text{var}(p_{\min}) = \frac{1}{|\mathbf{V}_o|} \sum_{V \in \mathbf{V}_o} (\min_{v \in \Omega_V} P_{\mathcal{M}}(V = v) - \bar{p}_{\min})^2$$

- Distance ($L_1$) of the joint distributions to the uniform one:

$$d(P_{\mathcal{M}}; \mathcal{U}) = \sum_{\mathbf{v}_o \in \Omega_{\mathbf{V}_o}} |P_{\mathcal{M}}(\mathbf{V}_o = \mathbf{v}_o) - \frac{1}{|\Omega_{\mathbf{V}_o}|}|$$

- Average distance ($L_1$) of the marginal distributions to the uniform one:

$$\overline{d(P_{\mathcal{M}}; \mathcal{U})} = \frac{1}{|\mathbf{V}_o|} \sum_{V \in \mathbf{V}_o} \sum_{v \in \Omega_V} |P_{\mathcal{M}}(V = v) - \frac{1}{|\Omega_V|}|$$

- Variance of the distance ($L_1$) of the marginal distributions to the uniform one:

$$\text{var}(d(P_{\mathcal{M}}; \mathcal{U})) = \frac{1}{|\mathbf{V}_o|} \sum_{V \in \mathbf{V}_o} \left( \sum_{v \in \Omega_V} |P_{\mathcal{M}}(V = v) - \frac{1}{|\Omega_V|}| - \overline{d(P_{\mathcal{M}}; \mathcal{U})} \right)^2$$

- Entropy of the joint distribution: $\text{H}(P_{\mathcal{M}}(\mathbf{V}))$

All the above-mentioned probabilities are computed from a set of 1M samples drawn from the SCM $\mathcal{M}$.

Let us note that $p_{\min}$ enables the user to check if the strong positivity assumption holds. If $p_{\mathbf{V}_o,\min} > 0$, then strong positivity is respected. In addition, if strong positivity does not hold, $p_{\mathbf{V}_o,\min}$ and $p_0$ indicate the extent to which the assumption is not met – the higher the metrics, the less the hypothesis is respected. On the other hand, $p_{\min}$ indicates whether the weak positivity assumption holds. If $p_{\min} > 0$, then weak positivity is respected. Finally, $d(P_{\mathcal{M}}; \mathcal{U})$, $\overline{d(P_{\mathcal{M}}; \mathcal{U})}$ and $\text{var}(d(P_{\mathcal{M}}; \mathcal{U}))$ enables the user to assess to which extent the observational distribution is imbalanced.

**Analysis of the causal mechanisms $\mathcal{F}$:**

- Average Pearson's correlation between the parent-child pairs[6]:

$$\bar{\rho}_P = \frac{1}{|\mathbf{V}|} \sum_{V \in \mathbf{V}} \frac{1}{|\boldsymbol{PA}(V) \cup U_V|} \sum_{V_j \in \boldsymbol{PA}(V) \cup U_V} \rho_P(V, V_j)$$

- Variance of Pearson's correlation between the parent-child pairs:

$$\text{var}(\rho_P) = \frac{1}{|\mathbf{V}|} \sum_{V \in \mathbf{V}} \frac{1}{|\boldsymbol{PA}(V) \cup U_V|} \sum_{V_j \in \boldsymbol{PA}(V) \cup U_V} (\rho_P(V, V_j) - \bar{\rho}_P)$$

- Average Spearman's correlation between the parent-child pairs:

$$\bar{\rho}_S = \frac{1}{|\mathbf{V}|} \sum_{V \in \mathbf{V}} \frac{1}{|\boldsymbol{PA}(V) \cup U_V|} \sum_{V_j \in \boldsymbol{PA}(V) \cup U_V} \rho_S(V, V_j)$$

---

[6]$\rho_P$ and $\rho_S$ respectively denote the Pearson's and Spearman's correlation

- Variance of Spearman's correlation between the parent-child pairs:

$$\text{var}(\rho_S) = \frac{1}{|\mathbf{V}|} \sum_{V \in \mathbf{V}} \frac{1}{|\mathbf{PA}(V) \cup U_V|} \sum_{V_j \in \mathbf{PA}(V) \cup U_V} (\rho_S(V, V_j) - \bar{\rho}_S)$$

- Average conditional entropy of a variable given its parents:

$$\overline{\mathrm{H}} = \frac{1}{|\mathbf{V}|} \sum_{V \in \mathbf{V}} \mathrm{H}(V | \mathbf{PA}(V))$$

- Variance of conditional entropy of a variable given its parents:

$$\text{var}(\mathrm{H}) = \frac{1}{|\mathbf{V}|} \sum_{V \in \mathbf{V}} (\mathrm{H}(V | \mathbf{PA}(V)) - \overline{\mathrm{H}})^2$$

In order to be able to use person correlations, spearman correlations, and conditional entropy as indicators of degrees of linearity, monotonicity, and stochasticity of causal mechanisms, we do not derive these quantities from samples drawn from the entailed distribution. Instead, for each variable, we create a dataset resulting from the application of its causal mechanism to the cartesian product of the values taken by its endogenous and exogenous parents[7]. In other words, we analyze the mechanisms' images of their input space. This allows us to analyze each mechanism independently of the others.

Thus, $\bar{\rho}_P$ and $\text{var}(\rho_P)$ can be interpreted as the average degree of linearity of causal mechanisms and their variance. Furthermore, $\bar{\rho}_S$ and $\text{var}(\rho_S)$ can be interpreted as the average degree of monotonicity of causal mechanisms and their variance. Finally, $\overline{\mathrm{H}}$ and $\text{var}(\mathrm{H})$ can be interpreted as the average level of stochasticity of causal mechanisms and its variance.

---

[7]For continuous SCMs, we first discretize the variables' domains of definition and then build the cartesian product.

# G. Analysis of the Empirical Distribution of the Generated SCMs

As we do not provide the user with an expression of the distribution of the sampled regional discrete SCMs, we need to investigate if some SCMs classes are over/underrepresented. This analysis is important to identify the potential biases CausalProfiler might create in order to take them into account when evaluating Causal ML methods. Indeed, as our goal is to provide a tool for rigorous empirical evaluation of causal methods, we need to be transparent on the limitations of our generator so that researchers and practitioners can interpret the results of their methods with full knowledge of the potential biases coming from CausalProfiler.

## G.1. Experiment

To visualize the distribution of the SCMs generated, we analyze the distribution of the metrics of the analysis module characterizing the SCMs. For each SCM sampled, all the implemented metrics (see Appendix F) are computed.

The studied SCMs are sampled from the SoIs defined by the cartesian product of the following parameters:

- **Number of endogenous variables**: $\{3, 4, 5\}$

- **Expected edge probability**: $\{0.2, 0.4, 0.6, 0.8\}$

- **Proportion of unobserved endogenous variables**: $\{0, 0.1, 0.2, 0.3\}$

- **Number of noise regions**: $\{2, 5, 10, 20, 50\}$

- **Cardinality of endogenous variables**: $\{2, 3, 4, 7\}$

- **Distribution of exogenous variables**: set to $\mathcal{U}[0, 1]$

For each *SoI* 10 SCMs are sampled, making a total of 9600 SCMs studied. Let us mention that we sample more SCMs than for verification (Section 6.1) for two reasons. First, it enables us to have a better approximation of the SCMs distribution. Second, the computation of all the assumptions and characteristics metrics is, in fact, less computationally expensive than computing all the independence tests that were required for verification.

## G.2. Results

The first conclusion, based on Figures 4 to 8, is that the generated SCMs do indeed belong to the specified SoIs and that their characteristics are consistent with the latter.

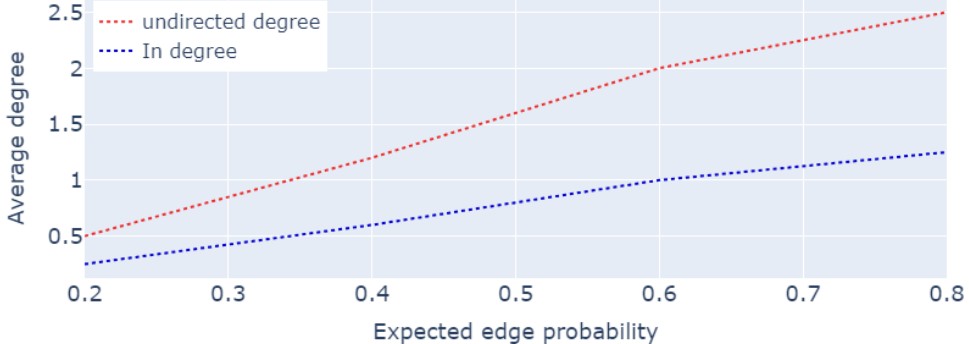

*Figure 4.* Average degree of the causal graphs for the generated SCMs depending on the expected edge probability. Observation: The average degree corresponds on average to the degree of the generated causal graphs.

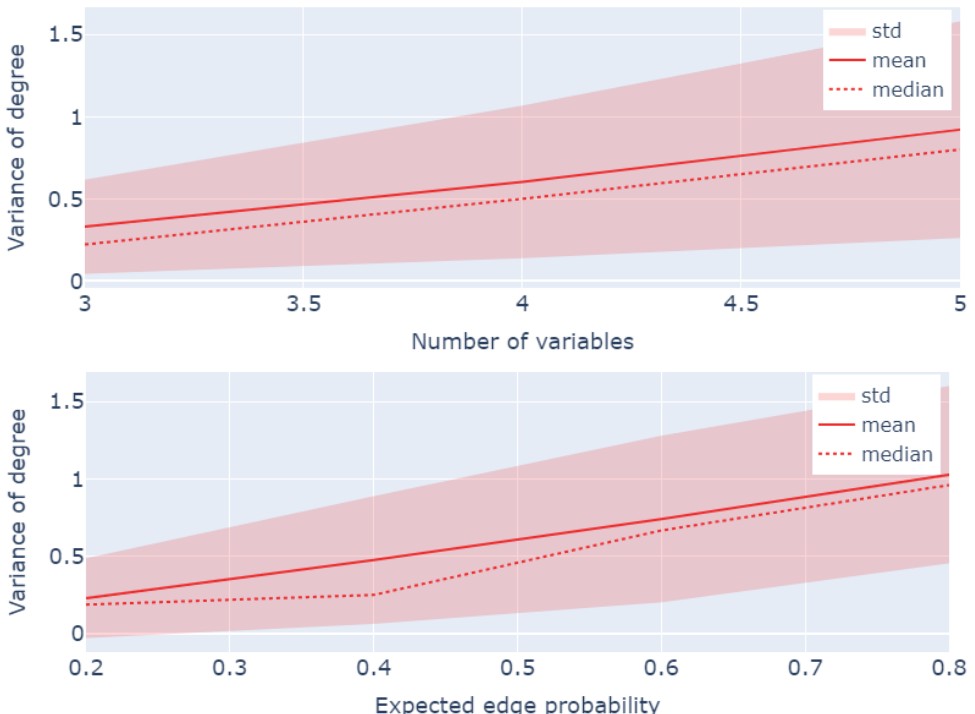

*Figure 5.* Variance of the causal graphs' degree of the generated SCMs depending on the number of variables and the expected edge probability. Observation: The variance of the degree increases with the size of the graph and its density.

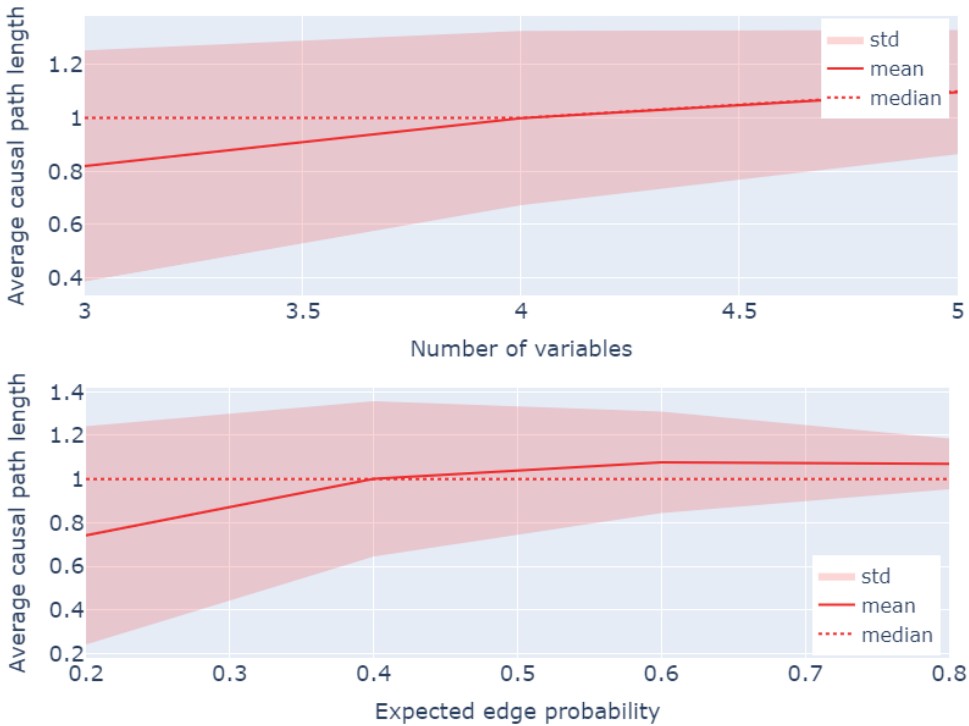

*Figure 6.* Average causal paths length of the causal graphs of the generated SCMs depending on the number of variables and the expected edge probability. Observation: The length of causal paths increases with the size of the causal graph and its density.

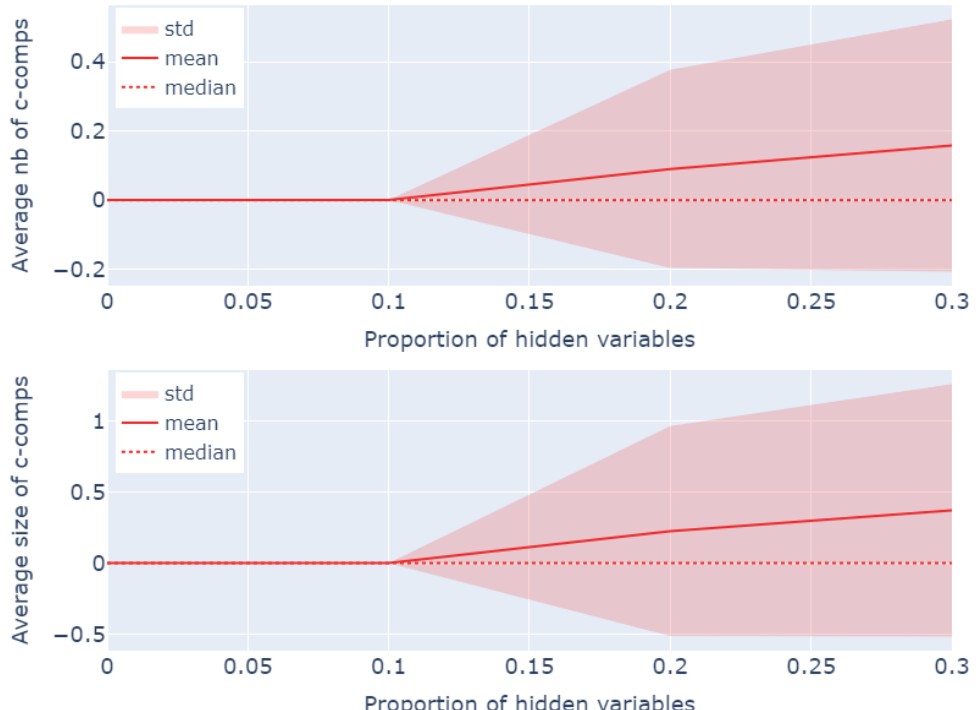

*Figure 7.* Average number and size of maximally confounded components in the projected causal graphs of the generated SCMs depending on the number of unobserved variables. Observation: The number and size of confounded components increase with the proportion of unobserved variables.

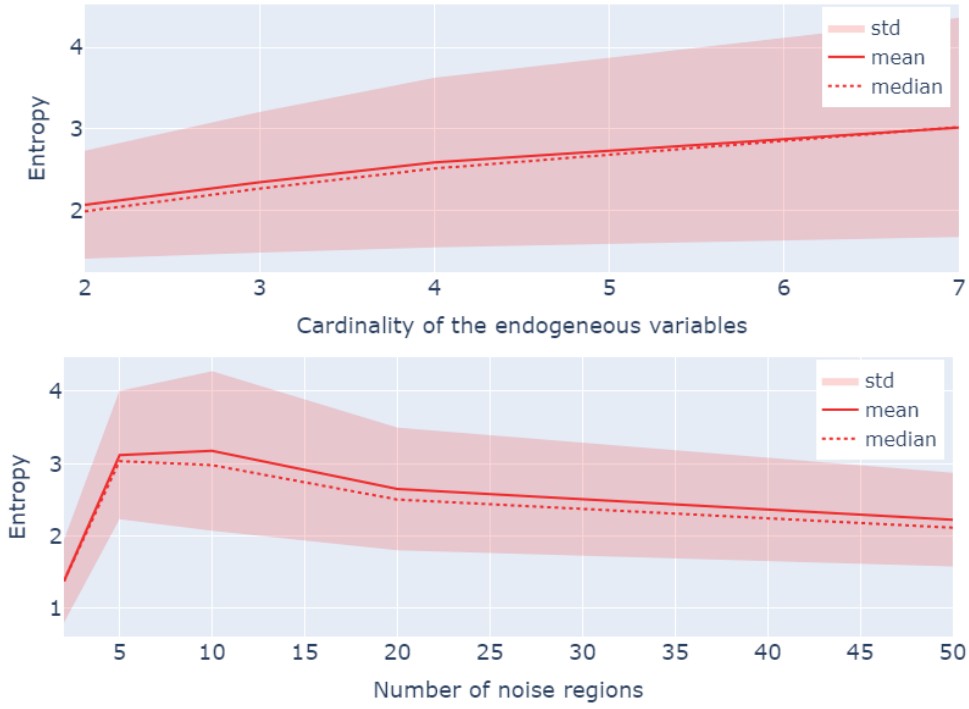

*Figure 8.* Average conditional entropy of a variable given its parents in the generated SCMs depending on the variables' cardinality and the number of noise regions. Observation: The stochasticity of causal mechanisms increases with the cardinality of endogenous and exogenous variables.

In addition, a number of findings about the distribution of the sampled SCMs can also be drawn. First, the number and size of confounded components often equals zero (see also Tables 4 and 5). As highly confounded SCMs are rare, we recommend that users sample SCMs with a large enough number of variables and edge probability, if they want to consider graphs containing hidden confounders. For instance, we recommend at least 10 variables with a 50% edge probability to have a large proportion of graphs with at least one confounded component when setting the proportion of hidden endogenous variables to 30%.

*Table 4.* Percentage of SCMs with confounded components depending on their proportion of unobserved endogenous variables.

| Unobserved endo. variables (%) | Number of maximally confounded components | | |
| --- | --- | --- | --- |
| | 0 | 1 | >1 |
| 0 | 100 | 0 | 0 |
| 10 | 100 | 0 | 0 |
| 20 | 90.9 | 9.1 | 0 |
| 30 | 83.3 | 16.7 | 0 |

*Table 5.* Percentage of SCMs with confounded components of different sizes depending on their proportion of unobserved endogenous variables. The size 1 of confounded components is not referenced, as if a confounded component is not empty, it is at least composed of two variables.

| Unobserved endo. variables (%) | Avg. size of maximally confounded components | | | | |
| --- | --- | --- | --- | --- | --- |
| | 0 | 2 | 3 | 4 | >4 |
| 0 | 100 | 0 | 0 | 0 | 0 |
| 10 | 100 | 0 | 0 | 0 | 0 |
| 20 | 90.9 | 5.7 | 2.3 | 1.0 | 0 |
| 30 | 83.3 | 10.8 | 4.9 | 1.0 | 0 |

Second, analyzing the stochasticity level (measured through the entropy of the $\mathcal{L}_1$ joint distribution, see Appendix F) of the generated SCMs, one can see that the latter can be controlled in part by the parameters of the *SoI*. Indeed, increasing the number of endogenous variables and their cardinality tends to increase the level of stochasticity, see Figure 9 and Tables 6 and 7. This behavior is expected as the discrete mechanisms are randomly sampled with an almost null probability of being deterministic (i.e., the probability of sampling a noise region with an empty support is almost null).

*Table 6.* Mean, standard deviation, and skewness of the distribution of stochasticity level (measured through the entropy of the $\mathcal{L}_1$ joint distribution) over the sampled SCMs depending on their number of endogenous variables. The distribution is displayed in Figure 9.

| Number of endogenous variables | Entropy of the joint distribution | | |
| --- | --- | --- | --- |
| | Mean | Std | Skewness |
| 3 | 2.09 | 0.77 | 0.19 |
| 4 | 2.54 | 1.03 | 0.28 |
| 5 | 2.88 | 1.21 | 0.46 |

*Table 7.* Mean, standard deviation, and skewness of the distribution of stochasticity level (measured through the entropy of the $\mathcal{L}_1$ joint distribution) over the sampled SCMs depending on the cardinality of their endogenous variables. The distribution is displayed in Figure 9.

| | **Entropy of the joint distribution** | | |
|---|---|---|---|
| **Cardinality** | Mean | Std | Skewness |
| 2 | 2.06 | 0.67 | 0.50 |
| 3 | 2.34 | 0.84 | 0.36 |
| 4 | 2.57 | 1.02 | 0.19 |
| 7 | 3.03 | 1.35 | 0.05 |

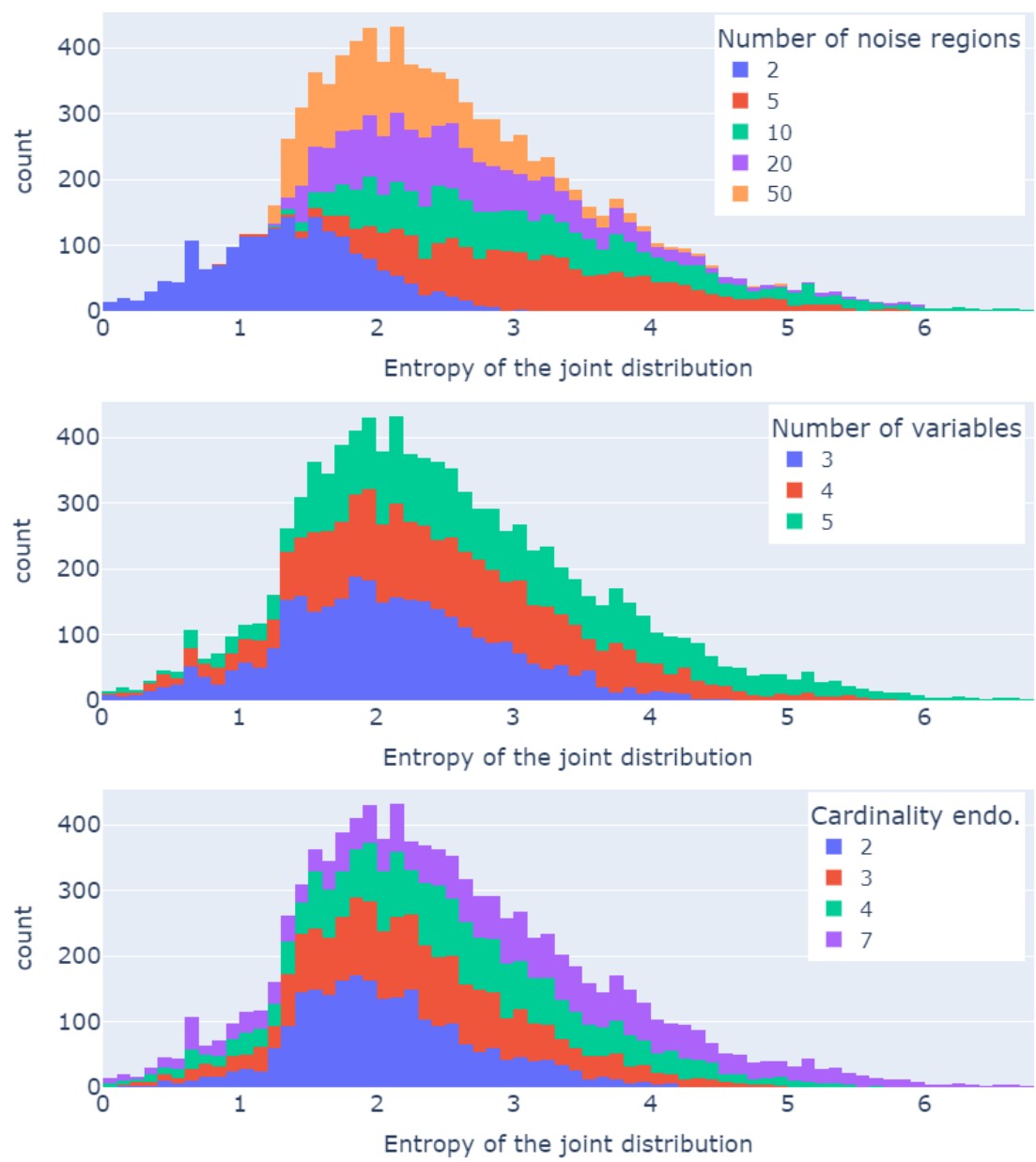

*Figure 9.* Stacked histograms of the stochasticity level (measured through the entropy of the $\mathcal{L}_1$ joint distribution) of the sampled SCMs depending on the number of noise regions, the number of variables, and their cardinality. Mean, standard deviation, and skewness of the distributions can be found in Tables 6 to 8.

*Table 8.* Mean, standard deviation, and skewness of the distribution of stochasticity level (measured through the entropy of the $\mathcal{L}_1$ joint distribution) over the sampled SCMs depending on their number of noise regions. The distribution is displayed in Figure 9.

| Number of noise regions | Entropy of the joint distribution | | |
| :---: | :---: | :---: | :---: |
| | Mean | Std | Skewness |
| 2 | 1.35 | 0.56 | 0.08 |
| 5 | 3.12 | 0.87 | 0.37 |
| 10 | 3.15 | 1.10 | 0.74 |
| 20 | 2.65 | 0.86 | 0.88 |
| 50 | 2.24 | 0.65 | 0.84 |

In addition, increasing the number of noise regions and the number of variables tends to increase the asymmetry of the distribution, see Figure 9 and Tables 6 and 8. This illustrates the fact that the number of degrees of freedom is increasing, and that it is therefore possible to generate increasingly stochastic mechanisms, although their probability of being sampled remains low. On the contrary, increasing the cardinality of the endogenous variables seems to reduce the asymmetry of the distribution, which may seem surprising. In reality, the distribution flattens out at higher stochasticity levels, making it more symmetrical. Indeed, both the mean and the standard deviation increase.

This analysis also reveals a surprising result: The number of noise regions does not seem to increase the level of stochasticity, cf. Figure 9 and Table 8. Theoretically, the more noise regions, the higher the number of mappings defining a causal mechanism. By complementing this mixture, we could expect to obtain a higher level of stochasticity. Further analysis is therefore required here to clarify the effect of the noise region parameter on stochasticity.

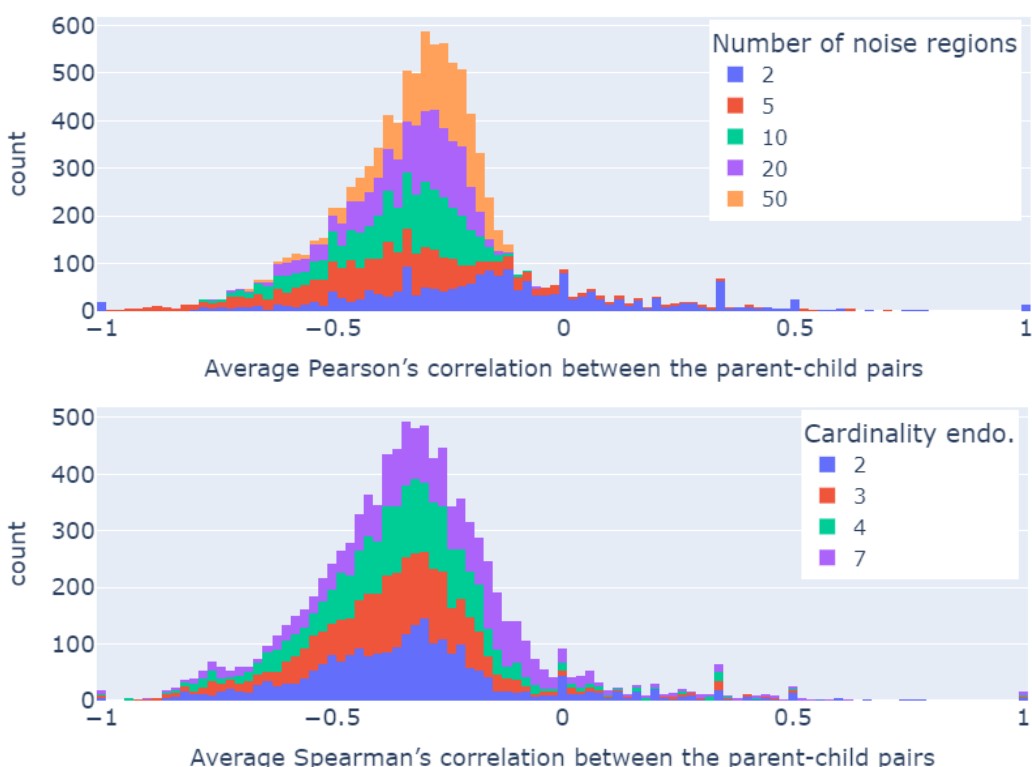

*Figure 10.* Stacked histograms of the average Pearson's and Spearman's correlation between the parent-child pairs of the generated SCMs. Mean, standard deviation, and skewness of the distributions can be found in Tables 9 and 10.

*Table 9.* Mean, standard deviation, and skewness of the distribution of the average Pearson's correlation between the parent-child pairs of the generated SCMs depending on their number of noise regions. The distribution is displayed in Figure 10.

| | Pearson's correlation | | |
|---|---|---|---|
| **Number of noise regions** | Mean | Std | Skewness |
| 2 | -0.15 | 0.30 | 0.54 |
| 5 | -0.38 | 0.22 | 0.60 |
| 10 | -0.36 | 0.13 | -0.73 |
| 20 | -0.34 | 0.11 | -0.68 |
| 50 | -0.30 | 0.10 | -0.80 |

*Table 10.* Mean, standard deviation, and skewness of the distribution of the average Spearman's correlation between the parent-child pairs of the generated SCMs depending on the cardinality of their endogenous variables. The distribution is displayed in Figure 10.

| | Pearson's correlation | | |
|---|---|---|---|
| **Cardinality** | Mean | Std | Skewness |
| 2 | -0.32 | 0.25 | 1.10 |
| 3 | -0.36 | 0.20 | 0.97 |
| 4 | -0.35 | 0.19 | 1.09 |
| 7 | -0.27 | 0.22 | 0.57 |

Third, the analysis of the levels of linearity and monotonicity (measured using Pearson and Spearman correlations) reveals that the sampled causal mechanisms are mostly neither linear nor monotonic, see Figure 10. Even if this result is to be expected, as the regional discrete mechanisms are discrete mappings without any notion of ordering, the fact that all the distributions are constituted of one peak on the negative side instead of two peaks, symmetric with respect to 0 is surprising. Hence, more investigation remains to be done to understand if our sampling algorithm tends to favor the generation of monotonically decreasing mechanisms.

One can also notice from Tables 9 and 10 that neither the cardinality of the endogenous variables nor the number of noise regions seems to affect the mean of the distributions, which is close to 0.35. In particular, the cardinality seems to have no effect on the distribution, while increasing the number of noise regions seems to increase the asymmetry of the distribution towards more linear mechanisms and decrease the standard deviation. Hence, we warn the users that choosing a high number of noise regions, hoping to be very diverse when generating mechanisms, might create the opposite effect over some metrics, as the distributions of Spearman's and Pearson's correlations seem to narrow down in this analysis.

*Table 11.* Percentage of sampled observational datasets with complete empirical joint support, grouped by the number of endogenous variables. Complete support means that every joint realization appears at least once in 10,000 samples.

| | Avg. min. proba. of the joint distribution | |
|---|---|---|
| **Number of variables** | 0 | >0 |
| 3 | 88.6 | 11.4 |
| 4 | 95.6 | 4.4 |
| 5 | 97.0 | 3.0 |

*Table 12.* Percentage of sampled observational datasets with complete empirical joint support, grouped by variable cardinality. Complete support means that every joint realization appears at least once in 10,000 samples.

| | Avg. min. proba. of the joint distribution | |
|---|---|---|
| **Cardinality** | 0 | >0 |
| 2 | 95.7 | 4.3 |
| 3 | 97.5 | 2.5 |
| 4 | 95.5 | 4.5 |
| 7 | 89.8 | 10.2 |

Finally, Tables 11 and 12 report an empirical support diagnostic for the sampled regional discrete SCMs. For each generated observational dataset of size 10,000, we compute whether every joint realization of the observed variables appears at least once. Under this conservative finite-sample criterion, all realizations are observed in approximately 6% of the generated datasets.

This diagnostic should not be interpreted as establishing population-level failure of positivity for 94% of the sampled SCMs. First, it is computed from finite samples: a realization with positive but small probability may simply not appear in a sample of size 10,000. Second, requiring empirical support over the full observed joint distribution is stronger than the overlap conditions needed for many specific causal queries, which depend on the treatment, covariates, and identification strategy under consideration. The support diagnostic therefore measures the prevalence of sparse empirical support under a demanding global criterion, rather than determining identifiability.

Nevertheless, the result is important for evaluation. In *SoIs* exhibiting sparse empirical support, causal estimators may need to extrapolate across poorly represented regions of the observational distribution. Consequently, performance in such regimes should be interpreted as reflecting robustness to limited support and estimator inductive bias, in addition to ordinary finite-sample estimation error. For evaluations intended to measure estimation accuracy within an identification-compatible regime, users should enforce the relevant support assumptions through the *SoI* specification or rejection sampling.

### G.3. Comparision to CausalNF synthetic SCMs used for evaluation

To illustrate the contribution in SCMs diversity that CausalProfiler can give to practitioners wishing to evaluate Causal ML methods, we compare the SCMs sampled in the previous Section with those used in the CausalNF work (Javaloy et al., 2023) for evaluation. We decided to first focus on the CausalNF synthetic SCMs because they have been reused by other papers (Sick & Dürr, 2025; Zhou et al., 2025) to evaluate new methods as if they were classical synthetic benchmarks for counterfactual evaluation.

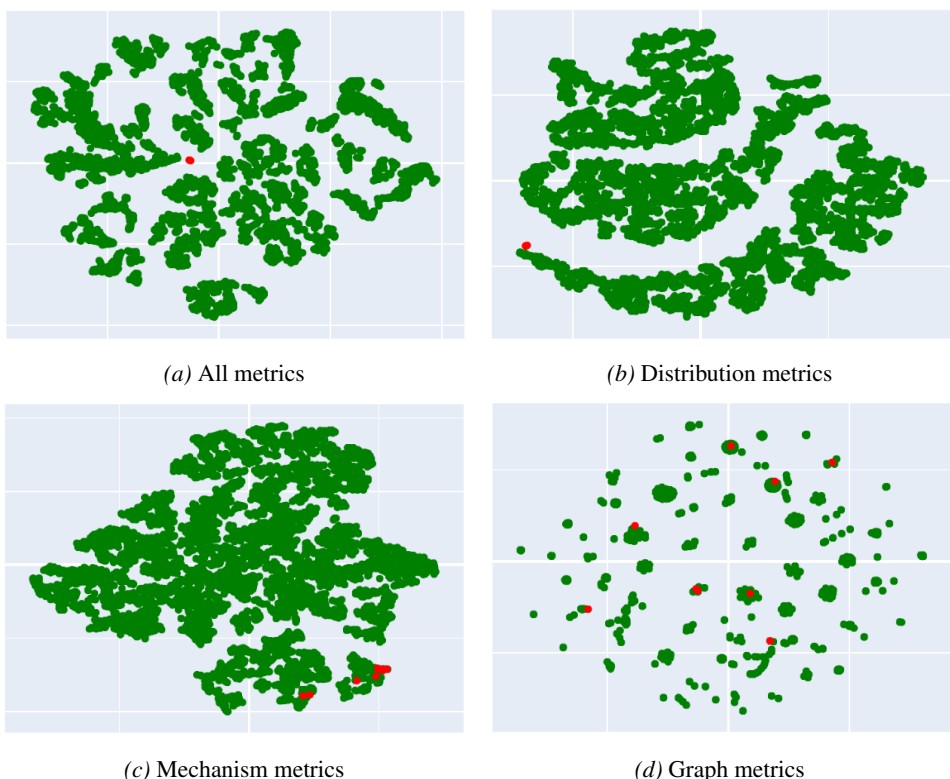

*(a)* All metrics

*(b)* Distribution metrics

*(c)* Mechanism metrics

*(d)* Graph metrics

*Figure 11.* Two-dimensional t-SNE plots representing our sampled SCMs (green) and the synthetic SCMs used for evaluation of CausalNF (red). The latter, less numerous, have been plotted in the foreground to highlight their distribution in relation to our SCMs. The SCMs are described using characterization metrics from the analysis module. (a) t-SNE plot using all metrics (b) t-SNE plot using distribution metrics only (c) t-SNE plot using mechanism metrics only (d) t-SNE plot using graph metrics only.

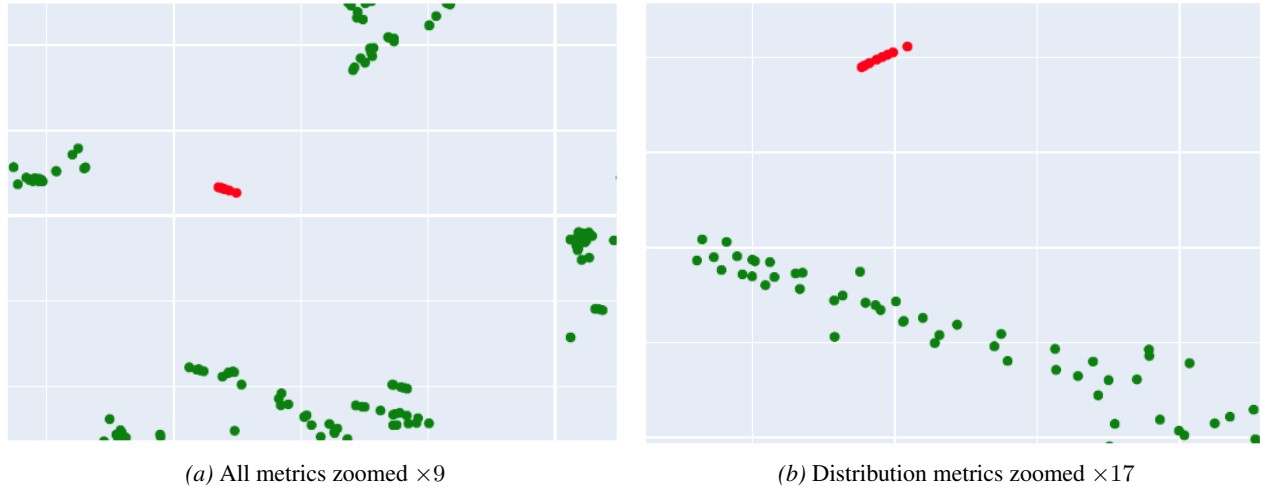

*(a)* All metrics zoomed ×9        *(b)* Distribution metrics zoomed ×17

*Figure 12.* t-SNE plots of Figures 11a and 11b zommed on CausalNF synthetic SCMs.

For this comparison, we reimplemented the synthetic SCMs of CausalNF using CausalProfiler, and applied all the metrics of the analysis module (cf. Appendix F). In this way, the CausalNF SCMs were processed in the same way as our SCMs. We then used these metrics to compare the two groups of SCMs. For the sake of having a fair comparison, not penalizing the fact that some assumptions were taken by the authors, we removed some metrics from the analysis: the hidden confounders and positivity metrics. Indeed, all CausalNF SCMs satisfy the causal sufficiency and strong positivity hypotheses, whereas, as presented in Appendix G.2, our SCMs do not by design. Finally, in order to obtain an easily interpretable visual result, we applied a two-dimensional t-SNE projection (Maaten & Hinton, 2008) to all these metrics and subgroups of metrics (Figure 11). Each t-SNE has been applied here with a perplexity of 30.

It can be seen that our SCMs are more diverse than those of CausalNF. Regarding graph metrics, it seems that CausalNF already has good diversity. The fact that we have greater support could mainly stem from the fact that we sampled a large number of SCMs. On the other hand, regarding distributions and mechanisms metrics, the increase in diversity is clear: The CausalNF SCMs are so similar compared to the total diversity that the dimension reduction projected them onto a confined space, cf. Figure 12. As a result, we can conclude that CausalProfiler can enable practitioners to evaluate Causal ML methods on a more diverse set of SCMs and naturally derive more conclusions.

**Remark on the patterns and empty spaces appearing for CausalProfiler's SCMs**: In Figure 11a, some patterns seem to appear for CausalProfiler's SCMs. Several reasons could explain this result. First, t-SNE preserves local neighborhoods but not global distances or density. As a result, non-neighboring points can be placed arbitrarily far apart, leading to large empty regions that do not correspond to true gaps in the underlying space (the embedding can create empty regions that did not exist before). Second, some configurations are infeasible or have a low-probability (e.g., too many edges for small sets of nodes).

**Remark on the distribution mismatch between CausalProfiler's SCMs and CausalNF**: CausalProfiler samples SCMs from a user-defined Space of Interest, whereas CausalNF consists of a small, fixed set of models. As a result, CausalNF SCMs may occupy regions of the metric space that are unlikely (or not targeted) under the current SoI, which explains why some of its points appear in areas with little or no CausalProfiler density. For instance, it is possible that over the sampled causal graphs, none of them correspond to the CausalNF SCMs ones.

### G.4. Comparision to bnlearn semi-sythetic graphical causal models

This section also illustrates the contribution CausalProfiler makes to SCMs' diversity by comparing them with other causal models used in the literature: CANCER and EARTHQUAKE from bnlearn (Scutari, 2019). Unlike the synthetic and continuous SCMs from CausalNF, CANCER and EARTHQUAKE are discrete causal graph models. The following analysis,

therefore, enriches the conclusions of the previous section.

CANCER and EARTHQUAKE were compared to the SCMs sampled by CausalProfiler in the same way as in the previous section: a two-dimensional t-SNE projection is applied to the metrics from the analysis module. The only difference here is that the mechanisms metrics cannot be computed on CANCER and EARTHQUAKE, as they are not proper SCMs, but graphical causal models. For the sake of having a fair comparison, we also excluded the hidden confounders metrics (as both bnlearn graphs are DAGs) but kept the positivity metrics for this analysis.

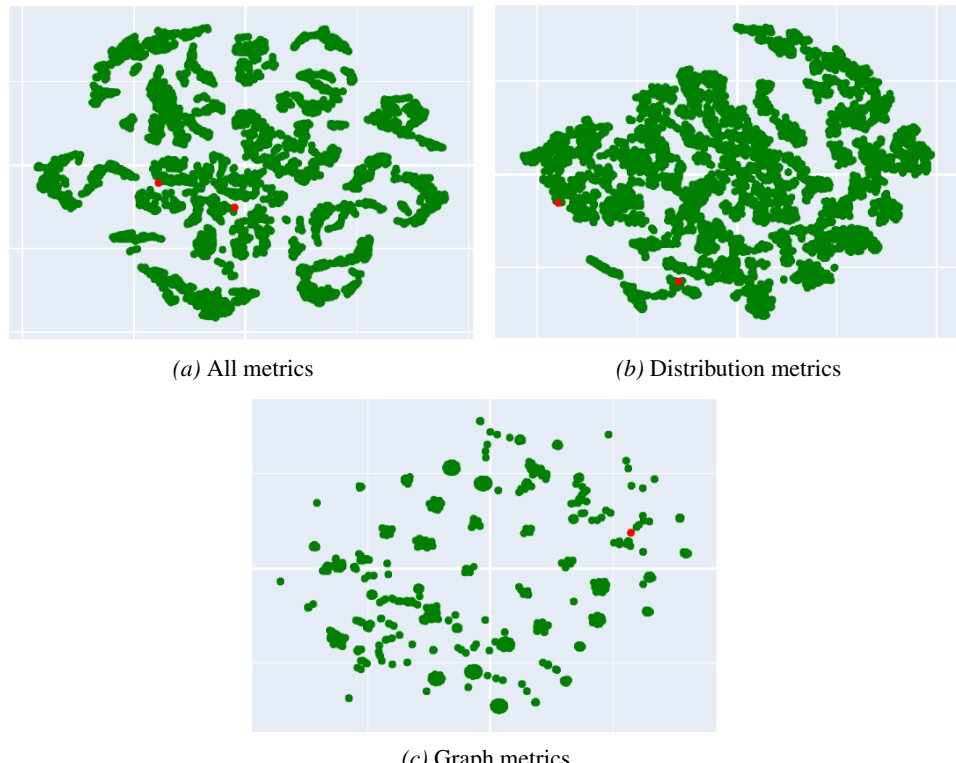

*(a)* All metrics          *(b)* Distribution metrics

*(c)* Graph metrics

*Figure 13.* Two-dimensional t-SNE plots representing our sampled SCMs (green) and the semi-synthetic graphical causal models CANCER and EARTHQUAKE from bnlearn (red). The SCMs are described using characterization metrics from the analysis module. (a) t-SNE plot using all metrics (b) t-SNE plot using distribution metrics only (c) t-SNE plot using graph metrics only. Mechanism metrics cannot be used as bnlearn models do not model mechanisms but rather distributions.

The results, presented in Figure 13, show that the two bnlearn datasets are not confined to a small region of the two-dimensional space. Instead, they fall within the bottom left region of the t-SNE plot, overlapping with some of our generated SCMs. Hence, the conclusion of this analysis is similar to the previous one: CausalProfiler can generate SCMs producing similar causal datasets to existing ones while also generating more diverse sets of SCMs.

**Why do we compare CausalProfiler SCMs to CausalNF synthetic SCMs and bnlearn datasets instead of datasets like IHDP, Twins, Syntren, or ACIC2016?**   Our comparison focuses on the diversity of underlying SCMs, which requires access to the full structural model (graph, mechanisms, and exogenous noise). These benchmarks do not expose their underlying SCMs needed to compute SCM-level metrics used in our analysis. Further, for this analysis we require datasets whose characteristics match those of the studied SoIs, in particular datasets with a small number of variables (3-5). This is why we include CausalNF and bnlearn networks, and exclude IHDP (Hill, 2011), Twins (Louizos et al., 2017), Syntren (den Bulcke et al., 2006), and ACIC2016 (Dorie et al., 2019), which contain substantially more variables. One might argue that we could simply sample higher-dimensional SCMs from CausalProfiler. While this is possible, computing the full set of assumption-analysis metrics (Appendix F) becomes computationally expensive as dimensionality and graph density increase; for example, Markov property checks and pairwise independence tests scale poorly with the number of variables. As a result, performing a detailed comparison with higher-dimensional datasets is not very tractable.

## H. Visual Overview of CausalProfiler's Sampling Strategy

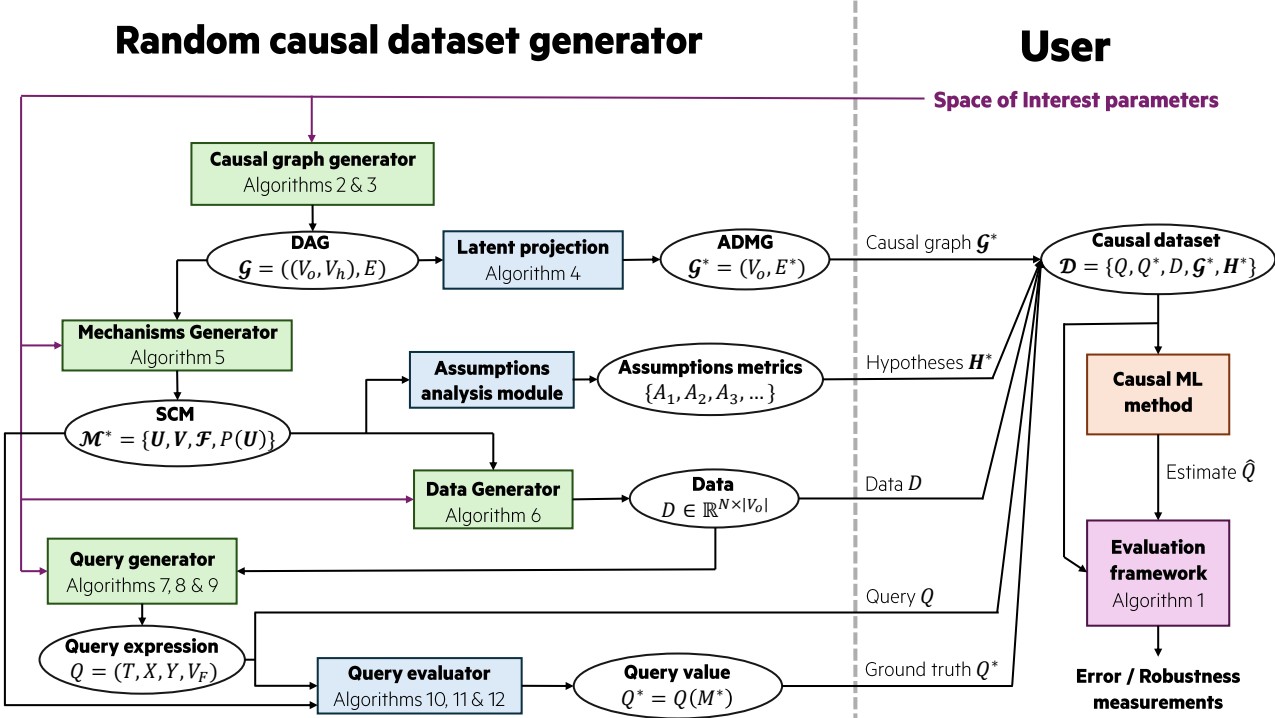

*Figure 14.* CausalProfiler structure. The left-hand side of the figure represents the code structure of the causal dataset generator. The right-hand side represents the user code. It illustrates how CausalProfiler can be used to evaluate a Causal ML method.

# I. Proof of Proposition 5.1 (Coverage)

This section presents the proof of Proposition 5.1 stating that: For a Space of Interest $\mathcal{S} = (\mathbb{M}, \mathbb{Q}, \mathbb{D})$, whose class of SCMs is a class of Regional Discrete SCMs with the maximum number of noise regions, any causal dataset $\mathcal{D} = \{Q, Q^\star, D, \mathcal{G}^\star, \mathbf{H}^\star\}$ has a strictly positive probability to be generated.

Firstly, let us note that:

- Stating that any query $Q$ can have any ground truth value $Q^\star$ given $\mathcal{S}$ is equivalent to saying that the class of considered SCMs, i.e., the class of Regional Discrete SCMs with the maximum number of noise regions, is $\mathcal{L}_3$-expressive with regards to the class of Markovian discrete SCMs (i.e., any $\mathcal{L}_3$-distribution of the class of Markovian discrete SCMs can be expressed with a Regional Discrete SCM).

- As the set of hypotheses $\mathbf{H}^\star$ can contain at most $\mathcal{L}_3$ conditions, if the class of considered SCMs is $\mathcal{L}_3$-expressive, then any set of hypotheses $\mathbf{H}^\star$ can be represented.

- If the class of considered SCMs is $\mathcal{L}_3$-expressive, then it is also $\mathcal{L}_1$-expressive, hence, $D$ can be sampled from any distribution.

As a result, our proof consists of showing that $P(Q, \mathcal{G}^\star | \mathcal{S}) > 0$ and that the class of Regional Discrete SCMs with the maximum number of noise regions, denoted $\mathbb{M}_{\mathrm{RD-SCM}, r=R_{\max}}$, is $\mathcal{L}_3$-expressive with regards to the class of Markovian discrete SCMs given an *SoI* $\mathcal{S}$ and a causal graph $\mathcal{G}$.

Let us consider a *SoI* $\mathcal{S} = (\mathbb{M}, \mathbb{Q}, \mathbb{D})$ with $\mathbb{M} \subseteq \mathbb{M}_{\mathrm{RD-SCM}, r=R_{\max}}$.

**Proving $P(\mathcal{G}^\star | \mathcal{S}) > 0$:**

$\mathcal{G}^\star$ is built through Algorithm 2 as the latent projection of a DAG $\mathcal{G} = \{(\mathbf{V}_H, \mathbf{V}_O), E\}$ over $\mathbf{V}_O$ where $\mathcal{G}$ is sampled using Algorithm 1. As a result, following the steps of Algorithms 1 and 2:

$$
\begin{aligned}
P(\mathcal{G}^\star | \mathcal{S}) &= P(\{(\mathbf{V}_H, \mathbf{V}_O), E\} | \mathcal{S}) \\
&= P(E | \mathbf{V}) P(\mathbf{V}_H, \mathbf{V}_O | \mathcal{S}) && \text{Edges are sampled independently of the} \\
& && \text{observability of the variables} \\
&= P(E | \mathbf{V}) P(\mathbf{V}_H, \mathbf{V}_O | |\mathbf{V}|) P(|\mathbf{V}|) && |\mathbf{V}| \text{ and } p_h \text{ are the only parameters influ-} \\
& && \text{encing the observability of the variables} \\
&= P(E | \mathbf{V}) P(\mathbf{V}_H, \mathbf{V}_O | |\mathbf{V}|) \frac{1}{N_{\max} - N_{\min}} && |\mathbf{V}| \sim \mathcal{U}[N_{\min}, N_{\max}] \\
&= P(E | \mathbf{V}) \frac{|\mathbf{V}_H|!}{|\mathbf{V}|!} \frac{1}{N_{\max} - N_{\min}} && \mathbf{V}_H \subseteq \mathbf{V} \text{ sampled without replacement} \\
&= \frac{|\mathbf{V}_H|!}{|\mathbf{V}|!(N_{\max} - N_{\min})} P(E | \mathbf{V})
\end{aligned}
$$

As $E = \{V_k \to V_i \mid V_k \in \mathbf{PA}(V_i), \forall V_i \in \mathbf{V}\}$ and the edges are sampled along the causal order $[1, N]$ with probability $p_{edge}$:

$$
\begin{aligned}
P(\mathcal{G}^\star | \mathcal{S}) &= \frac{|\mathbf{V}_H|!}{|\mathbf{V}|!(N_{\max} - N_{\min})} \prod_{i=1}^{N} P(\{V_k \to V_i \mid V_k \in \mathbf{PA}(V_i)\}) \\
&= \frac{|\mathbf{V}_H|!}{|\mathbf{V}|!(N_{\max} - N_{\min})} \prod_{i=1}^{N} p_{edge}^{|\mathbf{PA}(V_i)|} (1 - p_{edge})^{i-1-|\mathbf{PA}(V_i)|}
\end{aligned}
$$

Let us note that $p_{edge} = 0 \implies |\mathbf{PA}(V_i)| = 0$ and $p_{edge} = 1 \implies |\mathbf{PA}(V_i)| = i - 1$. As a result, $P(\mathcal{G}^\star | \mathcal{S}) > 0$.

**Proving that $\mathbb{M}_{\text{RD-SCM},r=R_{\max}}$ is $\mathcal{L}_3$-expressive with regards to the class of Markovian discrete SCMs:** Regional discrete SCMs are, by construction, Markovian Canonical SCMs (Zhang et al., 2022). Furthermore, if the number of noise regions is chosen to be large enough (typically set to its maximum value), any Markovian Canonical SCM can be represented using a Regional Discrete SCM[8]. Thus, applying Zhang et al. (2022) Theorem 2.4, we can assert that: for an arbitrary Markovian discrete SCM, there exists a Regional Discrete SCM such that they both have the same causal graph and the same $\mathcal{L}_3$-distribution. Consequently, the class of Regional Discrete SCMs is $\mathcal{L}_3$-expressive with respect to the class of Markovian discrete SCMs given the causal graph $\mathcal{G}$. Moreover, $P(\mathcal{G}) > 0$ for all $\mathcal{G}$ because $\prod_{i=1}^{N} p_{edge}^{|\boldsymbol{PA}(V_i)|}(1 - p_{edge})^{i-1-|\boldsymbol{PA}(V_i)|} > 0$ (cf. previous paragraph). Thus, more generally, the class of Regional Discrete SCMs sampled by our CausalProfiler is $\mathcal{L}_3$-expressive with respect to the class of Markovian SCMs.

**Proving $P(Q|\mathcal{G}^\star, \mathcal{S}) > 0$:** $Q$ is sampled given $\mathbb{Q}$, $D$ and $\mathcal{G}^\star$. Even though we currently only implement queries sampling for the classes $\mathbb{Q}_{\text{ATE}}$, $\mathbb{Q}_{\text{CATE}}$ and $\mathbb{Q}_{\text{Ctf-TE}}$ (cf. Appendix E and Algorithms 6, 7 and 8), we can generalize our proof to any other query class (e.g., CDE, NDE). We simply assume that these classes translate the set of constraints on the variables under consideration (e.g., conditioning variables have to be distinct from treatment variables or any other graphical constraints that can be checked with $\mathcal{G}^\star$) and express the probabilistic causal formula to be estimated. Once such a query class $\mathbb{Q}$ is defined, our method randomly samples variables from $\mathbf{V}_O$ in accordance with $\mathbb{Q}$ constraints and by sampling realizations from $D$. We showed in the previous paragraph that $\mathbb{M}_{\text{RD-SCM},r=R_{\max}}$ is $\mathcal{L}_3$-expressive implying that it is $\mathcal{L}_1$-expressive too. So, any realization can be present in $D$. As a result, for a given query class $\mathbb{Q}$, any $Q$ can be generated. Hence, $P(Q|\mathcal{G}^\star, \mathcal{S}) > 0$.

**Proving Proposition 5.1 by combining previous results:** We proved that $\mathbb{M}_{\text{RD-SCM},r=R_{\max}}$ is $\mathcal{L}_3$-expressive, hence any training set $D$, ground truth query $Q^\star$ and set of hypotheses $\mathbf{H}^\star$ can be generated given an *SoI* $\mathcal{S}$, a causal graph $\mathcal{G}$ and a causal query $Q$. In addition, $P(Q, \mathcal{G}^\star|\mathcal{S}) = P(Q|\mathcal{G}^\star, \mathcal{S})P(\mathcal{G}^\star|\mathcal{S})$ and we also prove that $P(Q|\mathcal{G}^\star, \mathcal{S}) > 0$ and $P(\mathcal{G}^\star|\mathcal{S}) > 0$. Hence, $P(Q, \mathcal{G}^\star|\mathcal{S}) > 0$. As a result, any causal dataset $\mathcal{D}$ has a strictly positive probability to be generated.

**Remark on continuous SCMs.** The universal approximation theorem (Hornik, 1991) states that NNs (with non-polynomial activation functions) are dense in the space of continuous functions, meaning that any continuous function can be approximated by a sequence of NNs converging to this function. However, this does not guarantee that they strictly cover the space of continuous functions. In particular, whenever the number of layers and neurons is finite, one can always build a continuous function too complex to be represented with this finite number of parameters. Hence, Proposition 5.1 cannot be extended to any class of continuous SCMs. However, it could potentially be adapted not to ask for strict coverage but rather density. We leave this question for future work.

---

[8]The distinction between $\mathbf{V}_O$ and $\mathbf{V}_H$ is of no importance for $\mathcal{L}_3$-expressiveness. $\mathbf{V}_O$ and $\mathbf{V}_H$ are only used to determine what will be visible to the user as benchmark.

# J. Verification Results

We design and run verification experiments targeting each level of the PCH.

All following experiments are done on discrete SCMs to reduce approximations. Indeed, distributions over continuous variables can only be approximated (e.g., using kernel methods) while discrete ones can be computed exactly. In addition, the experiments rely on conditional independence testing, which has been proven to be particularly difficult to use with continuous variables. Indeed, (Shah & Peters, 2020) proved that no conditional independence test with a continuous conditioning variable can have both a valid significance level and power.

## J.1. $\mathcal{L}_1$ verification

Consistency with $\mathcal{L}_1$ level of the PCH is tested through the verification that the Markov property holds on randomly sampled regional discrete SCMs. Below is a description of the experimental design choices made and the associated results.

### J.1.1. EXPERIMENT

For a given SCM $\mathcal{M} := (\mathbf{V}, \mathbf{U}, \mathcal{F}, P(\mathbf{U}))$, we check that the Markov property is satisfied by assessing whether there is a statistically significant amount of d-separations not leading to conditional independence in the entailed distribution.
To do so, we first enumerate the list of sets of variables $(\mathbf{A}, \mathbf{B}, \mathbf{C})$ in $\mathbf{V}$ corresponding to d-separations in $\mathcal{M}$'s causal graph $\mathcal{G}_{\mathcal{M}}$, ie $\mathbf{A} \perp\!\!\!\perp_{\mathcal{G}_{\mathcal{M}}} \mathbf{B} | \mathbf{C}$. Second, for each d-separated set $(\mathbf{A}, \mathbf{B}, \mathbf{C})$, we test whether $\mathbf{A} \perp\!\!\!\perp_{P_{\mathcal{M}}} \mathbf{B} | \mathbf{C}$ by sampling 50k data points from the entailed distribution $P_{\mathcal{M}}$.

In practice, enumerating all the d-separations can be very costly. Moreover, as the set of variables $\mathbf{C}$ increases, it becomes increasingly complicated to robustly test the conditional independence $\mathbf{A} \perp\!\!\!\perp_{P_{\mathcal{M}}} \mathbf{B} | \mathbf{C}$. Indeed, as the cardinality of $\mathbf{C}$ increases, so does the number of combinations of values for which to test independence between variables $\mathbf{A}$ and $\mathbf{B}$. Running the statistical test becomes costly, and the data volume required for robust independence test results increases exponentially. This is why we limit ourselves to listing the d-separated sets $(A, B, \mathbf{C})$ such that $A \in \mathbf{V}$, $B \in \mathbf{V} \backslash A$, and $C \in \mathbf{V} \cup \mathbf{V}^2 \cup \mathbf{V}^3$ by enumerating all the possible $(A, B, \mathbf{C})$ tuples, and testing whether they are d-separated in $\mathcal{G}_{\mathcal{M}}$.

As the sampled SCMs are regional discrete, the conditional independence $A \perp\!\!\!\perp_{P_{\mathcal{M}}} B | \mathbf{C}$ can be tested with Pearson's $\chi^2$ independence tests (Pearson, 1900). More precisely, $A$ and $B$ are considered independent conditionally to $\mathbf{C}$ if for all values $\mathbf{c}$ of $\mathbf{C}$, the $H_0$ hypothesis "$A$ and $B$ are independent" is not rejected. Since Pearson's $\chi^2$ test is based on the assumption that the number of samples is large, we decide to skip tests where the Koehler criterion (Koehler & Larntz, 1980) is not met. Based on empirical analyses, this criterion indicates whether the $\chi^2$ test is reliable depending on the number of samples considered. In addition, as we conduct tests for each observed value $c$, we need to control for the expected proportion of false positives (represented by the Type I error of the test). To do so, we apply the Benjamini-Hochberg correction (Benjamini & Hochberg, 1995).

For each *SoI*, defined by the Cartesian product of the following parameters, we sample 5 SCMs:

- **Number of endogenous variables**: $\{4, 5, 6\}$

- **Expected edge probability**: $\{0.1, 0.4\}$

- **Proportion of unobserved endogenous variables**: set to 0 because the Markov property only hold for Markovian SCMs

- **Number of noise regions**: $\{5, 10\}$

- **Cardinality of endogenous variables**: $\{2, 3, 10\}$

- **Distribution of exogenous variables**: set to $\mathcal{U}[0, 1]$

- **Number of data points**: 50000

### J.1.2. RESULTS

The experimental results are summarized in Table 13, where it can be seen that 5.4% of the conditional independence tests failed. Despite the use of the Koehler criterion and Benjamini-Hochberg correction, some tests can still be

*Table 13.* Conditional independence tests based on $\chi^2$ independence tests to assess compliance of sampled SCMs with the Markov property. Results are expressed as a percentage of the total of each test type for each conditioning set size. The number of tests is also shown in brackets.

| Conditioning set size | $A \perp\!\!\!\perp_{P_{\mathcal{M}}} B \vert \mathbf{C}$ tests | | | | $\chi^2$ independence tests | | | |
|---|---|---|---|---|---|---|---|---|
| | Total | Pass | Fail | Skip | Total | Pass | Fail | Skip |
| $\vert\mathbf{C}\vert = 1$ | 100 | 91.76 | 4.94 | 3.3 | 100 | 85.4 | 1.43 | 13.17 |
| | (2 391) | (2 194) | (118) | (79) | (9 130) | (7 797) | (131) | (1 202) |
| $\vert\mathbf{C}\vert = 2$ | 100 | 91.16 | 5.63 | 3.22 | 100 | 45.2 | 0.33 | 54.46 |
| | (2 986) | (2 722) | (168) | (96) | (53 040) | (23 976) | (177) | (28 887) |
| $\vert\mathbf{C}\vert = 3$ | 100 | 91.08 | 5.67 | 3.25 | 100 | 18.49 | 0.07 | 81.43 |
| | (1 693) | (1 542) | (96) | (55) | (145 320) | (26 874) | (106) | (118 340) |
| TOTAL | 100 | 91.34 | 5.40 | 3.25 | 100 | 28.26 | 0.2 | 71.54 |
| | (7 070) | (6 458) | (382) | (230) | (207 490) | (58 647) | (414) | (148 429) |

rejected due to the random nature of finite data sampling, which can produce slight artificial correlations in the data. Moreover, on closer inspection, the majority of the failed tests (at least 350 out of 382)[9] are unsuccessful because of a single failed $\chi^2$ independence test. This reinforces our previous argument about the random nature of finite data sampling.

One can also notice that the number of skipped $\chi^2$ independence tests increases with the size of the conditioning set. Such behavior is to be expected, since the number of realizations of the conditioning set increases exponentially with its cardinality, while the number of observations sampled to perform the independence tests remains constant. As a result, there are fewer and fewer observations available to perform each $\chi^2$ test. In contrast, the number of fully skipped conditional independence tests remains constant. This means that the $\chi^2$ skipped tests are relatively homogeneously distributed across all the conditional independence tests.

Someone might argue that the number of sampled observations should simply be automatically computed to verify the Koehler criterion. However, in general, such a calculation is complicated, if not impossible, to automate, as causal mechanisms are randomly sampled. As a result, all kinds of observational distributions can be induced with potentially very low probability realizations, for which the Koehler criterion could never be validated because the number of data to be sampled would be too large.

To conclude, these results are sufficient to conclude that the Markov property is empirically verified by the sampled SCMs.

## J.2. $\mathcal{L}_2$ verification

Consistency with $\mathcal{L}_2$ level of the PCH is tested through the verification that the Do-calculus rules hold on randomly sampled regional discrete SCMs. Below is a description of the experimental design choices made (Appendix J.2.1) and the associated results (Appendix J.2.2).

---

[9]Indeed, there is a total of 414 $\chi^2$ tests that failed corresponding to 382 failed conditional independence tests. It mean that, at most 32(=414-382) conditional independence tests can have more than one failed $\chi^2$ independence test.

### J.2.1. EXPERIMENT

> **Definition J.1. Do-Calculus rules** (Pearl, 2009)
> Given an SCM $\mathcal{M} \coloneqq (\mathbf{V}, \mathbf{U}, \mathcal{F}, P(\mathbf{U}))$ whose causal graph $\mathcal{G}$ is a DAG, and disjoint subsets $\mathbf{X}, \mathbf{Y}, \mathbf{Z}$, and $\mathbf{W}$ of $\mathbf{V}$, the rules of the **Do-Calculus** are defined as follows:
>
> 1. **Insertion/deletion of observation**: if $\mathbf{Y}$ and $\mathbf{Z}$ are d-separated by $\mathbf{X} \cup \mathbf{W}$ in $\mathcal{G}_{\overline{\mathbf{X}}}$, then $P(\mathbf{Y}|do(\mathbf{X} = \mathbf{x}), \mathbf{W}, \mathbf{Z}) = P(\mathbf{Y}|do(\mathbf{X} = \mathbf{x}), \mathbf{W})$
>
> 2. **Action/observation exchange**: if $\mathbf{Y}$ and $\mathbf{Z}$ are d-separated by $\mathbf{X} \cup \mathbf{W}$ in $\mathcal{G}_{\overline{\mathbf{X}}, \underline{\mathbf{Z}}}$, then $P(\mathbf{Y}|do(\mathbf{X} = \mathbf{x}), do(\mathbf{Z} = \mathbf{z}), \mathbf{W}) = P(\mathbf{Y}|do(\mathbf{X} = \mathbf{x}), \mathbf{Z}, \mathbf{W})$
>
> 3. **Insertion/deletion of action**: if $\mathbf{Y}$ and $\mathbf{Z}$ are d-separated by $\mathbf{X} \cup \mathbf{W}$ in $\mathcal{G}_{\overline{\mathbf{X}}, \overline{\mathbf{Z}(\mathbf{W})}}$, then $P(\mathbf{Y}|do(\mathbf{X} = \mathbf{x}), do(\mathbf{Z} = \mathbf{z}), \mathbf{W}) = P(\mathbf{Y}|do(\mathbf{X} = \mathbf{x}), \mathbf{W})$
>
> where $\mathcal{G}_{\overline{\mathbf{X}}}$ (resp. $\mathcal{G}_{\underline{\mathbf{X}}}$) represents the graph $\mathcal{G}$ where the incoming edges in (resp. outgoing edges from) $\mathbf{X}$ have been removed and $\mathbf{Z}(\mathbf{W})$ is the subset of nodes in $\mathbf{Z}$ that are not ancestors of any node in $\mathbf{W}$ in $\mathcal{G}_{\overline{\mathbf{X}}}$

For a given SCM, we check each rule by first enumerating the sets of d-separated variables of interest. Second, for each d-separated set, we test whether the distributions are statistically significantly similar by sampling 50k data points from the intervened SCMs and testing whether they are drawn from the same distribution.

For the same computational cost reasons as for $\mathcal{L}_1$ verification, we consider only univariate sets of variables $X, Y, Z$, and $W$. In addition, the studied SCMs are sampled from the same *SoIs* as defined in the $\mathcal{L}_1$-verification experiment (Appendix J.1.1). Finally, to assess whether two conditional distributions are identical, we used Pearson's $\chi^2$ goodness of fit tests (Pearson, 1900). As done in Section J.1, we also use the Koehler criterion (Koehler & Larntz, 1980) and the Benjamini-Hochberg correction (Benjamini & Hochberg, 1995).

For each *SoI*, defined by the Cartesian product of the following parameters, we sample 2 SCMs:

- **Number of endogenous variables**: $\{4, 5, 6\}$

- **Expected edge probability**: $\{0.1, 0.4\}$

- **Proportion of unobserved endogenous variables**: set to 0 because the Markov property only hold for Markovian SCMs

- **Number of noise regions**: $\{5, 100\}$

- **Cardinality of endogenous variables**: $\{2, 5\}$

- **Distribution of exogenous variables**: set to $\mathcal{U}[0, 1]$

- **Number of data points**: $50000$

Compared to the previous experiment (Appendix J.1.1), we reduce the number of sampled SCMs because comparing distributions two by two is more computationally expensive than conditional independence tests.

### J.2.2. RESULTS

The experimental results are summarized in Table 14 where it can be seen that they are very similar to the $\mathcal{L}_1$ verification ones: roughly $6\%$ of the conditional goodness of fit tests were not validated, some tests are rejected due to the random nature of finite data sampling but the majority them (at least 570 out of 755) are unsuccessful because of a single failed $\chi^2$ goodness of fit test.

One can also notice that the percentage of skipped $\chi^2$ goodness of fit tests is similar for rules 1 and 3 but increases by roughly $50\%$ for rule 2. Such behavior is to be expected as rule 2 is the only rule to have conditioning sets of size 3 on both sides of the equality. However, the number of skipped tests remains low, with a maximum of $16\%$.

*Table 14.* Conditional independence tests based on $\chi^2$ goodness of fit tests to assess compliance of sampled SCMs with the Do-Calculus rules. Results are expressed as a percentage of the total of each test type for each conditioning set size. The number of tests is also shown in brackets.

| Do-Calculus Rule | Cond. goodness of fit | | | | $\chi^2$ goodness of fit | | | |
|---|---|---|---|---|---|---|---|---|
| | Total | Pass | Fail | Skip | Total | Pass | Fail | Skip |
| **Rule 1** Insertion/deletion of observation | 100 (3 378) | 96.15 (3 248) | 3.85 (130) | 0 (0) | 100 (171 092) | 88.84 (152 004) | 0.1 (172) | 11.06 (18 916) |
| **Rule 2** Action/observation exchange | 100 (5 065) | 94.04 (4 763) | 5.96 (302) | 0 (0) | 100 (259 509) | 83.84 (217 578) | 0.09 (241) | 16.06 (41 690) |
| **Rule 3** Insertion/deletion of action | 100 (5 169) | 93.75 (4 846) | 6.25 (323) | 0 (0) | 100 (282 184) | 89.21 (251 731) | 0.06 (157) | 10.74 (30 296) |
| **TOTAL** | 100 (13 612) | 94.45 (12 857) | 5.55 (755) | 0 (0) | 100 (712 785) | 87.17 (621 313) | 0.08 (570) | 12.75 (90 902) |

As a result, we estimate that these results are sufficient to conclude that the Do-calculus rules are respected by the sampled SCMs.

## J.3. $\mathcal{L}_3$ verification

Consistency with $\mathcal{L}_3$ level of the PCH is tested through the verification that the axiomatic characterization of structural counterfactuals holds on randomly sampled regional discrete SCMs. Below is a description of the experimental design choices made (Appendix J.3.1) and the associated results (Appendix J.3.2).

---

**Definition J.2. Axiomatic characterization of structural counterfactuals** (Pearl, 2009)
Given an SCM $\mathcal{M} := \{\mathbf{V}, \mathbf{U}, \mathcal{F}, P(\mathbf{U})\}$ whose causal graph $\mathcal{G}$ is a DAG, the **axioms of structural counterfactuals** are defined as follows:

1. **Composition**: For any sets of endogenous variables $\mathbf{X}, \mathbf{Y}$, and $\mathbf{W}$ in $\mathbf{V}$ and any realization $\mathbf{u}$ of $\mathbf{U}$, if $\mathbf{W}_{do(\mathbf{X}=\mathbf{x})}(\mathbf{u}) = \mathbf{w}$ then $\mathbf{Y}_{do(\mathbf{X}=\mathbf{x}),do(\mathbf{W}=\mathbf{w})}(\mathbf{u}) = \mathbf{Y}_{do(\mathbf{X}=\mathbf{x})}(\mathbf{u})$

2. **Effectiveness**: For any disjoint sets of endogenous variables $\mathbf{X}$, and $\mathbf{W}$ in $\mathbf{V}$ and any realization $\mathbf{u}$ of $\mathbf{U}$, $\mathbf{X}_{do(\mathbf{X}=\mathbf{x}),do(\mathbf{W}=\mathbf{w})}(\mathbf{u}) = \mathbf{x}$

3. **Reversibility**: For any two distinct variables $Y$ and $W$ and any sets of other variables $\mathbf{X}$ in $\mathbf{V}$ and any realization $\mathbf{u}$ of $\mathbf{U}$, if $Y_{do(\mathbf{X}=\mathbf{x}),do(W=w)}(\mathbf{u}) = y$ and $W_{do(\mathbf{X}=\mathbf{x}),do(Y=y)}(\mathbf{u}) = w$ then $Y_{do(\mathbf{X}=\mathbf{x})}(\mathbf{u}) = y$

---

Note that we do not write $P(\mathbf{W}_{do(\mathbf{X}=\mathbf{x})}|\mathbf{U})$ but rather $\mathbf{W}_{do(\mathbf{X}=\mathbf{x})}(\mathbf{u})$ as it is a deterministic expression. Indeed, if $\mathbf{U}$ is fixed, there is no stochastically anymore, so we no longer need to reason in distributions but rather in functional forms.

### J.3.1. EXPERIMENT

For a given SCM, using Definition J.1 notations, we check that:

1. The **Composition** axiom is satisfied by assessing whether $\mathbf{W}_{do(\mathbf{X}=\mathbf{x})}(\mathbf{u}) = \mathbf{w}$ implies $\mathbf{Y}_{do(\mathbf{X}=\mathbf{x}),do(\mathbf{W}=\mathbf{w})}(\mathbf{u}) = \mathbf{Y}_{do(\mathbf{X}=\mathbf{x})}(\mathbf{u})$ for any sets of endogenous variables $\mathbf{X}, \mathbf{Y}$, and $\mathbf{W}$ in $\mathbf{V}$ and any realization $\mathbf{u}$ of $\mathbf{U}$

2. The **Effectiveness** axiom is satisfied by assessing whether $\mathbf{X}_{do(\mathbf{X}=\mathbf{x}),do(\mathbf{W}=\mathbf{w})}(\mathbf{u}) = \mathbf{x}$ for any sets of endogenous variables $\mathbf{X}$, and $\mathbf{W}$ in $\mathbf{V}$ and any realization $\mathbf{u}$ of $\mathbf{U}$

3. The **Reversibility** axiom is satisfied by assessing whether $Y_{do(\mathbf{X}=\mathbf{x}),do(W=w)}(\mathbf{u}) = y$ and $W_{do(\mathbf{X}=\mathbf{x}),do(Y=y)}(\mathbf{u}) = w$ implies $Y_{do(\mathbf{X}=\mathbf{x})}(\mathbf{u}) = y$ for any two (distinct) variables $Y$ and $W$ and any sets of variables $\mathbf{X}$ in $\mathbf{V}$ and any realization $\mathbf{u}$ of $\mathbf{U}$

For each *SoI*, defined by the Cartesian product of the following parameters, we sample 5 SCMs:

- **Number of endogenous variables**: $\{3, 5, 10\}$

- **Expected edge probability**: $\{0.1, 0.5, 0.7\}$

- **Proportion of unobserved endogenous variables**: set to $0$ because the Markov property only hold for Markovian SCMs

- **Number of noise regions**: $\{3, 5, 10\}$

- **Cardinality of endogenous variables**: $\{2, 5, 7\}$

- **Distribution of exogenous variables**: set to $\mathcal{U}[0, 1]$

- **Number of data points**: $50000$

For each SCM, instead of enumerating all the possible four sets of variables $\mathbf{X}, \mathbf{Y}$ and $\mathbf{W}$, we sample a partition of three elements of a randomly sampled subset of $\mathbf{V}$ of a size randomly picked in $[3, |\mathbf{V}|]$. This sampling strategy enables us to make sure the three sets are disjoint and of randomly varying size. In addition, for each four sets, we sample 50k realizations of $\mathbf{U}$.

Let us note that the axioms now correspond to exact realizations and not equal probabilities. As a result, we expect no failure as no approximation is made in this experiment.

### J.3.2. RESULTS

As expected, all the tested equalities are verified in our experiments. We can, therefore, consider that the SCMs created by our generator allows the estimation of any structural counterfactual queries.

# K. Extended Experimental Results

This appendix complements Section 6 with extended setup details and results. We first provide further details for Experiments 1 and 2, inlcuding the Algorithm 12 describing our evaluation protocol. We then include an additional experiment on ATE estimation under hidden confounding, and then an evaluation of runtime scalability on larger graphs.

---

**Algorithm 12** Evaluation process for causal machine learning methods

---

 1: **Input:** List of Spaces of Interest $SoIs$, list of seeds $seeds$ number of examples per SCM $num\_examples$
 2: **Initialize:** $method \leftarrow$ CausalMLMethod()
 3: **for** each $SoI$ in $SoIs$ **do**
 4:     **for** each $seed$ in $seeds$ **do**
 5:         setGlobalSeed(seed)
 6:         **for** each $examples$ in $num\_examples$ **do**
 7:             Generate samples, queries, and targets from the profiler
 8:             Get estimates using the $method$ on the generated samples and queries
 9:             Calculate (and store) error by comparing estimates with targets
10:         **end for**
11:         Compute performance statistics for seed
12:     **end for**
13:     Compute performance statistics for $SoI$
14: **end for**
15: **Output:** Final summary with evaluation results

---

## K.1. Experiment 1: Additional Information

Table 15 details the *SoI* used in our experiments, Table 16 reports extended performance metrics complementing Table 1, and Figure 15 shows box plots of ATE estimation errors.

Parameters not explicitly listed for a given *SoI* are set to their default values as per the benchmark configuration. Neural Networks for our experiments have two 8-neuron layers and use ReLU activation. Unless otherwise specified, we use 1000 samples per SCM in our experiments. This value was chosen as a stable default for these *SoIs* after testing several dataset sizes. More precisely, after testing the stability of the methods we evaluate (i.e., CausalNF, DCM, NCM, VACA) over the following dataset sizes, 50, 100, 200, 1000, and 2000, we found that 1000 samples was the smallest dataset size not drastically degrading the performance of the methods. This is why we decided to take this value as default for our experiments. We only vary it explicitly when studying the effect of limited data (e.g., in NN-Large-LowData).

*Table 15.* Specification of each *SoI* used in the general experiments. $N$ denotes the sampled number of nodes.

| Name | Linear-Medium |
|---|---|
| # Nodes | 15-20 |
| Mechanism | Linear |
| Expected Edges | $2 \times N$ |
| Variable Type | Continuous |
| Samples | 1000 |
| Query Type | ATE |
| Seeds | [10, 11, 12, 13, 14] |

| Name | NN-Medium |
|---|---|
| # Nodes | 15-20 |
| Mechanism | NN |
| Expected Edges | $2 \times N$ |
| Variable Type | Continuous |
| Samples | 1000 |
| Query Type | ATE |
| Seeds | [10, 11, 12, 13, 14] |

| Name | NN-Large |
|---|---|
| # Nodes | 20-25 |
| Mechanism | NN |
| Expected Edges | $2 \times N$ |
| Variable Type | Continuous |
| Samples | 1000 |
| Query Type | ATE |
| Seeds | [10, 11, 12, 13, 14] |

| Name | NN-Large-LowData |
|---|---|
| # Nodes | 20-25 |
| Mechanism | NN |
| Expected Edges | $2 \times N$ |
| Variable Type | Continuous |
| Samples | 50 |
| Query Type | ATE |
| Seeds | [10, 11, 12, 13, 14] |

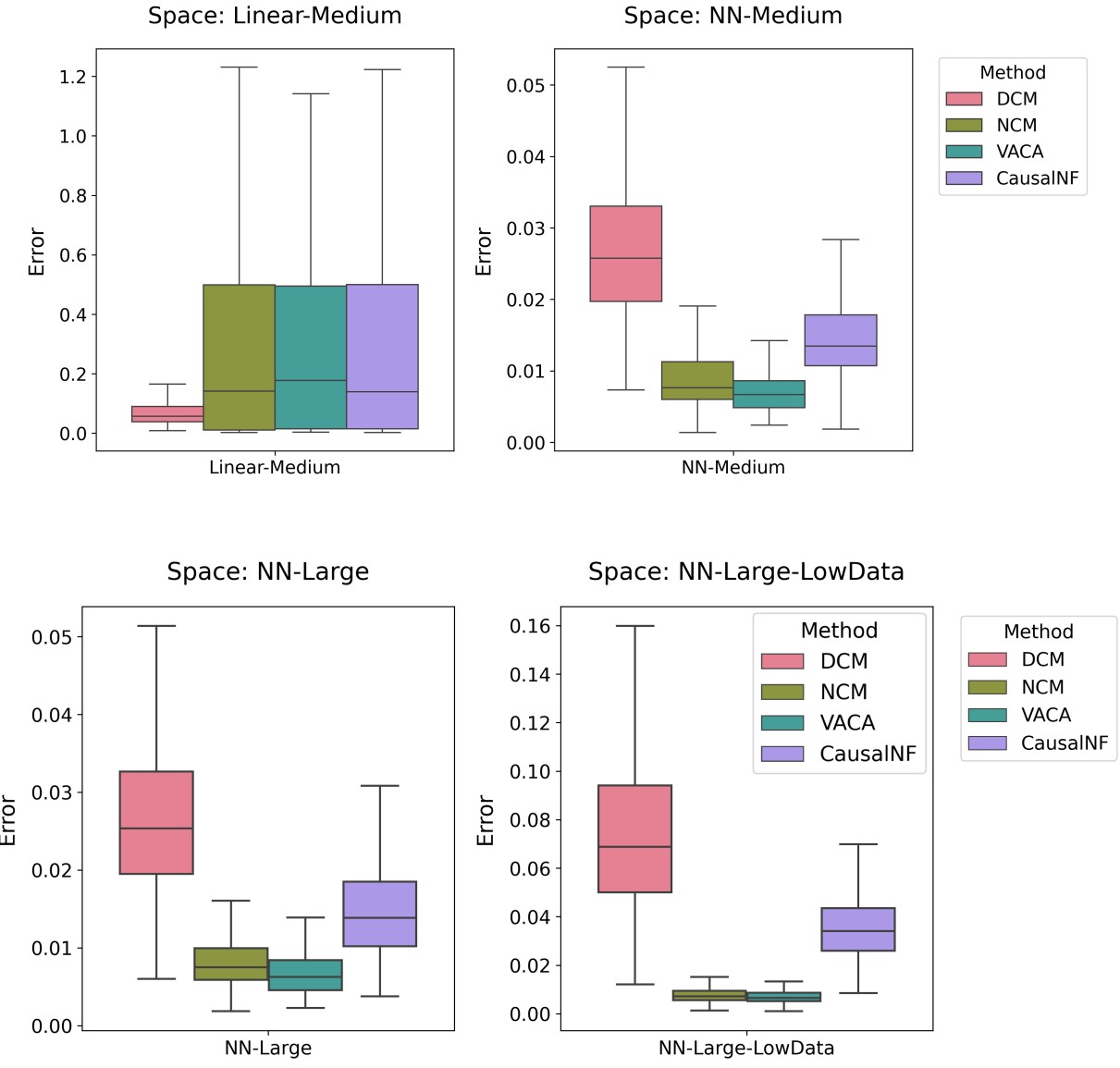

*Figure 15.* Box plots showing ATE estimation errors across different *SoIs*.

*Table 16.* Additional performance metrics of CausalNF, DCM, NCM, and VACA on the general experiments.

| Space | Method | Min Error | Total Fail | Runtime Mean | Runtime Std |
|---|---|---|---|---|---|
| Linear-Medium | CausalNF | 0.0024 | 0 | 27.58 s | 18.33 s |
| | DCM | 0.0086 | 0 | 33.08 s | 9.71 s |
| | NCM | 0.0024 | 0 | 14.77 s | 1.42 s |
| | VACA | 0.0038 | 1335 | 11.69 s | 4.54 s |
| NN-Medium | CausalNF | 0.0019 | 0 | 21.47 s | 19.52 s |
| | DCM | 0.0073 | 0 | 31.79 s | 10.62 s |
| | NCM | 0.0014 | 0 | 14.65 s | 1.43 s |
| | VACA | 0.0024 | 125 | 12.13 s | 4.41 s |
| NN-Large | CausalNF | 0.0038 | 0 | 30.23 s | 25.33 s |
| | DCM | 0.0060 | 0 | 38.33 s | 14.02 s |
| | NCM | 0.0018 | 0 | 18.90 s | 1.38 s |
| | VACA | 0.0023 | 290 | 12.88 s | 4.31 s |
| NN-Large-LowData | CausalNF | 0.0086 | 0 | 44.28 s | 17.10 s |
| | DCM | 0.0121 | 0 | 4.82 s | 1.34 s |
| | NCM | 0.0013 | 0 | 0.81 s | 0.11 s |
| | VACA | 0.0010 | 0 | 10.43 s | 4.59 s |

## K.2. Experiment 2: Additional Information

### K.2.1. SPACES OF INTEREST

Table 17 specifies the three discrete *SoIs* considered in this experiment. All settings use tabular discrete mechanisms, 500 observational samples per sampled SCM, and Ctf-TE queries. Parameters not explicitly listed for a given *SoI* are set to their default values as per the benchmark configuration.

*Table 17.* Specification of the Spaces of Interest used for evaluating discrete SCMs with Ctf-TE queries. $N$ denotes the sampled number of nodes.

| Name | D2-Reject |
|---|---|
| # Nodes | 10–15 |
| # Categories | 2 |
| Mechanism | Tabular |
| Sampling Strategy | Rejection |
| Edges | $N$ |
| Samples | 500 |
| Query Type | Ctf-TE |
| Seeds | [1, 2, 3, 4, 5] |

| Name | D4-Unbias |
|---|---|
| # Nodes | 10–15 |
| # Categories | 4 |
| Mechanism | Tabular |
| Sampling Strategy | Random |
| Edges | $N$ |
| Samples | 500 |
| Query Type | Ctf-TE |
| Seeds | [1, 2, 3, 4, 5] |

| Name | D2L-Unbias |
|---|---|
| # Nodes | 20–30 |
| # Categories | 2 |
| Mechanism | Tabular |
| Sampling Strategy | Random |
| Edges | $N$ |
| Samples | 500 |
| Query Type | Ctf-TE |
| Seeds | [1, 2, 3, 4, 5] |

### K.2.2. QUERY-GENERATION PROTOCOL

We restrict the types of queries we consider because a randomly sampled Ctf-TE query in a discrete SCM may be uninformative for either of two reasons. First, most ground-truth effects will be zero, which may favor estimators that simply output values close to zero. Second, a query may specify a partial factual condition $V_F = v_F$ for which no compatible complete factual instance occurs in the sampled observational dataset. Since both evaluated methods (CausalNF and DCM) perform abduction from complete factual instances, such a query cannot be instantiated under the common empirical-factual protocol used here.

We therefore retain a query only if it has a non-zero ground-truth effect and the observational dataset contains at least one complete factual instance satisfying the conditioning event $V_F = v_F$.

### K.2.3. DISCRETE COUNTERFACTUAL EVALUATION

Both evaluated methods require an adaptation for computing discrete Ctf-TE probabilities from their generated outputs.

**CausalNF.** CausalNF represents discrete variables through dequantization: a discrete category $c$ is associated with a continuous value in the interval $[c, c+1)$. For factual abduction, we select complete observational instances satisfying the discrete condition $V_F = v_F$ and provide their corresponding dequantized representations to the model. For a queried discrete outcome value $Y = y$, the counterfactual probability is recovered by binning the continuous counterfactual output.

**DCM.** When trained on integer-coded categorical observations, DCM may return continuous decoded counterfactual values for the target variable. We apply a minimal output-level adapter: before computing the queried discrete outcome probability, each final counterfactual target output is mapped to its nearest valid category for that variable.

### K.2.4. IMPLEMENTATION DETAILS

For CausalNF, we use Neural Spline Flows and normalize the data prior to training and inference.

## K.3. Experiment 3: ATE Estimation under Hidden Confounding

In this experiment, we demonstrate how our framework can be used to evaluate methods in the presence of latent confounders — a common challenge in real-world causal inference. A key goal here is not only to confirm theoretical limitations but to investigate how quickly and severely performance degrades when assumptions are violated. While theory can tell us whether identification holds, it is often agnostic to the *degree* of failure. See Table 19 for a summary of results, Table 20 for a few additional performance metrics, and Figure 16 for a boxplot of ATE estimation errors over the different *SoI*.

We focus on two linear SCM settings:

- **Linear-No-Hidden:** Linear SCMs with 10-15 nodes and full observability (no hidden confounders), using 1000 data points per SCM.

- **Linear-60-Hidden:** Same setup as above, but with 60% of the variables unobserved (hidden).

We provide more details about the *SoI* used in our experiments in Table 18. Parameters not explicitly listed for a given *SoI* are set to their default values as per the benchmark configuration.

*Table 18.* Specification of the *SoIs* used to evaluate performance under hidden confounding. $N$ denotes the sampled number of nodes.

| Name | Linear-No-Hidden |
| --- | --- |
| # Nodes | 10-15 |
| Mechanism | Linear |
| Expected Edges | $2 \times N$ |
| Variable Type | Continuous |
| Prop. Hidden Nodes | 0% |
| Samples | 1000 |
| Query Type | ATE |
| Seeds | [42, 43, 44, 45, 46] |

| Name | Linear-60-Hidden |
| --- | --- |
| # Nodes | 10-15 |
| Mechanism | Linear |
| Expected Edges | $2 \times N$ |
| Variable Type | Continuous |
| Prop. Hidden Nodes | 60% |
| Samples | 1000 |
| Query Type | ATE |
| Seeds | [42, 43, 44, 45, 46] |

**Setup.**   We evaluate three methods: CausalNF (Javaloy et al., 2023), DCM (Chao et al., 2024), and DeCaFlow (Almodóvar et al., 2026). The first two methods assume causal sufficiency, and therefore cannot, in theory, handle hidden confounding. DeCaFlow, in contrast, is explicitly designed for this setting but requires access to the full causal graph (including hidden variables) and does not run when all variables are observed. Thus, we include it only in the hidden confounding *SoI*.

**Results (Linear-No-Hidden).**   As expected, both CausalNF and DCM perform well when all variables are observed. DCM achieves lower mean error (0.0845) and standard deviation (0.1515), with a maximum error of 2.89. The upper whisker of DCM's box plot lies below the median of CausalNF, indicating consistent superior performance. These results serve as a reference point for comparison when introducing hidden variables.

**Results (Linear-60-Hidden).**   With 60% of variables hidden, method performance degrades significantly. DeCaFlow performs reliably, with an error mean of 0.3405 and low variance. In contrast, CausalNF—despite a box plot that visually appears well-behaved—has a massive error mean of $2.67 \times 10^{12}$ and a maximum error exceeding $10^{15}$. This is due to a small subset of SCMs producing extremely large errors (14 with error $> 1000$), illustrating that, when assumptions are violated, error can become arbitrarily large. While DCM does not show such instability on this particular sample, its theoretical limitations under hidden confounding still hold — the expectation is that if we evaluate over enough SCMs we will eventually also get arbitrarily large errors due to the violation of the causal sufficiency assumption.

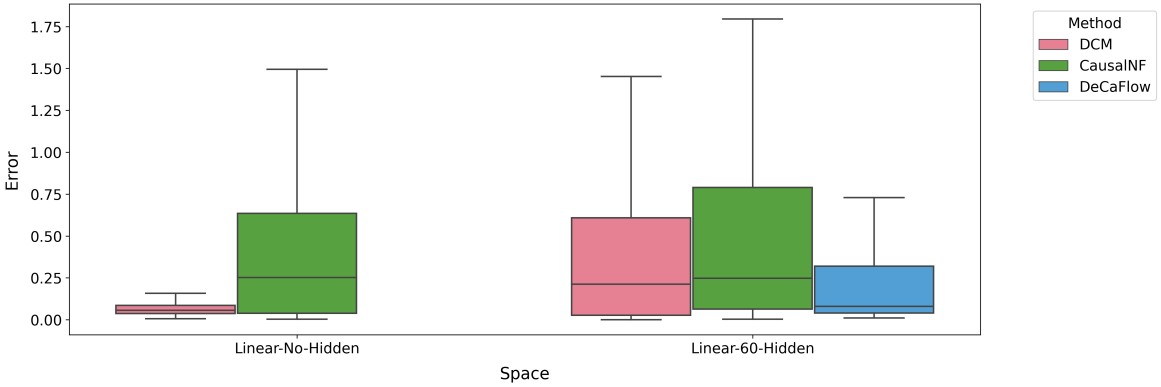

*Figure 16.* Box plots of ATE estimation errors in the presence and absence of hidden confounding. Each box shows the interquartile range and median, with whiskers extending to $1.5\times$ IQR. CausalNF and DCM are shown for both *SoIs*; DeCaFlow is shown only for the hidden setting.

*Table 19.* Performance summary of CausalNF, DCM, and DeCaFlow on the hidden confounder experiments.

| Space | Method | Mean Error | Std Error | Max Error | Runtime (s) |
|---|---|---|---|---|---|
| Linear-No-Hidden | CausalNF | 0.5538 | 0.9866 | 14.2495 | 8570.0 |
| | DCM | 0.0845 | 0.1515 | 2.8954 | 12144.6 |
| Linear-60-Hidden | CausalNF | 2.667e+12 | 5.497e+13 | 1.225e+15 | 293.2 |
| | DCM | 0.5584 | 1.2122 | 17.2049 | 4187.6 |
| | DeCaFlow | 0.3405 | 0.6799 | 5.9435 | 2264.0 |

*Table 20.* Additional performance metrics of CausalNF, DCM, and DeCaFlow on the hidden confounder experiments.

| Space | Method | Min Error | Total Fail | Runtime Mean | Runtime Std |
|---|---|---|---|---|---|
| Linear-No-Hidden | CausalNF | 0.0036 | 0 | 17.14 s | 10.61 s |
| | DCM | 0.0068 | 0 | 24.29 s | 7.64 s |
| Linear-60-Hidden | CausalNF | 0.0029 | 0 | 0.59 s | 0.02 s |
| | DCM | 0.0000 | 0 | 8.38 s | 3.45 s |
| | DeCaFlow | 0.0108 | 0 | 4.53 s | 1.27 s |

## K.4. Experiment 4: Causal Discovery

**Experimental setup.**  We consider continuous linear SCMs with uniformly distributed exogenous noise variables $U(-1, 1)$. We evaluate three representative families of causal discovery methods using the `causal-learn` library (Zheng et al., 2024): (i) PC (constraint-based), (ii) GES (score-based), and (iii) DirectLiNGAM (functional causal model based).

Tables 21 and 22 specify the two *SoIs* used in this experiment. `small1000` corresponds to moderate-sized graphs (10–20 variables) with 1000 samples, and `big50` corresponds to larger graphs (30–50 variables) with only 50 samples. Evaluation results can be seen on Table 23.

*Table 21.* Specification of the `small1000` *SoI* used in the causal discovery experiments. $N$ denotes the sampled number of nodes.

| Name | small1000 |
|---|---|
| # Nodes | 10–20 |
| Mechanism | Linear |
| Expected Edges | $3 \times N$ |
| Variable Type | Continuous |
| Samples | 1000 |
| Seeds | [10, 11, 12, 13, 14] |

*Table 22.* Specification of the `big50` *SoI* used in the causal discovery experiments. $N$ denotes the sampled number of nodes.

| Name | big50 |
|---|---|
| # Nodes | 30–50 |
| Mechanism | Linear |
| Expected Edges | $3 \times N$ |
| Variable Type | Continuous |
| Samples | 50 |
| Seeds | [10, 11, 12, 13, 14] |

| Space | Method | SHD | F1 | Precision | Recall | Runtime (s) |
|---|---|---|---|---|---|---|
| small1000 | DirectLiNGAM | 38.21 | 0.60 | 0.62 | 0.58 | 0.21 |
| small1000 | GES | 59.35 | 0.56 | 0.51 | 0.63 | 0.74 |
| small1000 | PC | 51.66 | 0.34 | 0.56 | 0.25 | 0.11 |
| big50 | DirectLiNGAM | 302.87 | 0.20 | 0.16 | 0.30 | 2.15 |
| big50 | GES | 225.13 | 0.18 | 0.17 | 0.20 | 42.12 |
| big50 | PC | 146.33 | 0.08 | 0.18 | 0.05 | 0.42 |

*Table 23.* Causal discovery performance across two representative Spaces of Interest. Results illustrate that relative method performance can vary substantially across evaluation regimes.

**Discussion.**  In the `small1000` setting, DirectLiNGAM achieves the strongest overall performance, obtaining the lowest SHD and highest F1 score. This suggests that its structural assumptions are beneficial when sufficient data is available. However, in the more challenging `big50` setting, performance deteriorates substantially across all methods and the relative ranking changes: PC achieves the lowest SHD, while DirectLiNGAM degrades considerably despite the underlying data satisfying its modeling assumptions (linear continuous mechanisms). Further, GES exhibits a substantial increase in runtime in the larger setting, illustrating known scalability limitations of score-based approaches. Overall, these results highlight that conclusions drawn from a single evaluation regime may not generalize across Spaces of Interest, motivating the need for controlled and diverse synthetic evaluation settings.

## K.5. Time and Space Complexity of Experiment 1

We provide a time and space complexity analysis based on a setting consistent with the experimental setup of Experiment 1. Assume a continuous SCM where each mechanism is modeled as a 2-layer neural network (with hidden size 8), the

dimensionality of each variable is 1, the expected number of edges scales linearly with the number of variables, and the queries are ATE. More details on the exact SoI can be found in Appendix K.1.

Parameters: $V$ = Number of variables, $E$ = Expected number of edges per variable, $N$ = Number of samples, $Q$ = Number of queries.

**Time Complexity.**

- **Graph generation:** $O(V^2)$

- **NN initialization (per variable):** $O(E)$

- **NN inference (per variable, per sample):** $O(E)$

- **Sample generation:**
    - For each of the $N$ samples, we:
        * sample noise
        * run a topological sort (once)
        * evaluate each of the $V$ variables via a forward pass through a neural network with on average $E$ inputs, so the cost per sample is $O(V \cdot E)$
    - Hence, in total $O(N \cdot V \cdot E)$.

- **Query generation and evaluation:** $O(Q \cdot N \cdot V \cdot E)$

- **Overall dominant term (worst case):** $O(Q \cdot N \cdot V \cdot E)$

This scaling is intuitive: each query requires $N$ samples, where each sample involves computing all $V$ variables, and each variable depends on approximately $E$ parents through a neural network. Note that in this setting the $O(V^2)$ graph-generation term is always dominated, since $VE = \Theta(V^2)$.

In practice, we get constant-time speedups using vectorized operations and batch processing, e.g., we do not have a loop over samples but process them by batch.

**Space Complexity.**

- **Graph structure:** $O(V + E)$

- **NN parameters:** $O(V \cdot E)$

- **Sample storage:** $O(N \cdot V)$

- **Query outputs:** $O(Q)$, working memory to compute a single query: $O(Q \cdot V \cdot N)$

- **Total:** $O(V \cdot E + N \cdot V + Q)$

### K.6. Runtime Scalability on Larger Graphs

We additionally evaluate the scalability of CausalProfiler with respect to the number of variables. Batch processing and vectorized operations enable efficient dataset generation even for graphs with hundreds of variables. Table 24 reports the average generation time (over 5 runs) for producing 10,000 samples and 50 queries (each estimated using 10,000 additional datapoints), using the same CPU hardware described in Section 6.3.

For completeness, Table 25 reports the runtime of each evaluated method in Experiment 1 (Section 6.3.1) on the *NN-Large SoI* as the number of nodes increases (with the expected number of edges fixed to $N$, the number of nodes). While some methods scale better than others, dataset generation with CausalProfiler remains efficient.

The apparent reduction in average VACA runtime is explained by its increasing failure rate. All other methods exhibit a $0\%$ failure rate.

*Table 24.* Average runtime (seconds) of CausalProfiler for generating datasets across increasing numbers of variables. Each value is the mean over 5 runs with standard deviation in parentheses.

| Num Variables | Mean Time (s) | Std Dev (s) |
|---|---|---|
| 10 | 0.19 | 0.01 |
| 50 | 0.89 | 0.03 |
| 100 | 1.81 | 0.03 |
| 500 | 9.61 | 0.11 |
| 1000 | 19.24 | 0.21 |

*Table 25.* Runtime scaling of causal inference methods (in seconds). Each entry reports mean and standard deviation across runs.

| Node Range | CausalNF | DCM | NCM | VACA |
|---|---|---|---|---|
| 30–40 | (1, 0.6) | (24, 8.4) | (12, 1.2) | (11, 4.6) |
| 50–70 | (2, 0.4) | (43, 12.8) | (22, 2.6) | (12, 4.8) |
| 70–90 | (3, 0.3) | (53, 18.0) | (29, 2.4) | (12, 4.7) |
| 90–110 | (4, 0.4) | (60, 23.0) | (36, 2.5) | ( 9, 2.5) |

