# OpenReview forum: "CausalProfiler: Generating Synthetic Benchmarks for Rigorous and Transparent Evaluation of Causal Machine Learning"
_ICML.cc/2026/Conference — ICML 2026 regular_

### Official Review · Reviewer_a8W8 · 2026-03-03

**Soundness:** 2
**Presentation:** 3
**Significance:** 3
**Originality:** 3
**Overall Recommendation:** 4
**Confidence:** 4

**Summary:**

CausalProfiler is a innovative evaluation method for causal machine learning methods. Currently, causal machine learning relies on a variety of semi-synthetic and synthetic benchmarks that may differ between papers. This challenges comparability of progress. CausalProfiler introduces a broad framework that can generate a large number of sample synthetic datasets under various SCMs and assumptions for more comparable and comprehensive evaluation.

**Compliance With Llm Reviewing Policy:**

Affirmed.

**Final Justification:**

Overall, my concerns are addressed in the rebuttal, and I think this is a valuable contribution to the causal inference literature.

**Key Questions For Authors:**

- What is the definition of fail rate?

- What is the mean error in Table 1 on? Factual error? ATE error? CATE error?

- What do medium and large mean for dataset generation? (Section 6.4)

- How big are the ATEs in CausalProfiler datasets? Should errors on ATE be reported in proportion to the effect size to make datasets comparable?

- How many datasets are in each space in Table 1, 2 / Figure 2?

**Limitations:**

Yes

**Strengths And Weaknesses:**

**Strengths**

- The work addresses an important problem in causal ML. The introduction lays this out nicely. (significance, originality)

- The framework is intentionally broad and widely applicable. (significance)

**Weaknesses**

-  There are many causal ML methods and methods for treatment effect estimation. The four chosen miss many and are not specifically justified in the paper. Moreover, a comprehensive set of the current causal inference benchmarks (IHDP, Twins, etc) would also be a nice addition to Figure 1. (presentation, soundness)

- It would be helpful to demonstrate the difference between current evaluation and evaluation with CausalProfiler. Which methods that once seemed best are no longer? On what regimes do they fail that contrasts with what was already known or in original evaluation? (presentation, significance)

- There is no evaluation of changing the structural equations, assumptions, or number of relationships between variables in quantifiable ways. The only evaluation is on data size and linear versus non-linear. (presentation, soundness)

- It is not clear the value add of Proposition 5.1 for the purposes of better understanding evaluation. What is a reasonable number of datasets to generate for sufficient evaluation? This would be a more useful result. I also think Section 6.1 could be relegated to the appendix. (presentation)

- It is not entirely clear how to go from the broad framework in Definition 4.1 to the benchmark generation, i.e. what assumptions are chosen (H) and other particulars not completely explained. (soundness, presentation)

---

> ### Author Rebuttal · Authors · 2026-03-31
>
> We thank the reviewer for their thoughtful feedback and constructive questions.
>
> # W1. Choice of methods and comparison to existing benchmarks
>
> We agree that our selection of methods is not exhaustive. Our goal is not a comprehensive experimental study, but to demonstrate analyses enabled by CausalProfiler. That said, we are happy to include additional methods in the final revision and welcome suggestions.
>
> We also ran a causal discovery experiment to reflect broader applications (see W4 of reviewer p8Ac, omitted for space).
>
> We do not include semi-synthetic datasets (e.g., IHDP, Twins) in Figure 1, as the visualization relies on ground-truth SCM properties unavailable for these datasets. Meaningful comparison would require a synthetic-to-real bridge, discussed in Section 7.
>
> # W2. "Which methods are no longer best?" and failure regimes
>
> We clarify that a "best" method is not well-defined in causal inference, as performance depends on assumptions.
>
> This precisely motivates CausalProfiler: different SoIs yield different rankings. Rather than a single ranking, it enables users to identify success/failure regimes, assess sensitivity to assumptions, and explore trade-offs.
>
> Our experiments illustrate this. In Experiment 1, DCM has the lowest error in Linear-Medium but performs worst in NN-Medium. We also observe distinct failure modes: DCM is sensitive to data scarcity, while VACA is more stable but exhibits higher failure rates.
>
> # W3. Limited range of evaluated variations
>
> We agree that additional dimensions can be explored. First, we clarify that the experiments in the paper are a subset of a larger number of experiments. In total, we evaluated the methods across 48 distinct SoIs spanning mechanism types (linear, neural), graph sizes (5-110 nodes), sparsity levels, dataset sizes (50-5000), hidden confounding (0-60%), and multiple sampling strategies.
>
> We selected the reported spaces as a small set that illustrates interesting insights without duplication. Our goal is to demonstrate the evaluation paradigm rather than provide exhaustive benchmarking. We are happy to include full results in an appendix.
>
> # W4. Value of Proposition 5.1 and number of datasets
>
> We thank the reviewer for this question.
>
> Proposition 5.1 provides a minimal expressiveness guarantee: the generator can represent all SCMs in the Space of Interest. While modest, we include it since the properties of the data-generating process can be controlled and analyzed. We will clarify its positioning in the text.
>
> There is no universal number of datasets: it depends on the SoI, task variability, and method sample efficiency. In practice, one can monitor metrics or learning curves and increase samples until estimates stabilize. We will clarify this guideline in the revision.
>
> # W5. From Definition 4.1 to benchmark generation
>
> We agree this transition could be clearer.
>
> In summary:
> - The Space of Interest (Definition 4.1) specifies distributions over SCMs, queries, and data
> - Sampling (Section 5.3 and App. C-E) generates:
>   (i) graph
>   (ii) mechanisms
>   (iii) datasets
>   (iv) queries with ground-truth
> - Assumptions $H^*$ arise from:
>   (a) SoI parameters (e.g., variable types, hidden variables)
>   (b) properties identified via the assumption-analysis module.
>
> We will add a concise pipeline summary in the main text with pointers to the appendix.
>
> # Q1. Definition of fail rate
>
> The fail rate is the proportion of runs in which a method does not return a valid estimate (e.g., due to numerical instability). We will clarify this definition in the experimental section.
>
> # Q2. Definition of mean error
>
> The reported error is the mean squared error between the estimated and ground-truth Average Treatment Effect.
>
> # Q3. Meaning of "Medium" and "Large"
>
> "Medium" and "Large" refer to the size of the causal graph in the SoI (15-20 and 20-25 variables respectively).
>
> # Q4. Magnitude of ATEs and normalization
>
> Normalization by effect size can be useful when comparing across SoIs with different scales, and CausalProfiler supports such custom metrics.
>
> However, it can over-emphasize small effects. For example, if the true ATE is $0.01$ and the estimate is $0.1$, the absolute error is $0.09$ (MSE $= 0.0081$), while the relative error is $9$, disproportionately penalizing a moderate deviation.
>
> In some experiments, ATE scales are similar (e.g., Linear-Medium: median $0$, mean $\pm$ std $= 0.0129 \pm 1.0099$; Linear-60-Hidden: median $0$, mean $\pm$ std $= 0.0102 \pm 0.9539$), making MSE a natural choice. In others, normalization may be more informative.
>
> ATE scale also depends on modeling choices (e.g., weights, cardinalities), so normalization effectively re-weights regions of the SoI. Overall, we view this as a case-by-case design choice rather than a one-size-fits-all answer. We will clarify this in the paper.
>
> # Q5. Number of datasets per SoI
>
> Each result in Tables 1/2 and Figure 2 is based on 1000 sampled datasets.

---

> > ### Author Rebuttal · Reviewer_a8W8 · 2026-04-02
> >
> > I think evaluating standard meta learners for heterogenous effects would all be interesting in addition to adding evaluation of deep learning methods i.e. TARNet or DragonNet. I also still think that evaluation on standard semi-synthetic benchmarks would be a nice comparison to existing practice. Clarification of the fact that Proposition 5.1 has no evaluation guarantees, and it is not yet known a good number of datasets to evaluate would be good to emphasize as limitations. Overall, my concerns are addressed in the rebuttal, and I think this is a valuable contribution to the causal inference literature. I have raised my score to weak accept.

---

### Official Review · Reviewer_Y1ig · 2026-03-13

**Soundness:** 4
**Presentation:** 3
**Significance:** 3
**Originality:** 3
**Overall Recommendation:** 5
**Confidence:** 4

**Summary:**

The intersection of causality and machine learning is a topic of much research, yet benchmarking remains a problem, with many results obtained in specific settings and environments not transferring to other applications, sometimes even when the required assumptions are met. The proposed CausalProfiler is a benchmark generator for causal datasets that supports a large variety of parameters, assumptions, and metrics to help evaluate causal models in a wide variety of settings. It considers all three rungs of the causal hierarchy and guarantees coverage for discrete problems. In a series of experiments, it is shown how CausalProfiler generates diverse problems and how it can be used for benchmarking causal models.

**Compliance With Llm Reviewing Policy:**

Affirmed.

**Final Justification:**

The paper is written well and the framework is very extensive and extendable. While successful application of causal methods on this framework does not guarantee real-world applicability (see the discussion in the rebuttal here), CausalProfiler is valuable in that methods can be analyzed more rigorously and extensively. Having a comprehensive framework on synthetic data generation published can help improve evaluation of new (and existing) methods, providing a more balanced assessment of their advantages and disadvantages and, to an extent, also reducing the gap between synthetic and real-world settings.

**Key Questions For Authors:**

1. Given the problems with relying on synthetic benchmarks in causal research (e.g., [4]), how do the authors think that a broad synthetic benchmark generator can help overcome these problems? Is the goal of this benchmark rather to allow a fairer and more general comparison of different models on synthetic problems or is there more to be gained when it comes to generalizing from results on this benchmark to real-world problems? (weakness 1)
2. What are the patterns for CausalProfiler in Figure 1? Why is there still so much open space? I was wondering whether that could just be the sets of impossible configurations (e.g., too many edges for small sets of nodes), but then the CausalNF SCMs seem to lie somewhere, where CausalProfiler has no data. Why is that the case?
3. Does the benchmark include the causal models used in the evaluation (or more)? This would make usage even more convenient, as one could easily compare their own models to those baselines (or add new baselines to the benchmark easily and comfortably).

Minor Question

4. Why is the failure rate for DCM in the experiments so high?

[4] Brouillard, Philippe, et al. "The Landscape of Causal Discovery Data: Grounding Causal Discovery in Real-World Applications." *Proceedings of Machine Learning Research* 275 (2025): 1-40.

---

**Edit after rebuttal:** I increased my score from 4 to 5 (see rebuttal for more information).

**Limitations:**

Yes

**Strengths And Weaknesses:**

**Strengths**

- **S1** Benchmarking causal models is still a significant problem that benefits from well-designed benchmarks. CausalProfiler can help avoid biases and, especially, it is also valuable that one can easily test their causal models for robustness, e.g., evaluate the performance on settings where the theoretical assumptions are (partly) violated. As causal methodology often requires very specific assumptions, analysing the performance in case of assumption violations (where theoretical guarantees are lost, but performance might still be strong empirically) is an important step towards understanding and improving performance on real-world applications.
- **S2** The problem generation of CausalProfiler is sound and the extent of the benchmark generator is very broad. It covers queries from all rungs of the causal hierarchy, supports a long list of metrics (Appendix F), and includes coverage guarantees for discrete data and a diverse range of functions for continuous data. Section 6.2 provides empirical evidence for this coverage. The usage guide (B.2) can help users manage the vast options and parameters when designing their own evaluations.
- **S3** Limitations are spelled out clearly and transparently.
- **S4** The paper is well-written and easy to follow. It's nicely structured and contains a long list of appendices with additional details.

**Weaknesses**

- **W1 Synthetic instead of real-world (major weakness).** While the authors acknowledge this weakness, the benefits of a benchmark for causal models are inherently restricted by the constraint of only considering synthetic but not real-world problems. Despite this paper's strengths, its focus on synthetic benchmarks and missing connection to actual real-world problems limit the significance of its contribution to the causal community. While the coverage guarantees are generally nice, it is unclear how often problems are generated that actually represent real-world problems, as there are no obvious "ground truth statistics" of real-world problems.
  - While it is unreasonable to expect a benchmark with the same extent of options as CausalProfiler with real-world data, the authors could try to make the connection to real-world problems clearer. For example, a longer discussion on which assumptions are commonly met or violated in different domains would be useful (even if it, clearly, could not ever be comprehensive).
  - Another possibility would be adding real-world datasets to the benchmark. While there is no real-world dataset for every setting, these datasets could still be analyzed for their metrics (Appendix F) and, thus, compared to equivalent synthetic experiments. This would help to estimate how well results from synthetic benchmarks generalize to real-world problems.
- **W2 No streamlined usage guide (minor weakness).** While the large amount of options for this benchmark is primarily a strength, I think it can be overwhelming for people that want to use this benchmark, especially if they are not experts in all details regarding causal models (although, to be fair, you would naturally expect people interested in this benchmark to be relatively familiar with most, if not all concepts). The guide in Appendix B.2 is already a great step for helping uncertain users, but support here could be extended. Note that this is not a major weakness, but still something that could improve the benchmark quality and usability even further. For example:
  - There could be a set of default "templates" for typical causal models/sets of assumptions. E.g., a typical linear problem would contain this set of parameters. Then you could have a template for small problem, medium problem, large problem,... sparse, dense... There is no ground truth for such templates, but it could make it more accessible, as the default parameters by themselves might not be as valuable for any kind of problem.
  - Similar to the previous suggestions, but implemented pipelines or suggested experimental series could also add another degree of convenience. E.g., imagine someone designs a method for a specific set of assumptions and then a pipeline could be automatically proposed that ablates different assumptions (robustness), shows how well the method scales by suggesting increasingly large and/or dense causal problems, and automatically writes an experimental report.

**Minor Points**

1. I suggest adding a few references that are related to the benchmarking problems of causality. One showcases problems of relying on synthetic data for causal representation learning [1], the second one discusses a problem of synthetically generated data for causal discovery [2], and the last one is another benchmark that focuses primarily on causal discovery for dynamical systems [3].
2. In 6.1, L3, a reference to the appendix (Definition J.2) for composition, effectiveness, and reversibility would be nice.

Typos

3. Line 721: "Appendix Appendix"
4. Line 794: Missing space before "Hence"
5. The equation in Definition D.2 uses two different parent notations (pa vs PA)
6. Line 1086: "varaibles"
7. Line 1424: $^3$ instead of ":"
8. Line 1500: missing closing brackets
9. Line 2299: missing punctuation

[1] Gamella, Juan L., Simon Bing, and Jakob Runge. "Sanity Checking Causal Representation Learning on a Simple Real-World System." *Forty-second International Conference on Machine Learning*.

[2] Reisach, Alexander, Christof Seiler, and Sebastian Weichwald. "Beware of the simulated dag! causal discovery benchmarks may be easy to game." *Advances in Neural Information Processing Systems* 34 (2021): 27772-27784.

[3] Herdeanu, Benjamin, et al. "CausalDynamics: A large‐scale benchmark for structural discovery of dynamical causal models." *The Thirty-ninth Annual Conference on Neural Information Processing Systems Datasets and Benchmarks Track*.

---

> ### Author Rebuttal · Authors · 2026-03-31
>
> We thank the reviewer for the careful reading of our work and insightful questions. Below, we address the issues raised.
>
> # W1: Synthetic vs real-world
>
> We thank the reviewer for this point and agree that connecting synthetic evaluation to real-world applications is important.
>
> At the same time, synthetic data plays a complementary role [1]: counterfactual ground truth is available, assumptions can be systematically varied, and failure modes can be isolated and potentially interpreted. CausalProfiler is specifically built for this complementary role, as a first step towards more rigorous and transparent evaluation.
>
> Real-world datasets lack ground-truth SCMs, preventing use of our assumption analysis module. Bridging this gap would require a semi-synthetic approach (discussed in limitations), e.g., learning a distribution over Spaces of Interest from observational data.
>
> However, mapping observational data to SCMs is underconstrained and raises challenges in realism, identifiability, and inductive bias. We view this as promising but beyond the scope of this work.
>
> [1] Poinsot, A. et al. "Position: Causal machine learning requires rigorous synthetic experiments for broader adoption", ICML 2025.
>
> # W2: Usage guide
>
> We thank the reviewer for the helpful suggestion.
>
> We will create online documentation (synchronized with the codebase, similar to PyTorch) including tutorials (e.g., reproducing experiments) and suggested workflows.
>
> We intentionally avoided templates to not encouraging users to reuse a small number of configurations and reintroduce the very benchmarking bias we aim to address. However, we will include some SoIs with the tutorials, along with an interactive flowchart to guide users to create an SoI (e.g. variable types, number of variables).
>
> We will also extend Appendix B (e.g., add a counterfactual example after B.2.1. and more usage guidance), and link to the documentation.
>
> We thank the reviewer for the references and will include them and fix typos.
>
> # Q1: Role of synthetic benchmarks vs real-world generalization
>
> We thank the reviewer for highlighting this work.
>
> CausalProfiler can help address some of the issues raised in the paper: narrow datasets, hidden assumptions, lack of assumption-violation testing/study of failure cases, and a lack of diversity. Issues like the realism gap won't be addressed.
> These benefits depend on how the SoIs are specified.
>
> Our view is that synthetic and real-world datasets are complementary: synthetic benchmarks enable controlled access to ground truth and failure analysis, while real-world datasets assess practical applicability.
>
> Regarding generalization to real-world problems: Synthetic benchmarks help understand behavior under controlled assumptions. If these don't correspond to the real world, bridging to real-world performance likely requires semi-synthetic approaches (discussed as future work).
>
> # Q2: Patterns in Figure 1 and empty regions
>
> We thank the reviewer for this observation.
>
> There are a few reasons for the empty space:
>
> 1. t-SNE distortion: t-SNE preserves local neighborhoods but not global distances or density. As a result, non-neighboring points can be placed arbitrarily far apart, leading to large empty regions that do not correspond to true gaps in the underlying space (the embedding can create empty regions that did not exist before).
>
> 2. Infeasible and low-probability configurations (as pointed out by the reviewer)
>
> 3. Distribution mismatch between the benchmarks: CausalProfiler samples SCMs from a user-defined Space of Interest, whereas CausalNF consists of a small, fixed set of models. As a result, CausalNF SCMs may occupy regions of the metric space that are unlikely (or not targeted) under the current SoI, which explains why some of its points appear in areas with little or no CausalProfiler density. For instance, it is possible that over the sampled causal graphs, none of them correspond to the CausalNF SCMs ones.
>
> # Q3: Inclusion of evaluated models
>
> CausalProfiler already includes the training pipelines and evaluation scripts.
> We are happy to also include the trained models in a separate repository.
>
> # Q4: Failure rate of DCM
>
> We thank the reviewer for raising this point. We should clarify that the reported failures for DCM are not due to numerical instability, but correspond to unsupported queries.
>
> DCM performs counterfactual inference via abduction, which requires a factual instance consistent with the conditioning variables ($V_F$). Since $V_F$ may not include all the variables, this instance is obtained by selecting matching samples from the observational data. In some cases, no match exists, so abduction cannot be performed, and the query is unsupported.
>
> This issue can be mitigated, for example by restricting $V_F$ to be sampled from the observational data. We have re-performed the experiment with this change, and DCM has 0% failure rate. We will clarify this distinction in the revision.

---

> > ### Author Rebuttal · Reviewer_Y1ig · 2026-04-03
> >
> > I thank the authors for their detailed response. Regarding Q2, Q3, Q4: these questions have been answered well and I have no follow-up questions. Perhaps small clarifications in the final manuscript are beneficial. Regarding W2, I even agree with the statement that "We intentionally avoided templates to not encouraging users to reuse a small number of configurations and reintroduce the very benchmarking bias we aim to address.", and, therefore, welcome the changes that were proposed instead.
> >
> > In some sense, the "missing real-world data problem" remains, however, the authors' response here led me to readjust my opinion. In my view, CausalProfiler is an excellent synthetic benchmarking framework and, as such, will be an appreciated tool by the causal community. Crucially, I still think that **a method performing well in CausalProfiler does not imply that it will perform well on real-world problems**. In other words, the goal of this framework is not the real-world application. It is the more detailed, intricate, and nuanced evaluation and perhaps categorization of causal methods, **increasing the amount of knowledge regarding the methods' strengths, weaknesses, and limitations**, and reducing limitations of other **synthetic** benchmarking strategies. This is still a very valuable contribution, and, while it does not immediately ensure real-world applicability, it still provides help in that direction, as failure cases in synthetic environments are likely to also persist in real-world scenarios.
> >
> > As I understand it, this is the point the authors were also making, correct? I believe clarifying this and contrasting it to the "real-world goal" early in the paper could also be valuable, as real-world settings are currently only discussed in the final section on limitations.
> >
> > Overall, while, therefore, the real-world problem is still not "solved", I value the paper's contributions for what they are and plan to increase my score from 4 to 5. I do think that this paper **improves synthetic evaluation over existing benchmarks** and, also in its current form, will therefore be a valuable contribution to the community.

---

> > > ### Author Response · Authors · 2026-04-03
> > >
> > > We thank the reviewer for engaging deeply with our rebuttal.
> > >
> > > Yes. The reviewer's understanding is aligned with our intent. CausalProfiler is not a tool for direct real-world deployment, but an evaluation framework for more detailed and assumption-aware evaluation. We view understanding strengths, weaknesses and failure modes in controlled settings as an important step toward real-world application.
> > >
> > > We will revise the introduction accordingly to make this distinction clearer.

---

### Official Review · Reviewer_Ww2L · 2026-03-15

**Soundness:** 2
**Presentation:** 3
**Significance:** 2
**Originality:** 3
**Overall Recommendation:** 3
**Confidence:** 4

**Summary:**

The authors introduce a synthetic benchmark generator for causal machine learning methods. They sample SCMs data and causal queries from user defined interests. It then computes ground truth values for queries like the ATE. The tool provides a guarantee that any discrete causal dataset can be generated under specific maximum noise region settings. The authors demonstrate the generator by benchmarking methods like Causal Normalizing Flows and Diffusion Causal Models across parameter configurations.

**Compliance With Llm Reviewing Policy:**

Affirmed.

**Final Justification:**

Because the issues raised are rooted in the core data generating process they require more than text revisions to address. I have read the rebuttal carefully but my evaluation remains unchanged. I wish the authors the best in developing this project further.

**Key Questions For Authors:**

1. How do you justify comparing causal estimators on datasets where strong positivity is violated 94 percent of the time?

2. Can you provide a coverage guarantee for continuous variables or formally bound the approximation error when using neural network mechanisms?

3. Given that the exhaustive partition strategy is intractable for graphs of practical size what are the actual distributional properties of the datasets generated using the sample rejection strategy?

4. Would you consider adding constraints to the discrete mechanism sampling to enforce smoothness or monotonicity so the synthetic tasks better resemble real world continuous processes?

**Limitations:**

yes

**Strengths And Weaknesses:**

Strengths

- The paper provides a configurable tool for testing causal inference algorithms under varying structural assumptions.
- The inclusion of a latent projection module to simulate hidden confounding is practical.
- The authors clearly separate the structural generation from the query estimation.

Weaknesses

The theoretical coverage guarantee described in Proposition 5.1 is technically trivial and somewhat unusable. The guarantee relies on exhaustive partitioning of the noise space where the number of regions equals the total number of possible mappings from parents to a child variable. This quantity grows superexponentially restricting the guarantee to extremely small discrete graphs. The authors mention in the appendix that this doesn't extend to continuous variables.

The causal framing regarding identifiability is problematic. The authors acknowledge that strong positivity is violated in almost all of their generated datasets. Evaluating causal estimators on non-identifiable queries conflates theoretical unidentifiability with finite sample estimation error. When positivity fails estimators are forced to extrapolate based on their functional inductive biases.

The evaluation metrics are skewed by extreme outliers. For example Diffusion Causal Models yield maximum errors over 33 while the mean is 0.15. Using mean squared error without trimming or analyzing the bounds of the support can make the benchmark unstable.

The discrete mechanisms use uniform independent sampling over the function space. This leads to an unnatural distribution of causal mechanisms that are not monotonous or smooth. The authors note that the generated mechanisms are rarely monotonic or linear. This doesn't reflect physical or social processes, where causal relationships often follow some regularities.

---

> ### Author Rebuttal · Authors · 2026-03-31
>
> We thank the reviewer for the careful reading of our work and the constructive feedback.
>
> # W1. Role of Proposition 5.1
>
> We agree that Exhaustive Partitioning is not tractable for large graphs.
>
> Proposition 5.1 provides a minimal expressiveness guarantee: with sufficiently rich discrete mechanisms the generator can represent all SCMs in the Space of Interest. This requires Exhaustive Partitioning, which scales super-exponentially and is not intended for practical use outside of small settings. CausalProfiler can use more efficient sampling schemes that trade coverage for scalability.
>
> This guarantee is modest, but we believe it is worth making explicit in a synthetic evaluation setting, since the properties of the data-generating process can be controlled and analyzed. We will clarify its positioning in the text.
>
>
> # W2 & Q1: Evaluation under violated positivity and identifiability
>
> We disagree that this is inherently problematic, it depends on the user's objective and is a deliberate design choice.
>
> CausalProfiler evaluates methods both within and outside their identification regime.
> When strong positivity fails, causal queries may be non-identifiable, and estimators rely on inductive biases to extrapolate. However, in practice, such violations are common, and methods are often applied outside their theoretical guarantees. Evaluating performance in these regimes provides insight into robustness, failure modes, and implicit assumptions encoded in different estimators.
>
> This regime is optional: users can enforce positivity via appropriate SoIs (e.g. continuous SCMs with full-support noise, or discrete SCMs with small variable cardinalities or constrained mechanisms) or rejection sampling.
>
> The reported 94% violation is from a highly general SoI designed to maximize expressivity, and is not a limitation of the entire framework. It is also estimated from finite samples and thus conservative.
>
> We will clarify that (i) identifiable regimes can be enforced, (ii) our experiments intentionally probe beyond identifiability, and (iii) results reflect both estimation and extrapolation error.
>
>
>
> # W3: Metrics skewed by outliers
>
> We thank the reviewer for this observation.
>
> Here "outliers" are not anomalous/corrupted/noisy datapoints but instances where a method produces a large error. It's a failure mode rather than a data issue.
>
> We therefore report both mean/std and box plots (median and IQR). The median alone would hide occasional large errors, which are important for assessing robustness.
>
> In the referenced error (DCM in Linear-Medium), both the mean and the median are the lowest of the methods, suggesting that large errors are infrequent. However, their presence remains relevant in practice and may affect method choice.
>
> We agree this was not clearly explained and will revise the text to (i) clarify the role of box plots and (ii) include median-based metrics. Our goal is to expose both typical performance and failure modes, as both are important for practitioners.
>
> # W4 & Q4: Unrealistic discrete mechanisms (non-smooth / non-monotonic)
>
> We agree that when sampling over the space of all functions, monotonic or smooth functions form a negligible subset of all possible mappings.
>
> CausalProfiler is designed to be configurable. Users can already select linear mechanisms in the continuous setting, but not yet in the discrete. These, and further mechanism constraints, can be incorporated by defining restricted mechanism classes. This is not an inherent limitation of the evaluation framework, but a missing feature that we will be happy to add, as we agree this is a valuable extension.
>
> CausalProfiler will be released as an open-source project that will grow with community feedback. We aim to prioritize extensions based on community interest.
>
> # Q2: Coverage guarantee for continuous variables
>
> We thank the reviewer for this question.
>
> We do not believe a strict coverage guarantee (like Proposition 5.1) is achievable for continuous SCMs. The discrete result relies on enumerating all possible parent-to-child mappings, and we're not aware of an equivalent for continuous SCMs. For example, while neural networks are universal approximators, this does not establish that every function has non-zero probability of being sampled.
>
> We agree that formalizing approximation properties (e.g. density or error bounds) for continuous SCMs is an interesting direction for future work, and we are happy to clarify this in the revision.
>
> # Q3: Distributional properties under sample rejection
>
> We note that distributional properties depend on the selected Spaces of Interest. Due to rebuttal constraints (no room for additional figures), we cannot re-present the entire Appendix G, but with a different sampling strategy. If the reviewer is interested in a specific property, we can comment on it directly. We are, however, happy to include an additional appendix in the final revision where we repeat Appendix G but with a different sampling strategy.

---

> > ### Author Rebuttal · Reviewer_Ww2L · 2026-04-04
> >
> > I thank the authors for their detailed response. I appreciate the clarifications regarding the design choices, but my core concerns about the evaluation framework remain.
> >
> > Evaluating causal estimators in regimes where strong positivity fails shifts the task from causal identification to functional extrapolation. Testing models on unidentifiable queries primarily measures how well an estimator aligns with the specific neural network or tabular function generating the synthetic data.
> >
> > Regarding Proposition 5.1 the acknowledgment that the exhaustive partition strategy is computationally intractable for standard graphs and lacks a continuous equivalent aligns with my initial assessment. The theoretical guarantee is too restricted to provide a working foundation for the proposed tool.
> >
> > Additionally the current sampling of discrete mechanisms does not enforce properties like smoothness or monotonicity. Unconstrained random mappings struggle to simulate realistic processes. Leaving these structural constraints to future work limits the practical utility of the discrete benchmark.

---

> > > ### Author Response · Authors · 2026-04-05
> > >
> > > We thank the reviewer for the follow-up. We briefly expand on the remaining points below.
> > >
> > > # 1. Evaluation under violated assumptions (e.g. positivity)
> > >
> > > We respectfully note that **evaluation under violations of identifying assumptions has been studied in both causal inference [1,2] and causal discovery [3,4]**. Such analyses are valuable for empirically characterizing robustness, inductive biases, and failure modes. This is particularly relevant in practice, where assumptions such as strong positivity may be especially difficult to satisfy in realistic high-dimensional settings [5].
> > >
> > > In addition, we would like to emphasize that **the assumption-violation regime in CausalProfiler is optional**. Users can construct SoIs where positivity always holds or enforce it via constrained sampling (e.g. discarding datasets that violate the desired property). Regarding our specific experiments: Experiments 1-3 use regimes where assumptions are violated to understand how methods behave beyond idealized conditions. We note that, as part of the rebuttal, we ran additional experiments in causal discovery where the assumptions of the methods were satisfied (see W4 of reviewer p8Ac, omitted for space). We are also happy to include *an additional causal inference experiment in the Appendix where positivity is satisfied*.
> > >
> > > Finally, real-world datasets are the gold standard for real-world evaluation. Synthetic data plays a complementary role: counterfactual ground truth is available, assumptions can be systematically varied, and failure modes can be isolated and interpreted. CausalProfiler is specifically built for this complementary role. CausalProfiler is not a tool for direct real-world deployment, but an evaluation framework for more assumption-aware evaluation. We will revise the introduction to make this distinction clearer.
> > >
> > > [1] Genbäck M, de Luna X. Causal inference accounting for unobserved confounding after outcome regression and doubly robust estimation. Biometrics. 2019;75:506–515. https://doi.org/10.1111/biom.13001
> > >
> > > [2] Poinsot, A. et al. "Position: Causal machine learning requires rigorous synthetic experiments for broader adoption", ICML 2025.
> > >
> > > [3] Montagna, Francesco, et al. "Assumption violations in causal discovery and the robustness of score matching." Advances in Neural Information Processing Systems 2023.
> > >
> > > [4] Yi, Huiyang, et al. "The Robustness of Differentiable Causal Discovery in Misspecified Scenarios." ICLR 2025.
> > >
> > > [5] D'Amour, Alexander, et al. "Overlap in observational studies with high-dimensional covariates." Journal of Econometrics 221.2 (2021): 644-654.
> > >
> > > # 2. Role of Proposition 5.1
> > >
> > > It is not clear to us what the reviewer refers to by "standard graphs". Exhaustive partitioning is tractable under small graphs. We highlight that **Proposition 5.1 is not a foundation for the proposed tool** (i.e. the evaluation can work without it, no further results depend on it). It exists, as a formal property of the generator, to guarantee coverage to all possible functions in a subset of cases (in small discrete graphs). For larger graphs, other sampling strategies are implemented that trade off full coverage for efficiency (e.g. unbiased random assignment).
> > > This Proposition may not be relevant for all settings (e.g. large-scale continuous settings), but it is included because it is relevant for characterizing discrete synthetic evaluation and for understanding the limits of mechanism sampling. For example, it highlights that coverage is not automatic and other reasonable sampling schemes may fail to cover the space.
> > >
> > > # 3. Discrete mechanism realism (monotonicity / smoothness)
> > >
> > > Unconstrained function sampling is used as a default to maximize expressivity and is not a fundamental limitation of our framework. In particular, **CausalProfiler is explicitly designed to allow restricting mechanism classes** (e.g. linear mechanisms or other task-specific mechanism classes). Additional constraints on discrete functions, such as monotonicity or smoothness, can be added easily. For instance, monotonicity can be enforced by sampling output values and sorting them before assigning them to ordered input states (breaking ordering ties arbitrarily), which requires a relatively small code modification. We view such restrictions as natural extensions of the framework rather than inherent limitations of our approach. We are happy to add both of the mentioned constraints for discrete functions for the camera-ready version, if the paper is accepted.
> > >
> > > CausalProfiler will be released as an open-source project and is designed to evolve with community needs and contributions. In particular, everyone will be able to extend the framework by adding missing functionality, as it is not feasible to implement all possible features from the start.

---

### Official Review · Reviewer_p8Ac · 2026-03-15

**Soundness:** 3
**Presentation:** 3
**Significance:** 3
**Originality:** 2
**Overall Recommendation:** 5
**Confidence:** 4

**Summary:**

This paper proposes CausalProfiler, a framework for evaluating causal ML methods through diverse synthetic generation. The authors introduce the Space of Interest (SoI) abstraction for users to specify the class of SCM, queries, and data rather than manual construction. Furthermore, this framework offers coverage guarantees and assumptions on all three rungs of Pearl’s causal hierarchy, including those that fall out of the identifiability regime.

**Compliance With Llm Reviewing Policy:**

Affirmed.

**Final Justification:**

My concerns have been sufficiently addressed. I thank the authors for providing clarifications and additional results. I believe including these in a revised version of the paper will make it stronger. The paper is well written and tackles an important problem. I have increased my score to Accept.

**Key Questions For Authors:**

- How does the CausalProfiler scale with respect to the number of causal variables?
- Does this approach support sampling from interventional SCMs?


I am more than willing to raise my score after clarification from the authors regarding my concerns in weaknesses and questions.

**Limitations:**

The authors provide a detailed limitations and future work section that proposes several extensions to the current work.

**Strengths And Weaknesses:**

**Strengths**

- The paper is very well written and clearly motivates why robust evaluation frameworks are necessary to make reliable decisions from causal models and data.
- The SoI abstraction provides a flexible way to describe user specifications through a simple combination of SCM class, query type, and data characteristics.
- Causal ML seems to largely be constrained to theoretical formulations and methodological contributions without concrete steps for real-world use-cases. However, this paper tackles the causal ML problem from the practitioner’s viewpoint, where the ability to choose the right causal model and assumptions for each problem is critical for the reliable use of any causal ML methods in practice. I believe there is a lot of value in this under-explored direction.
- The verification procedure to ensure that sampled SCMs adhere to Markov property, do-calculus rules, and counterfactual axioms increases the value of this generator framework for sampling high-quality and diverse SCMs.
- Figure 1 is an excellent illustration of why diversity of sampling is critical for reliable use of causal ML methods. The authors point out that it is possible we sample an SCM that our model overfits to, but have no guarantee of generalizability to other SCMs with the same structure. This is an interesting point and the random sampling with coverage guarantees directly addresses this concern.
- The experimental evaluation with several causal ML methods for estimation (DCM, VACA, CausalNF, etc.) is rigorous and demonstrates that choosing causal ML methods is highly particular to the assumptions and task. The results show a lot of variation in how the models perform as the SCM assumptions are changed and in low-data settings.

**Weaknesses**

- I am a bit confused about what the NaN value outputs represent. How does this happen when training models and what does it indicate?
- Proposition 5.1 seems to boil down to verifying the positivity assumption for any causal dataset sampled from the SoI. However, the authors mention distributional bias of the approach where some SCMs may have low likelihood to be sampled. It is not clear why this proposition is necessary. I would appreciate a clarification from the authors.
- This paper mainly focuses on synthetic data generation, but obtaining real-world datasets from which we can extract causal insights is ideally the gold standard. However, I understand the limitations of getting high-quality real-world data, and synthetic simulations are often necessary.
- The experiments seem to be largely constrained to causal effect estimation and do not consider causal discovery or more complex data scenarios (e.g., access to interventional dataset). I believe including more tasks would make the benchmark more rigorous. It would also better justify why coverage is so important for causal ML. Experimenting on a few more datasets would also further illustrate this point. Is there a specific reason why a wider range of tasks/scenarios were not included?

---

> ### Author Rebuttal · Authors · 2026-03-31
>
> We appreciate the reviewer's careful reading and thoughtful comments on our work.
>
> # W1. NaN values
>
> NaN (Not-A-Number) values indicate that a method failed to return a valid numerical output due to numerical issues (e.g. gradient explosion, division by near-zero number) or runtime exceptions. These failures reflect instability under certain assumptions or data regimes.
>
> For example, in Experiment 1, VACA sometimes fails with an exception mentioning that the decoder network outputs NaN. Input normalization did not resolve this.
>
> We agree that the term "NaN" might be unclear or overly implementation-specific, and will rephrase this for clarity.
>
> # W2. Role of Proposition 5.1
>
> As the reviewer notes, sampling is not uniform, and some SCMs have low probability of being sampled. The proposition does not address this.
>
> Instead, it provides an expressiveness guarantee: the generator can represent all SCMs within the specified Space of Interest (SoI). Achieving this requires sampling mechanisms using "Exhaustive partitioning", as naive approaches such as "Unbiased Random Assignment" may not cover the full space.
>
> This guarantee is modest, but we believe it is worth making explicit in a synthetic evaluation setting, since the properties of the data-generating process can be controlled and analyzed. We will clarify its positioning in the text.
>
> # W3. Synthetic vs real-world data
>
> We agree that real-world datasets are the gold standard for evaluating causal methods. Synthetic data plays a complementary role [1]: counterfactual ground truth is available, assumptions can be systematically varied, and failure modes can be isolated and interpreted.  CausalProfiler is specifically built for this complementary role. We also agree that bridging to real-world settings is an important direction, and discuss it in Section 7 (Limitations and Future Work).
>
> [1] Poinsot, A. et al. "Position: Causal machine learning requires rigorous synthetic experiments for broader adoption", ICML 2025.
>
> # W4. Scope of experiments
>
> We thank the reviewer for raising this point and agree that broader tasks are valuable. We focused on causal effect estimation as it is widely studied, practically relevant, and most importantly, highlights that our framework gives access to both interventional and counterfactual ground truths.
>
> First, we clarify that the experiments in the paper are a subset of a larger number of experiments. In total, we evaluated the methods across 48 distinct SoIs spanning mechanism types (linear, neural), graph sizes (5-110 nodes), sparsity levels, dataset sizes (50-5000), hidden confounding (0-60%), and multiple sampling strategies.
>
> We selected the reported spaces as a small set that illustrates interesting insights without duplication.
> Our goal in this paper is not to provide a comprehensive benchmarking of all tasks and methods, but to demonstrate the evaluation paradigm.
> We are happy to include the full results in an appendix or repository.
>
> Following the reviewer suggestion, we ran causal discovery experiments on continuous linear SCMs with $U(-1, 1)$ noise and edge weights sampled from $±[0.5, 2]$. We compare PC, GES, and DirectLiNGAM on two spaces: small1000 (10-20 nodes, 1000 samples) and big50 (30-50 nodes, 50 samples).
>
> | Space|Method|SHD|F1|Precision|Recall|Avg Runtime (s)|
> |-----------|----------------|--------|-------|-----------|--------|-------------|
> |small1000|DirectLiNGAM|38.21|0.60|0.62|0.58| 0.21|
> |small1000|GES|59.35|0.56|0.51| 0.63| 0.74|
> |small1000|PC|51.66|0.34|0.56| 0.25| 0.11|
> |big50|DirectLiNGAM|302.87| 0.20|0.16|0.30| 2.15|
> |big50|GES|225.13|0.18|0.17| 0.20|42.12|
> |big50|PC|146.33|0.08|0.18| 0.05|0.42 |
>
> In small1000, DirectLiNGAM performs best (lowest SHD, highest F1), indicating the usefulness of its structural assumptions when sufficient data is available. However, in big50, performance deteriorates sharply across all methods, and the ranking reverses. This illustrates that performance depends strongly on the SoI, and conclusions from a single regime may not generalize.
>
> # Q1. Scaling with number of variables
>
> Complexity analysis is provided in Appendices K.4-K.6. Here, we summarize the result for continuous SCMs with neural mechanisms (as in Experiment 1):
>
> Let $V$ be variables, $E$ edges per variable, $N$ samples, and $Q$ queries.
>
> Time:
> $O(Q \cdot N \cdot V \cdot E)$
>
> Space:
> $O(V \cdot E + N \cdot V + Q)$
>
> We also evaluated runtime scaling empirically. On the same hardware as the paper, generating 10k samples and evaluating 50 queries takes ~0.2s for 10 variables, ~0.9s for 50, ~1.8s for 100, ~9.6s for 500, and ~19.2s for 1000.
>
> # Q2. Sampling from interventional SCMs
>
> Yes, the approach supports sampling from interventional SCMs.
> For continuous SCMs, there is an open question about how the intervention values should be specified. On this end, we note that CausalProfiler will evolve organically with community feedback.
>
>
> We hope these clarifications address the reviewer's concerns.

---

> > ### Author Rebuttal · Reviewer_p8Ac · 2026-04-02
> >
> > My concerns have been sufficiently addressed. I thank the authors for providing clarifications and additional results. I believe including these in a revised version of the paper will make it stronger. I have increased my score accordingly.

---

### Decision · Program_Chairs · 2026-04-30

**Decision:**

Accept (regular)

**Comment:**

After the discussion phase, most reviewers agree that this paper presents a valuable addition to the field benchmarking causal ML methods. The main unresolved concern of one reviewer concerns the violation of strong positivity in the majority of the generated datasets, which renders the causal estimand unidentifiable and shifts the focus to extrapolation and inductive biases of the function class. While this point is valid, the authors (and other reviewers) point out that positivity rarely holds in practice but that the benchmark settings can also be chosen to enforce it.

After careful consideration and in light of the otherwise positive reception, I think the merits outweigh the weaknesses. Under the condition that the authors follow through on the modifications promised during the discussion and, in particular, include a paragraph/section in the main paper of the camera ready version that transparently discusses the role of positivity violations and how this relates to what properties of a method are actually being evaluated (i.e., estimation accuracy in identifiable settings vs. inductive biases in unidentifiable ones), I therefore recommend acceptance and think that the revised version and accompanying open source tool will be a valuable contribution.